# Emerging Therapies for Acute Myelogenus Leukemia Patients Targeting Apoptosis and Mitochondrial Metabolism

**DOI:** 10.3390/cancers11020260

**Published:** 2019-02-22

**Authors:** Germana Castelli, Elvira Pelosi, Ugo Testa

**Affiliations:** Department of Oncology, Istituto Superiore di Sanità, Viale Regina Elena 299, 00161 Rome, Italy; germana.castelli@iss.it (G.C.); elvira.pelosi@iss.it (E.P.)

**Keywords:** apoptosis, acute myeloid leukemia, leukemia, TRAIL, BCL-2, MCL-1

## Abstract

Acute Myelogenous Leukemia (AML) is a malignant disease of the hematopoietic cells, characterized by impaired differentiation and uncontrolled clonal expansion of myeloid progenitors/precursors, resulting in bone marrow failure and impaired normal hematopoiesis. AML comprises a heterogeneous group of malignancies, characterized by a combination of different somatic genetic abnormalities, some of which act as events driving leukemic development. Studies carried out in the last years have shown that AML cells invariably have abnormalities in one or more apoptotic pathways and have identified some components of the apoptotic pathway that can be targeted by specific drugs. Clinical results deriving from studies using B-cell lymphoma 2 (BCL-2) inhibitors in combination with standard AML agents, such as azacytidine, decitabine, low-dose cytarabine, provided promising results and strongly support the use of these agents in the treatment of AML patients, particularly of elderly patients. TNF-related apoptosis-inducing ligand (TRAIL) and its receptors are frequently deregulated in AML patients and their targeting may represent a promising strategy for development of new treatments. Altered mitochondrial metabolism is a common feature of AML cells, as supported through the discovery of mutations in the isocitrate dehydrogenase gene and in mitochondrial electron transport chain and of numerous abnormalities of oxidative metabolism existing in AML subgroups. Overall, these observations strongly support the view that the targeting of mitochondrial apoptotic or metabolic machinery is an appealing new therapeutic perspective in AML.

## 1. Introduction: Apoptosis

Apoptosis is a physiological cellular mechanism for programmed cell death. Recently, it has become apparent that necrosis can also be a programmed, regulated form of cell death, and various types of programmed necrosis have been identified, including necroptosis, pyroptosis, ferroptosis, mitotic catastrophy and autophagic cell death. A recent study provides an updated view on the current criteria, morphological, biochemical and functional, required to define unambiguously the various types of cell death; this considerable effort has provided a widely accepted nomenclature for cell death, a field in continuous development [1].

Although each of these forms of programmed cell death has an important physiological role, it is evident that apoptosis and necroptosis are the two cell death processes with a more consistent impact on cellular and tissutal homeostasis and also the most investigated in normal and pathologic conditions. A deregulation of apoptosis has been identified as a key pathogenic mechanism in various pathologic processes and blocking or evasion of apoptosis is currently considered as one of the properties acquired during malignant transformation.

The apoptotic signaling is executed through two different signaling pathways, the extrinsic and the intrinsic signaling pathways (Figure 1). The extrinsic pathway is triggered by changes occurring in the extracellular microenvironment and is triggered by two types of membrane receptors: death receptors activated through the binding with specific death ligands pertaining to the Tumor Necrosis Factor (TNF) family; dependence receptors activated through the drop of the concentration of their specific ligand, below a critical threshold. Death ligands include Fas Ligand, TNF-related apoptosis inducing ligand (TRAIL) and Tumor Necrosis Factor (TNF) which binds to their respective receptors (TRAIL-Rs, Fas and TNF-Rs, respectively). Following the binding of death ligand to their receptors, adaptor proteins are recruited to the death receptor, including Fas-associated death domain (FADD) and TNF receptor-associated death domain signaling complex (DISC). The DISC represents a molecular platform regulating the activation and function of activator caspases-8 and -10. Pro-caspases-8 and -10 have a death effector domain (DED) through which bind to the adaptor protein, interacting with its DED. Executioner caspases, including caspases-3, -6 and -7 are activated and start the cleavage of various proteic substrates and cytoskeleton, ultimately leading to cell death.

Extrinsic apoptosis follows two different pathways (Figure 1), cell-type specific: in type I cells, caspase-8 induces a proteolytic maturation of executioner caspase-3 and caspase-6, sufficient to induce cell death, not inhibitable by overexpression of antiapoptotic B-cell lymphoma 2 (BCL-2) proteins; in type II cells, caspase-3 and caspase-7 activation is restrained by the antiapoptotic XIAP protein and induction of cell death requires the caspase-8-mediated cleavage of BID (BH3-interacting-domain), with consequent generation of a truncated form of BID, tBID, migrating at the level of outer mitochondrial membrane where it acts as a BH3-only activator, leading to caspase-9 activation via BAX/BAK (BCL2-associated X/BCL2-antagonist/killer)-dependent mechanism [2]. 

DISC activity is regulated by FLICE-like inhibitory protein (FLIP), a caspase-8 homologue that is capable of competing with caspase-8 for binding to FADD, but unable to induce death signaling because it lacks caspase activity [3]. FLIP is present in two molecular forms, known as long (FLIP_L_) and short (FLIP_S_) splice forms. FLIP is frequently overexpressed in solid and hematologic malignancies, in which its high expression is often associated with a poor prognosis [3].

In addition to cell death by apoptosis, death receptors can also induce cellular death by necroptosis, a form of regulated necrosis that can be activated by ligands of death receptors and various stimuli inducing the expression of death receptor ligands under conditions in which apoptosis is inhibited or deficient [4] (Figure 1). Activation of necroptosis involves a molecular pathway different from apoptosis and implies the kinase activity of Receptor Interacting Serine/Threonine Kinase (RIPK1), which triggers the activation of RIPK3 and Mixed Lineage Kinase Domain-Like (MLKL), two essential downstream mediators of necroptosis [4]. Necroptosis is inhibited by caspase-8/FADD-mediated apoptosis, as well as by necrostatins, inhibitors of RIPK1 [4]. In line with these findings, deficiency in caspase-8, as well as in FADD, leads to embryonic lethality, tissue degeneration and inflammation, which can be suppressed by inhibition of RIPK1 [4]. 

Intrinsic apoptosis (also known as mitochondrial apoptosis) can be activated by numerous stimuli, including intracellular damage, oncogenic stress, DNA damaging drugs and oxidants. Cells have a number of cell sensors to respond to perturbations, first by activating compensatory signaling pathways to try to reestablish cell homeostasis and to repair the damage and then, if these reparatory mechanisms are insufficient, by inducing intrinsic apoptosis [5]. The intrinsic apoptotic process is triggered by mitochondrial outer membrane permeabilization (MOMP), which is triggered in response to pores formation induced by BAXZ and BAK (proapoptotic members of the BCL-2 family) or to mitochondrial permeability transition (MPT) [5]. The whole intrinsic apoptotic pathway is regulated by the BCL-2 family of proteins. The critical step of intrinsic apoptosis, MOMP is mediated by BCL-2 associated X Apoptosis Regulator (BAX) and BCL-2 Antagonist Killer (BAK), containing four BH domains: BAX and BAK, together with BOK, are the only BCL-2 family members capable of forming pores across OMM [5]. MOMOP is antagonized by anti-apoptotic members of the BCL-2 family, such as BCL-2, BCL-X_L_, MCL-1; basically, these anti-apoptotic proteins inhibit BAX and BAK by preventing their oligomerization and pore-forming activity directly or indirectly. Thus, the interaction of BCL-2 family members, as well as BCL-2 effector protein regulation by post-translational modifications orchestrate MOMP.

The permeabilization of the mitochondrial membrane determines the release of intermembrane proteins such as cytochrome c, second mitochondria-derived activator of caspase (SMAC) and Omi (a mitochondrial serine protease). Following the release of cytochrome c, the apoptosome is formed from the cytochrome c, the protease activating factor-1 (APAF-1), dATP and procaspase-9; within the apoptosome, procaspase-9 is converted to active caspase-9, which activates the executioner caspases-3 and -7; finally, the activated caspases-3 and -7 start a process of degradation of various cellular targets, leading to cell death.

Apoptosis is frequently deregulated in cancer. The prevention of cancer if one of the main biologic functions of apoptosis and a loss of stringent apoptotic mechanisms favors tumor development, allowing cancer cells to survive longer and increasing the opportunities for the accumulation in these cells of mutations favoring the development of a more aggressive tumor phenotype [6]. Cancer cells evade apoptotic mechanisms through different mechanisms: the most frequent mechanism is represented by overexpression of anti-apoptotic BCL-2 proteins and loss of BAX or BAK; inhibition of caspase function [6]. A very negative consequence of the deregulation of apoptotic mechanism in cancer cells is that these cells become less sensitive to many anticancer drugs; this finding is not surprising since the large majority of traditional anticancer drugs depend on BCL-2/BAX-dependent mechanisms to kill cancer cells [6].

## 2. Acute Myeloid Leukemia (AML)

AML is a malignant disease of the hematopoietic system, characterized by the uncontrolled clonal expansion of undifferentiated myeloid progenitors/precursors, resulting in impaired normal hematopoiesis and bone marrow failure. Studies of whole genome sequencing have shown that AML is a complex, heterogeneous disease, characterized by the presence of many leukemic genes, some recurrently and other infrequently mutated and each patient usually display more than one driver mutation [7]. Like other malignant tumors, the leukemic process evolves over the time, with the coexistence of multiple malignant competing clones [7]. 

For the last four decades, the standard of care of AML has been a 3+7 chemotherapy regimen, based on three days of an anthracycline and seven days of cytarabine [7]. However, the discoveries on the molecular abnormalities of AMLs have markedly improved our understanding of the leukemic process, the classification of this disease and have indicated the opportunity of some targeted therapies for subgroups of AML patients [7].

The natural history of AML development was related to the so-called age-related clonal hematopoiesis (ARCH) occurring by age 70 in about 10% of healthy individuals [8]. The pre-malignant condition predisposes to the acquisition of additional mutations, determining the progressive evolution to a myelodysplastic syndrome (MDS) and to an AML secondary to MDS. Particularly, the studies carried out in these last years have led to propose a sequence of pre-malignant and malignant conditions culminating in AML: initially ARCH, followed by clonal cytopenias of undetermined significance (CCUS), then by MDS and finally, by AML; this sequence of events is not obligatory in that it is possible also the direct jumping from ARCH to AML [9]. Stochastic events, related to aging-associated mutagenesis may initially generate mutations in ARCH; selective pressure on somatic variants dictated by aging or exogenous stress determine clonal outgrowth and ARCH formation; additional mutational events and selection of mutational events associated with a better fitness underline the leukemic progression [9]. Mutational events at the level of genes involved in RNA splicing (*SRFS2*, *serine/arginine-rich splicing factor 2*), DNA methylation (*DNMT3A (DNA(cytosine-5)-methyltransferase 3 A), TET2 (tet methycytosine dioxygenase 2), IDH 1-2 (isocitrate dehydrogenase 1 or 2*)), chromatin modification (*ASXL1*, *ASXL transcriptional regulator 1*) or the cohesion complex (*STAG2*, *stromal antigen 2*) are observed both in ARCH and in MDS; the gain of mutations at the level of genes encoding transcription factors (such as *RUNX1*, runt-related transcription factor 1 and CEBPA, CCAAT/enhancer binding protein α) or signal transduction proteins (such as *FLT3*, fms-like tyrosine kinase 3, *c-kit*) leads to the development of AMLs secondary to MDSs [9]. Patients developing directly de novo AMLs have *RUNX1*, *CEBPA*, *FLT3* or *MLL (mixed lineage leukemia)* mutations, but not mutations associated with MDS [9]. Mutations in epigenetic modifiers *DNMT3A*, *TET2* or *ASXL1* are particularly well-suited to offer a selective advantage over non-mutated clones through a sustained action on self-renewal and differentiation blockade of HSCs (hematopoietic stem cells) [9]. Thus, *TET2* and *DNMT3A* coordinated DNA methylation in stem cells, while *ASXL1* mutations regulate the polycomb repressive complex exerting an important regulatory effect on stem cell biology and homeobox gene regulation [9]. 

Ultra-sensitive sequencing identified a high prevalence of clonal-hematopoiesis-associated mutations throughout adult life, identifying 224 somatic mutations, of which some were in oncogenic driver genes, such as *DNMT3A, TET2, ASXL1, JAK2 (janus activated kinase)* and *GNAS* (*guanine nucleotide-binding alpha subunit*); in a healthy cell population up to 40 years the frequency of ARCH was <5%, from 10 to 12% in the population ranging from 40 to 59 years and about 24% in the population 60 to 69 years old [10]. Recent studies have directly assessed the potential risk conferred by ARCH to develop AML. A recent study reached the conclusion that the presence of ARCH was associated with a clearly increased risk of developing AML: particularly, mutations in *IDH1, IDH2, TP53* (*tumor protein 53*)*, DNMT3A, TET2* and spliceosome genes increased the risk of developing AML; increased progression to AML was seen for those with >1 mutated gene by targeted sequencing (increased complexity) and ≥10% variant-allele fraction; interestingly, all patients with *TP53* or *IDH1/IDH2* mutations developed AML [11]. The median time of AML progression in the studied cohort was of 9.6 years [11]. Abelson and coworkers have analyzed a population of healthy individuals with benign ARCH and a population of pre-AML ARCHs and observed remarkable differences between these two groups: pre-AML samples had more mutations per sample, higher variant allele frequencies, suggesting greater clonal expansion, and showed mutations in specific genes (*U2AF1 (small nuclear RNA auxillary factor 1), TP53, SRSF2*) [12]. These observations suggest that it is possible to discriminate ARCH from pre-AML many years before malignant transformation [12].

Hematopoietic clones harboring specific mutations may expand over time. Different cellular stressors influence the expansion of hematopoietic clones and play a potential important role into the neoplastic progression of these clones. A number of recent studies have shown that different hematopoietic stressors promote the expansion of different long-lived clones, carrying distinct specific mutations, whose leukemic potential is in part related to the mutation they harbor. Thus, cytotoxic therapy results in the expansion of clones carrying mutations in DNA response damage genes, including *PPM1D* and *TP53* [13]. PPM1D, protein phosphatase Mn^2+^/Mg^2+^-dependent 1D, is a DNA damage response regulator that is frequently mutated in clonal hematopoiesis and is present in about 20% of patients with therapy-related AML or MDS; *PPM1D* mutations confer a survival advantage onto hematopoietic clones by rendering them resistant to DNA-damaging agents, such as cisplatin [13]. *TET2*-mutated clones expand under the effect driven by inflammatory cytokines after bacterial infection [14]. Mutant *DNMT3A* clones expand after autologous bone marrow transplantation, while PPM1D mutant clones often decrease in size [15].

It is important to note that clonal hematopoiesis was observed in about 25% of patients with non-hematological malignancies, with 4.5% harboring presumptive leukemia driver mutations [16]. Two studies explored patients who had previously undergone anti-tumor treatment based on chemotherapy either for non-hematological [16] or as part of a conditioning regimen for autologous stem cell transplantation [17]. These studies identified recurrent mutations at the level of epigenetic modifiers (*DNMT3A*, *TET2* and *ASXL1*) and of some genes, such as *TP53, PPM1D, ATM* (*ataxia-teleangectasia mutated*) and *CHEK2* (*checkpoint kinase 2*), all involved in the response to DNA damage [16,17]. *PPM1D* and *TP53* mutations were associated with prior exposure to chemotherapy [16,17]. These studies suggest that expansion of DNA-damage resistant clones occurs under the effect of a genotoxic stress mediated either by chemotherapy or irradiation.

*TET2* gene is one of the genes most frequently mutated in patients with myeloid neoplasia, with most of mutations being truncating mutations leading to *TET2* inactivation [18]. *TET2* mutations were found in 17% of patients with MDS, 46% of MDS/myeloproliferative neoplasms, 19% of myeloproliferative neoplasms, 21% of primary AMLs and 20% of treatment-related myeloid neoplasia. *TET2* mutations increased with age, irrespective of the type of myeloid neoplasia [18]. Interestingly, 43% of the patients with *TET2* mutations displayed more than one *TET2* mutation, with single mutations being more frequent than multiple mutations. *TET2* mutations may be ancestral (>40%) and secondary. In these neoplasia, *TET2* mutations most often occurred with another mutation in *TET2*, and with mutations in *ASXL1, SRSF2* and *NPM1* (*nucleophosmin1*) [18]. MDSs driven by ancestral *TET2* mutant is likely derived from *TET2*-mutant ARCH with a penetrance of about 1%. Following ancestral *TET2* mutations, individual disease course is determined by secondary hits: *ASXL1, EZH2*, and *SF3B1* (*splicing factor 3B subunit 1*) secondary hits are common in MDS; *DNMT3A* and *NPM1* secondary hits are common in AML [18].

There is a clear difference between ARCH-associated and non-ARCH-associated mutations in their capacity to predict AML relapse. In fact, the assessment of measurable residual disease post-induction or post-consolidation therapy is very important and allows to assess, through analysis of leukemia-specific genetic alterations, the efficacy of anti-leukemic therapy and to predict the risk of recurrence [19]. Jongen-Lavrenic et al have explored through next generation sequencing 430 AML patients in complete remission after two cycles of induction therapy; leukemia-specific mutations persisted in 51% of these patients [20]. The mutations detected in these patients were subdivided into those associated with ARCH (such as *DNMT3A, TET2* and *ASXL1*) and others; importantly, the ARCH-associated mutations were usually present in variable allele frequencies, indicating their presence in <5% of bone marrow cells, consistent with the existence of non-leukemic clones [20]. Rothenberg-Thurley and coworkers have characterized paired pre-treatment and remission samples from 126 AML patients for mutations in all major leukemia-associated genes: at remission, 40% of patients retained ≥1 mutation, with a VAF (variant allele frequency) of ≥2%, with mutation persistence most frequent in *DNMT3A, SRSF2, TET2* and *ASXL1*, associated with older age and with an inferior overall survival [21]. Importantly, patients with persisting mutations have a higher cumulative incidence of relapse before, but not after allogeneic stem cell transplantation [21].

The detection of MRD (minimal residual disease) in AML patients who achieve complete remission after standard chemotherapy or allogeneic hematopoietic stem cell transplantation can predict hematological relapse. A recent study showed the use of MRD as a tool for the selection of AML patients to be treated with Azacytidine, after an initial treatment with standard induction therapy. In this study, AML patients were screened for MRD using a quantitative PCR for detection of leukemia-specific abnormalities; patients who developed MRD were treated with azacytidine: relapse-free survival in these patients at 12 months of follow-up was 46%, compared to 88% in the group of MRD-negative patients [22]. These observations support the view that MRD monitoring in AML patients is prognostic and may guide therapeutic options, such as Azacytidine administration, to prevent or delay relapse [22]. Recent studies support a key role for detection of MRD to predict the outcome of MDS patients undergoing HSC transplantation. The risk of disease progression was higher among patients with MRD after HSC transplantation than among those in whom these mutations were not detected [23]. 

The definition of the mutational events and of the clinical dynamics underlying the transition from MDS to AML is of fundamental importance. These studies were based on the serial analysis of MDS patients developing secondary AMLs. The analysis of paired AIDS and sAML samples provided evidence that in all the cases analyzed, progression to sAML was defined by the persistence of an antecedent founding and by the outgrowth or emergence of at least one subclone, harboring dozens to hundreds of new mutations [24]. New driver oncogenic mutations acquired with the progression to AML were mutations of *PTPN11, RUNX1* and *WT1* [24]. A second study carried out on a large set of MDS patients undergoing AML progression showed that the *FLT3, PTPN11, WT1, IDH1, NPM1, IDH2* and *NRAS* (*neuroblastoma-rat sarcoma*) mutations (type-1 mutations) were enriched in sAMLs and tended to be newly acquired and were associated with faster sAML progression and shorter overall survival. In contrast, TP53, GATA2 (GATA-binding factor 2), KRAS (Kristen-rat sarcoma), *RUNX1, STAG2, ASXL1, ZRSR2* (*zinc finger CCCH-type, RNA binding motif and serine/arginine rich 2*) and *TET2* mutations, significantly enriched in high-risk MDSs compared to low-risk MDSs, displayed a weaker impact on sAML progression and overall survival than type-1 mutations [25]. Kim and coworkers analyzed the varying allele frequencies between pre- and post-AML transformation of 124 MDS-to-sAML patients and found that while the total number of mutations increased only moderately, sAML patients often displayed an expansion of previously existing mutations from the MDS condition [26]. The most frequently acquired mutations in sAML samples were observed at the level of activated signaling pathway genes, such as *PTPN11* and *NRAS* [25]. These genes wer usually subclonal in MDS, but tended to be clonal in sAML, as well as ASXL1 and *EZH2* [25]. These findings were confirmed also by Stoch and coworkers, providing evidence that no sAML developed genetically independent of a pre-existing clone present in the pre-existing MDS and that the gained mutations mostly affected genes encoding signaling proteins [27]. A recent study carried out at the level of the leukemic stem cell population showed that leukemic stem cells (LSCs) of MDS patients who have undergone an AML progression have a high subclonal complexity, with distinct subclones contributing to the generation of MDS blasts or progression to AML, respectively [28].

Shiozawa and coworkers have purified CD34^+^ cells from MDS patients and have analyzed their transcriptome; according to the gene expression patterns, MDSs can be subdivided into two groups: an immature progenitor subtype, associated with a high risk of transformation to AML and with *TP53, RUNX1* and *NRAS* mutations; an erythroid megakaryocyte subtype, associated with a low risk of AML transformation [29].

It was estimated that about 24–48% of all AMLs are secondary AMLs, developing from a pre-existing myelodysplastic syndrome. However, about half of these patients present with de novo AML with myelodysplasia-related changes and in these cases, it is not easy to determine whether AML in these patients occurs de novo or if there was a preceding MDS that was clinically silent. A recent study showed that the analysis of mutations at the level of different myeloid cell populations allowed to identify AML with myelodysplasia-related changes: in fact, overlapping mutations, defined as those shared by blast and granulocyte fractions, are markedly enriched in patients with MDS and AML-derived from MDS; in contrast, blast-specific mutations are enriched in patients with de novo AML [30]. Particularly, the presence of overlapping mutations, excluding those of *DNMT3A, TET2* and *ASXL1*, segregated patients with MDS and AML MDS-related from patients with de novo AML [30]. The presence of ≥3 mutations in the blast fraction was a tool to distinguish MDS from AML MDS-related [30].

Initial studies of genomic characterization (The Cancer Genome Atlas Study) of AMLs have shown that these leukemias have fewer mutations than the majority of solid tumors, with an average of 13 mutations per case: 23 genes were found to be recurrently mutated, but 237 additional mutated genes were observed in rare cases [31]. The studies of molecular characterization have led to a new molecular subgroup of AMLs [32,33]. In this context, particularly important was the molecular classification proposed by Papaemmanuil et al. [32], proposing the following classification of adult AMLs: (a) AMLs characterized by unique molecular events, such as inv (16) with the fusion gene *CBFB-MYH11* (core-binding factor subunit beta/myosin 11), t(15;17) with the fusion gene *PML-RARA* (promyelocytic leukemia/reitinoic acid receptor alpha), t(8;21) with the fusion gene *RUNX1-RUNXT1*, inv (3) with the fusion gene *DEK-NUP214 (DEK-nucleoporin 214)* (the *CBFB-MYH11* and *RUNX1-RUNXT1* AMLs together form the CBF leukemia group); (b) the AML chromatin-spliceosome group (18% of total) characterized by mutations in genes regulating RNA splicing (*SRFS2, SF3B1, U2AF1* (U2 auxillary factor 1) and *ZRSR2 (zinc finger CCCH-type, RNA binding motif and serine/arginine rich 2)*), chromatin (*ASXL1, STAG2, BCOR (BCL6 corepressor), MLL^PTD^, EZH2* and *FHF6*) or transcription (*RUNX1*); (c) a third AML group is characterized by mutations in *TP53*, complex karyotype alterations, cytogenetically detectable copy-number alterations, or a combination, forms 13% of all adult AMLs; (d) a fourth group is represented by Nucleophosmin-1 (*NPM1*)-mutated AMLs, representing 27% of all AMLs, with the majority of cases displaying mutations in DNA methylation and hydroxymethylation genes (*DNMT3A, IDH1, IDH2^R140^* and *TET2*); (e) a fifth group is represented by *CEBPA* double-mutated AMLs, representing 4% of all AMLs, displaying frequent *NRAS* and *GATA2* mutations; (f) a sixth group is represented by AMLs with *IDH2^172^* mutations, representing 1% of all AMLs: at variance with IDH2^140^-mutated AMLs which show strong co-mutation with *NPM1*, *IDH2^172^*-mutated AMLs are mutually exclusive with *NPM1* mutations [31]. Furthermore, about 11% of AMLs displayed at least one driver mutation, but no class-defining genetic alteration, while 4% of AMLs do not show any driver mutation [32].

The genes most frequently mutated in adult AML patients are: (a) the TK membrane receptor FLT3, mutated in about 30–40% of cases, of whom about 30% *FLT3-Internal Tandem Duplication* (*ITD*) mutation and about 10% *FLT3-Tyrosine Kinase Domain* (*TKD*) mutation (*FLT3-ITD* is frequently associated with *NPM1* mutations, t(15;17) and t(6;9) and its prognostic significance is dependent on the presence or not of concomitant *NPM1* mutation and ITD allelic ratio); (b) the *nucleophosmin 1* gene, mutated in about 30–35% of cases (frequently associated with mutations in *DNMT3A* (50%), *FLT3-ITD* (40%) cohesin genes (20%), NRAS (20%); the prognosis is largely related to its association with FLT3-ITD mutations); (c) the DNA methyltransferase *DNMT3A*, mutated in about 20–30% of cases (associated with AMLs with *NPM1, FLT3-ITD* and *IDH1* and *IDH2* mutations, associated with clonal hematopoiesis in healthy elderly persons; early event in leukemogenesis; unclear prognostic significance, context-dependent); (d) the signal transducer gene *NRAS* is mutated in about 15–22% of cases (associated with NPM1 and biallelic *CEBPA* mutation and with *inv(16)t(16;16)*; in association with NPM1 mutation may confer a favorable prognosis) (e) the transcription factor *RUNX1* is mutated in about 15% of cases (this mutation makes part of AMLs with mutated chromatin, RNA splicing or both and is usually associated with older age and poor outcome; frequent in secondary AMLs evolving from MDS) (f) the methylcytosine dioxygenase 2 *TET2*, a DNA demethylase is mutated in 15–20% of cases (associated with karyotype-normal AMLs, with *NPM1* mutations, mutually exclusive with *IDH1/IDH2* mutations, associated with clonal hematopoiesis in healthy elderly persons; an early event in leukemogenesis); (g) the isocitrate dehydrogenase 2, *IDH2*, gene is mutated in 12–16% of AMLs (*IDH2^A140^*, frequently associated with NPM1 mutation, with clonal hematopoiesis in elderly persons; early event in leukemogenesis; increased frequency with age, usually associated with a more favorable prognosis; *IDH2^R172^*, mutually exclusive with *NPM1* mutation, associated with clonal hematopoiesis in elderly persons, early event in leukemogenesis, unclear role in AML prognosis) (h) Wilms tumor protein 1, *WT1*, a tumor suppressor, is mutated in 10–13% of AMLs (associated with *FLT3-ITD* and *CEBPA* mutations, with younger age and with chemotherapy resistance and poor outcome); (i) ASXL transcriptional regulator 1, *ASXL1*, is mutated in 10–12% of AMLs (associated with chromatin-spliceosome AML group, RUNX1 mutation, +8 and -7 karyotype, associated with clonal hematopoiesis in elderly persons; early event in leukemogenesis; increased frequency with age, associated with poor prognosis); (l) the Protein Tyrosine Phosphatase, non-Receptor type 11, *PTPN11*, involved in RAS/MAPK signaling pathway, is mutated in 8–10% of AMLs (frequently associated with *TET2* (52%), *NPM1* (31%) and *FLT3-ITD* (26.5%) mutation, with a normal cytogenetics (49%); associated with variable outcome, mostly poor outcome (50%), linked to variant allelic frequency); (m) the Serine and Arginine Splicing Factor 2, *SRSF2*, a pre-mRNA splicing factor, is mutated in 8–10% of AMLs (this mutation makes part of the AML class with mutated chromatin, RNA-splicing, or both, associated with older age and poor outcome); (n) the Isocitrate Dehydrogenase IDH1 is mutated in 6–10% AMLs (frequently associated with *NPM1* mutations, associated with clonal hematopoiesis in healthy elderly persons; early event in leukemogenesis and with a prognostic impact context-dependent); (o) the tyrosine kinase membrane receptor KIT (kinase-inducing tumors) is mutated in <5% of AMLs (frequently associated with core-binding factor AMLs (25–35%), associated with unfavorable prognosis in AMLs with t(8;21)) [33,34,35]. For a classification of AMLs in three risk groups (low, intermediate and high) and their respective molecular features, according to the European Leukemia Network [36], see Table 1.

The definition of the genomic landscape of AML cells allowed to define the spectrum of molecular abnormalities and offered the unique opportunity to explore their sensitivity to a large panel of small-molecule inhibitors. AMLs evolved from a preceding pre-leukemic condition show less sensitivity than de novo AMLs to most drugs [37]. The analysis of drug sensitivity of the various AML subtypes allowed to define gene signatures of drug responses: FLT3-mutated AMLs displayed sensitivity to FLT3 inhibitors; *TP53* or *ASXL1* mutations were associated with a broad pattern of drug resistance; mutations in *NRAS* or *KRAS* correlated with resistance to most drugs; *IDH2* mutations conferred sensitivity to a broad spectrum of drugs, whereas mutations in IDH1 conferred resistance to most drugs; mutations in *RUNX1* correlated with sensitivity to PI3K and mTOR inhibitors; mutations in splicesome components correlated with sensitivity to several drugs; *NPM1/FLT3-ITD*-mutant AMLs displayed sensitivity to ibrutinib (an inhibitor of BTK and TEC family kinases) and entospletinib (a spleen-associated tyrosine kinase inhibitor) [37].

Recent studies have shown the existence of a strong connection between leukemic driver mutations, network of transcription factors and signaling components in primary AML cells [38]. These mechanisms contribute to establish and maintain within leukemic cells specific gene regulatory and signaling networks that are distinct from those observed in normal hematopoietic cells; aberrantly expressed transcription factors are the key mediators of this leukemic program of gene expression, essential for the maintenance of the leukemic growth [38]. Targeting of these deregulated transcription factors may represent a new therapeutic strategy for targeted treatment.

In old AML patients (i.e., ≥75 years) the frequency of *TET2* (42%), *SRSF2* (25%) and *ASXL1* (21%) was higher than in adult AML patients, while *DNMT3A* and *NPM1* mutations were observed at comparable frequencies in these two groups of AMLs; finally, *FLT3-ITD* mutations are less frequent in old than in adult AML patients [39].

The pattern of clinical ontogeny defines three different categories of AMLs: (a) de novo AMLs, occurring in the absence of any previous known stem cell disorder and of any previous exposure to known leukemogenetic events; (b) secondary AMLs (s-AMLs), resulting from an acute transformation event occurring in antecedent myelodysplastic or myeloproliferative disorder; (c) therapy-related AMLs (t-AMLs), occurring as a late transforming event in patients previously exposed to leukemogenetic therapies [40]. The study of s-AMLs allowed to define a mutational pattern typically associated with these leukemias and involving the presence of a mutation in *SRSF2, SF3B1, U2AF1, ZRSR2, ASXL1, EZH2, BCOR* or *STAG2*, genes commonly mutated in myelodysplastic syndromes [40]; three genetic alterations are under-represented in s-AML compared to de novo-AMLs and are represented by *NPM1* mutations, *MLL* and *CBF* rearrangements [40]. 

AML occurs also at the pediatric age and represents about 20% of all pediatric acute leukemias. The genetic abnormalities observed in pediatric AMLs have been characterized in detail only recently. Two key studies have shown that pediatric AMLs display several recurrent structural alterations, with some age-related specific features: (a) only eight mutations (*FLT3, NRAS, KRAS, NPM1, WT1, CEBPA, WT1, PTPN11* (*protein tyrosine phosphatase, non-receptor type 11*)) and five structural aberrations (fusions involving *RUNX1, CBFB* and *KMT2A* (*histone-lysine methyltransferase 2A*); trisomy 8 and loss of chromosome Y) are observed in more than 5% of patients; (b) *DNMT3A* mutations are absent in pediatric AMLs; (c) *NPM1, IDH1, IDH2, RUNX1, TP53* and *TET2* mutations are clearly less frequent in pediatric than adult AMLs; (d) *NRAS, KRAS, KIT, WT1, CBL* (*casitas B-lineage lymphoma*)*, PTNP11* and *CEBPA* mutations are more frequent in pediatric AMLs than adult AMLs; (e) new *GATA2, FLT3* and *CBL* mutations were observed in pediatric AMLs; (f) new gene fusions and focal deletions of *MBNL1* (*muscleblind-like splicing regiulator 1*)*, ZEB2* (*zinc finger E-box binding homeobox 2*) and *ELF1* (*E74-like ETS transcription factor 1*) were much more frequent in pediatric than adult AMLs [41,42]. Other remarkable differences between pediatric and adult AMLs are represented by: (a) in children, the large majority of patients present with de novo AMLs, while in adults, a significant proportion of AMLs arises from an underlying MDS or myeloproliferative neoplasm; (b) in pediatric AML, only 20% of cases have a normal karyotype and the number of somatic mutations is lower than in adult [41,42]. Studies in animal models clearly showed that pediatric AML is biologically different to adult AML, and that the age of the cell of origin clearly influences leukemia latency, lineage, molecular profile, and the leukemic microenvironmental niche [43].

In both adult and pediatric AML patients the survival is poor, as most patients relapse despite achieving complete initial remission. The failure of therapy was initially ascribed to the generation of mutations that produce drug resistance, but it is now related to the pre-existence in AML patients of drug-resistant leukemic clones, showing two major clonal evolution patterns occurring during AML relapses, mediated either by acquisition of new mutations in the founding clone or by a subclone of the founding clone, surviving to the induction therapy [44]. According to these findings, the resistant leukemic cells are generated during the evolutionary process of AML development and are selected by anti-leukemic therapy. A recent study based on combined genetic and functional analysis of purified leukemic progenitor cell populations and xenografts from paired diagnosis and relapse samples, showed the purification and characterization of leukemic cells present at diagnosis and resistant to therapy; two patterns of relapse were observed, either mediated by rare leukemic stem cells with a hematopoietic stem/progenitor cell phenotype or by committed leukemic blasts that, in spite their phenotype, retained stemness signatures [45]. In line with these findings, leukemia stem cell gene expression signature is predictive of therapy failure in adult [46] and pediatric AMLs [47]. 

In the majority of AML patients, the CD34^+^/CD38^−^ fraction contains cells with the potential of leukemia initiating cells. The assessment of this cell population in primary AML samples represents a useful parameter to predict outcome, in that AMLs with high 34^+^/CD38^−^ frequency have a poor prognosis [48]. This property is strictly related to the presence of 34^+^/CD38^−^ cells in that AMLs with high CD34, but low 34^+^/CD38^−^ content have a significantly better prognosis that AMLs with 34^+^/CD38^−^ high [48]. Importantly, the prognosis of AMLs with 34^+^/CD38^−^ high and MRD positivity was particularly negative [48].

The key role of the leukemic stem cell compartment in leukemia genesis is also supported by the observation that a minority of high-risk AMLs display the features of both AMLs and ALLs and are known as mixed phenotype acute leukemias (MPALs). These leukemias are differentiated into three subtypes: T/M (T-lymphoid/myeloid); B/M (B-lymphoid/myeloid); KMT2-rearranged; these three subtypes are genetically distinct: T/M PALs display frequent WT1, FLT3 and ETV6 alterations; B/M PASLs show frequent *ZNF384* (*zinc finger protein 384*) rearrangements and *NRAS* mutations; KMT2Ar MPALs display *KMT2* rearrangement [49].The intratumoral immunophenotypic heterogeneity of MPALs is independent of somatic genetic variation and founding genetic lesions arise in primitive hematopoietic progenitors that maintain their myeloid and lymphoid differentiation potential after leukemic transformation, thus propagating similar mutation profiles in cells of different lineage [49].

The existence of a clonal mutational heterogeneity in many tumors, including AMLs, is a mechanism strongly limiting the efficacy of current therapies. The consistent variability of allelic mutational burdens strongly suggested sob-clonal architecture in AML diagnostic samples [50]. In fact, strategies for estimating and tracking tumor clonal diversity performed by next generation sequencing on the tumor bulk samples are assumed to occur simultaneously in the same clone of cells, while mutations that occur at lower frequency are assumed to have occurred later in the tumor evolution and to represent a sub-clone in which all mutations of higher frequency are present together with the lower frequency mutation; all the mutations occur heterozygously in a single tumor cell; the mutations of a given gene are unique events in a tumor cell. However, to assess clonal diversity within AMLs without these assumptions, implying some potential biases, several recent studies have directly genotyped single AML cells. Paguirigan and coworkers have used single cell genetics (multiplexed quantitative PCR) to investigate the patterns of segregation of two concurrent mutations in AML, *FLT3-ITD* and *NPM1* mutations, showing that mutations of FLT3 and NPM1 occur in both homozygous and heterozygous states, distributed among at least 9 different clonal populations in all AML samples analyzed, thus indicating more subclonal heterogeneity that could be inferred from analysis of the bulk population [51]. A very pronounced clonal diversity observed in the majority of AML patients with *FLT3-ITD* mutations plays also a great role in the mechanisms of resistance of these leukemic cells to FLT3 inhibitors [52]. Klco and coworkers have fractionated immunophenotypically distinct cell populations from AML patients and have sequenced the amplified DNA from single cells for ten known mutations, showing a consistent clonal complexity, compatible with a branching sub-clonal architecture [53]. Quek and coworkers have analyzed individual leukemic cells for mutations in immunophenotypically-defined cell subsets and deduced a sequence of mutational events at clonal level during leukemia development [54]. The technology of single-cell DNA sequencing is complex and not suitable for routine applications. Recently, it was reported a novel microfluidic approach that barcodes amplified genomic DNA from thousands of individual cancer cells confined to droplets; the barcodes are then used to reassemble the genetic profiles of cells from next-generation sequencing data [55]. This approach should make feasible the routine analysis of AML heterogeneity [55].

The definition of these molecular abnormalities has thus led to identify AML subsets and to better define their biology and the potential impact of their mutational spectrum. Thus, the molecular studies have defined the *NPM1*-mutated AML subset. This mutant-*NPM1*, whose normal location is nuclear, accumulates in the cytoplasm, dislocating the master regulator of myeloid differentiation PU.1 from the nucleus to cytoplasm; the other two transcription factors regulating myeloid differentiation, CEBPA and RUNX1, remained nuclear, but, in the absence of PU.1, shifted their biologic activity from co-activators to co-repressors, thus repressing the expression of target granulo-monocytic genes [56]. *NPM1* mutations are almost invariably small insertions in the terminal exon of NPM1 gene that result in the loss of nucleolar localization, with consequent cytoplasmic localization of the mutant NPM1 protein and gain of a novel nuclear export signal (NPM1c). Specific loss of NPM1c from the cytoplasm leads to downregulation of HOX genes and differentiation of *NPM1* mutant AMLs [57]. Blocking NPM1c nuclear export with a chemical inhibitor reduces cytoplasmic NPM1c, promotes AML differentiation, and prolongs the survival of a mouse model of NPM1c^+^ AMLs [57]. *NPM1* mutation has a favorable effect on the outcome for AML, as supported by various clinical studies carried out on adult AML patients [47]. Co-mutations affect the outcome of *NPM1*-mutated AMLs: the presence of *DNMT3A* mutations worsen the outcome of *NPM1*-mutant (as well as *NPM1*-WT) AMLs; the presence of *FLT3-ITD* mutations, worsen the outcome of *NPM1*-mutant AMLs [58]. The co-occurrence of *NPM1* mutation and *FLT3-TKD* mutation defines an AML subgroup exhibiting a highly favorable prognostic profile compared to *NPM1*-mutated AMLs without *FLT3-TKD* mutation [59]. The relative abundance of the mutated allele has an important impact on the prognosis of *NPM1*-mutant AML: in fact, patients with high *NPM1*-mutant allele burden have a shortened overall survival, as well as event-free survival, compared with other *NPM1*-mutant AMLs [60].

The analysis of clonal dynamics in *NPM1*-mutated AMLs allowed to show the clonal heterogeneity and to propose a model of mutational development of these leukemias. NPM1 mutation is a subclonal event and therefore is a secondary mutation event rather than an initial, truncal mutational event, as supported by numerous observations: (a) the allele burden for mutated *NPM1* was consistently less than that of other driver mutations, such as *DNMT3A* and *TET2* [61]; (b) in the CD34^+^/CD33^−^ stem/progenitor fraction of these leukemias, the existence of clones ancestral to those with *NPM1* mutation was observed [61]; (c) *DNMT3A*, but not *NPM1* mutations were present in purified hematopoietic stem cells and putative pre-leukemic clones [62,63]; (d) the preservation of diagnostic *DNMT3A*, but not *NPM1* mutations in remission [64]. According to these observations, effective therapeutic targeting of either *NPM1* or *FLT3* mutations is expected to greatly reduce leukemic blast burden, but with only transient benefit. The order of mutations during development of *NPM1*-mutated AMLs suggest that *NPM1* mutations and seemingly *FLT3* mutations may act as potent leukemic drivers only when arising in myeloid stem/progenitor cells displaying enhanced proliferation/self-renewal induced by mutation of epigenetic regulators, such as *DNMT3A* or *TET2* [61]. 

In AML patients 60 years old or older, the presence of *NPM1* mutation, co-occurring with mutations in chromatin remodeling, cohesion complex, methylation-related, spliceosome, and/or *RAS* pathway genes, *FLT3-TKD*, without *FLT3-ITD*, identifies a subset of patients with favorable response to chemotherapy; patients with *NPM1* mutations and *SF1* mutations displayed a favorable overall survival [65]. In contrast, the presence of *BCOR, FLT3-ITD, TP53, U2AF1, WT1* mutations, of a complex karyotype or t (9;11) was associated with a poor survival [65]. These studies emphasize that the prognostic effect of a given mutation depends on what other mutations are present. 

The presence of a molecular marker of the malignant leukemic clone, such as mutant NPM1, offers the opportunity to monitor the response to therapy and to evaluate a possible disease evolution in relapsing patients. The majority of patients relapsed with a *NPM1^mut^* AML (about 86% of cases), while a minority (about 14%) relapsed with a *NPM1^WT^* AML [66]. Importantly, *NPM1^WT^* relapse occurred significantly later than *NPM1^mut^* relapse (43 months versus 14 months) [66]. At diagnosis, *FLT3-ITD* mutations were more frequent among patients with NPM1^mut^ relapse, whereas *DNMT3A* mutations were more frequent among NPM1^WT^ relapse [66]. These observations suggest that AML relapsing with *NPM1^WT^* are a distinct disease and in these patients, initial leukemia and relapse arise from a premalignant clonal hematopoiesis [66]. Although about 10% of patients bearing a *NPM1^mut^* AML at diagnosis relapse with a *NPM1^WT^* AML, the detection of mutant *NPM1* using sensitive molecular techniques is a sensitive assay to detect the persistence of MRD after induction and consolidation therapy and to predict the risk of recurrence [67]. 

A recent study showed that mutated NPM1 is a neoantigen and this property can be exploited to rise a immune response against AML cells bearing mutated *NPM1* [68]. Particularly, it was shown that mutated NPM1-derived peptides are presented on the surface of AML and one of these peptides (CLAVEESL) is a neontigen that can be targeted on AML by mutant NPM1 T cell receptor gene transfer [68].

The ensemble of these studies allowed to offer for many AML subsets a reasonable reconstruction of the natural history underlying the development of these AMLs at the level of the main molecular events and of the clonal dynamics. The resulting fusion genes, *RUNX1-RUNX1T1* and *CBFB-MYH11* involve members of the CBF (core binding facots) family of transcription factors, RUNX1 and its binding partner CBFB. In this context, paradigmatic were the studies concerning the CBF AMLs. As above reported, AMLs with chromosomal rearrangements inv(6)(p13q22) or t(16;16)(p13q22), referred as inv(16) and t(8;21)(q22;q22) form the group of CBF AMLs. These AMLs are associated with a good prognosis, with >85% of complete responses to induction therapy; however, about 50% of these patients relapse, with an overall survival at 5 years of about 51%. Patients with core-binding factor AMLs are among ten-year disease-free AML group not treated with allogeneic transplantation after >10 year follow up [69]. The fusion proteins generated in CBF AMLs interfere with normal hematopoiesis by deregulating hundreds of genes through competition with RUNX1 for the DNA binding to RUNX1-binding sites. The RUNX1-RUNX1T1 fusion protein was not sufficient alone to induce the transformation of human hematopoietic progenitors, but requires additional genetic events. In line with these findings, sequencing studies have shown that in *RUNX1-RUNX1T1* AMLs several recurrent genetic alterations co-occurred, such as *RAS* mutations, RTK mutations (*FLT3, CBL, JAK2, KIT, PTPN11*), chromatin modifier/epigenetic regulators (*ASXLK1, ASXL2, TET2, BCOR*) MYC signaling and components of the MYC signaling pathway [70,71]. FLT3 and KIT mutations at high allelic ratio confer a poor prognosis. In *CBFB-MYH11* AMLs, *NRAS, KIT, NF1, WT1* and *FLT3* mutations were frequently observed [71]. *CCND1* (*cyclin D1*) and *CCND2* (*cyclin D2*) genes are frequently (about 15% of cases) mutated in t(8;21), but not in inv(16) AMLs (about 0.8%) [72]. CCND mutations in t(8;21) AMLs determine increased phosphorylation of the retinoblastoma protein, causing cell cycle changes and increased proliferation of leukemic blasts [72]. Interestingly, *CCND2* was identified as a crucial transcriptional target of *RUNX1/ETO*: the upmodulation of CCND2 is essential for the *RUNX1/ETO*-driven leukemic expansion in vivo [73]. Importantly, t(8;21) AML cells are markedly sensitive to a CDK4/6 inhibitor [73].

The analysis of the major genetic events at various stages of leukemia development allowed to propose a model of clonal evolution of CBF AMLs involving an initiating event of inv(16) or t(8;21); at a later stage of leukemia development, additional genetic abnormalities occurs, mainly represented by mutations in driver genes (*FLT3, KIT, DNMT3A, SM1CA, SMC3, EZH2, NRAS, WT1, DHX15*) or chromosomal abnormalities (trisomy 8, del 6p). Clonal evolution during relapse is variable and may imply two different mechanisms: (a) a subclone of the diagnosis leukemia survived therapy and reemerged after the accumulation of additional mutations; (b) a preleukemia clone survive therapy, acquired additional mutations during remission and generates the relapsing leukemia [70]. A very recent study evaluated on a large set of t(8;21) AML patients the types of clonal evolution occurring during disease relapse: 43% of mutations were found in both diagnosis and relapse samples; 26% were acquired/selected during disease progression and 31% present at diagnosis were lost at relapse [74]. The t(8;21) translocation was found in all patients both at diagnosis and at relapse. The analysis of the mutational evolution suggested that some clones were eradicated during induction chemotherapy, while other clones escaped or were selected and expanded at relapse by acquiring new mutations; according to these criteria, these patients can be subdivided into two different groups with either >40% of stable mutations (group A) or >60% of gained/lost variants (group B) [74]. The dynamics of mutational evolution at clonal level followed gene-related patterns: (a) mutations in epigenetic regulators and genes involved in cell cycle control are stable or lost, but never acquired at relapse, supporting their role in leukemia initiation; (b) in contrast, mutations in RTK, RAS signaling pathway, transcription factors and members of the cohesion complex are stable, lost or acquired at relapse, supporting their role at various and later stages of leukemia development [74].

The final translational endpoints of studies of molecular characterization of AML subsets consist in the identification of gene drivers and/or class-defining mutations suitable to be targeted with specific drugs, with assumption that the inhibition of these genes will affect the proliferation/survival of leukemic cells. The tremendous progresses achieved by all-trans retinoic acid (ATRA) and arsenic trioxide in acute promyelocytic leukemia (APL) and tyrosine kinase inhibitors in hematological neoplasia with the Philadelphia chromosome has strongly encouraged the search for other effective targeted therapies in AML subsets [20]. So far, the Food and Drug Administration (FDA) has approved three new targeted therapies in AML patients: (a) the antibody-drug conjugate gentuzumab ogozamycin (GO) in relapsing CD33^+^ AMLs; (b) the tyrosine kinase inhibitor midostaurin in *FLT3-ITD*-mutated AMLs; (c) the IDH2 inhibitors Enasidernib in *IDH2*-mutated AMLs [75].

## 3. Emerging Therapies for Patients with AML Targeting the BCL-2 Pathway

In addition to these drugs, emerging therapies for AML are represented by a number of new drugs that target deregulated apoptosis. Among these drugs, the most developed are those targeting the anti-apoptotic protein BCL-2. The BCL-2 family proteins play a key role in regulating mitochondria-mediated apoptosis; these proteins share one or more of the four BCL-2 homology domains (BH1-4) and comprise proteins with pro-apoptotic effector multi-domain structure (BAX and BAK), the pro-apoptotic BH3-only effectors with activator function (BID, BIM and PUMA) and with sensitizing function (BAD, BMF and NOXA) and the anti-apoptotic effectors (BCL-2, BCL-XL, BCL-2-like 2, MCL-1 and BFL-1/A1) [13]. The anti-apoptotic proteins prevent the activation of BAX and BAK and maintain the integrity of mitochondrial outer membrane [76].

Many of the drugs currently used in cancer therapy mediate, at least in part, their cytotoxic effects through activation of the BCL-2-controlled apoptotic pathway; however key initiators of this pathway, such as TP53, are very frequently altered in cancer, creating a mechanism of drug resistance. To bypass this bottleneck, a novel class of pro-apoptotic drugs, called BH3-mimetics, was developed [77,78]. Among the various BCL-2 members, BCL-2 is the most advanced therapeutic target. Preclinical studies have shown that BCL-2 is frequently overexpressed in AML blasts, compared to their normal hematopoietic counterpart and BCL-2 levels are particularly high in progenitor/stem leukemic cells [79,80]. Anti-BCL-2 agents showed anti-leukemic activity alone or in combination with other anti-leukemic agents [77,78]. Importantly, mitochondrial priming measured by BH3 profiling was shown to be a key determinant of initial response to induction chemotherapy treatment and relapse after remission; interestingly, BH3 profiling identified BCL-2 inhibition as a targeted strategy to have a useful therapeutic index [81]. The presence of functionally apoptosis-resistant subpopulations of AML cells determine chemoresistance in AML: the chemoresistant cells were enriched for anti-apoptotic proteins [82]. The BCL-2 inhibitor ABT-735 and its clinical analogue ABT-263 (navitoclax) binds BCL-2, BCL-XL and BCL-w with high affinity and induce regression of solid tumors in preclinical models [83]. Initial clinical studies in patients with relapsed or refractory lymphoid malignancies showed a significant activity as single agent; however, the efficacy and the clinical use of this agent were consistently limited by frequent induction of cytopenia due to BCL-XL inhibition. Given these limitations, a more specific BCL-2 inhibitor was developed, ABT-199 (Venetoclax), which showed a higher affinity for BCL-2 than ABT-263 and a much lower affinity for BCL-XL than ABT-263 [84]. In a phase 2 multicenter trial of venetoclax administered as monotherapy in patients with relapsed/refractory AML an overall response rate of 19% was observed: interestingly, three of the five patients exhibiting a complete response have IDH-mutant AMLs [85]; 33% of AML patients with IDH1/2 mutations achieved a response to venetoclax monotherapy treatment (Table 2) [85]. In vitro studies on primary AML blasts confirmed that IDH1-IDH2 mutant AML cells are more sensitive than IDH1-IDH2-WT leukemic cells to venetoclax [86]. Interestingly, BH3 profiling was carried on pretreatment bone marrow biopsies of a part of the patients participating to this study, showing that baseline BCL-2 dependence of blast mitochondria correlated with clinical response, as well as lack of myeloblast dependence on the anti-apoptotic proteins BCL-XL or MCL-1 (factors of resistance to a selective BCL-2 inhibitor, such as venetoclax) [85]. Analysis of genomic predictors of response provided evidence that IDH1-IDH2 mutations were associated with a higher probability of response, while FLT3-ITD or PTPN11 mutations were associated with lack of response [87].

In vitro screening on a large panel of primary AML blasts showed that 15% of samples displayed high sensitivity, 32% were resistant and 53% displayed an intermediate sensitivity to venetoclax [88]. BCL-2 inhibitor sensitivity was associated with genetic aberrations in chromatin modifiers WT1 and IDH1-IDH2 and with overexpression of HOXA and HOXB gene transcripts [88]. 

The results obtained using venetoclax in AML patients in monotherapy showed limited and short-lived responses, a finding that triggered the evaluation of combination therapy studies in association with standard anti-leukemic drugs. Preclinical studies have shown that concomitant BCL-2 inhibitory agents and inhibition of methyltransferase potentiate the anti-leukemic effects on AML blasts [89,90]. Venetoclax synergizes with cytarabine and idarubicin to increase anti-leukemic efficacy in a TP53-dependent manner [36]. Importantly, higher-dose idarubicin was able to markedly reduce MCL-1 and to induce cell death independently of TP53 [91].

These studies have prompted the development of clinical trials based on the combined administration of venetoclax with the cytidine analogues 5-azacytidine or decitabine (these cytidine analogues upon their incorporation into DNA inhibit DNA methyltransferases and thus indirectly induce S-phase arrest and DNA hypomethylation) (Table 2). In a phase 1b clinical study based on venetoclax plus hypomethylating agents (decitabine or 5-azacytidine) in 145 elderly newly diagnosed AML patients ≥65 years of age, not eligible for standard induction therapy, an acceptable toxicity profile was observed, in association with a high rate of clinical responses (67% of complete responses), a median overall survival of 17.5 months and at 2 years an estimate of overall survival corresponding to 46% [92]. These data compare very favorably with historical data on the response rate of older AML patients to 5-azacytidine monotherapy [93]. Interestingly, the retrospective analysis of 33 consecutive adult relapsed AML patients treated with the combination of venetoclax plus 5-azacytidine or decitabine showed a high rate of objective responses which compares favorably with historical controls [94]. Interestingly, complete responses were observed also among patients who are expected to respond poorly to conventional combination chemotherapy (i.e., patients with FLT3 and TP53 mutations) [94]. The response to venetoclax plus 5-azacytidine or decitabine treatment, would offer to some patients the opportunity to undergo a hematopoietic stem cell transplantation procedure, the only potentially curative therapy for relapsed AML. The treatment based on venetoclax plus hypomethylating agents could became the standard of care for elderly AML patients.

Other studies are evaluating the combination of venetoclax with low-dose cytarabine in elderly AML patients (≥65 years aged) who are ineligible for intensive chemotherapy (Table 2). Preliminary results of the completed phase 1–2 combination trial who enrolled 61 patients showed an acceptable safety profile [95] and a high response rate with CR/CRi rate of 62% and median overall survival of 11.4 months [96]. Response rates were significantly higher in the group of patients with an intermediate cytogenetic risk (median overall survival 15.7 months), compared to those with high-risk cytogenetics (median overall survival 5.7 months); furthermore, all NPM1-mutated or CEBPA biallelic AMLs responded to treatment [96].

Another recent study explored the combination between venetoclax and the MEK inhibitor cobimetinib or MDM2 inhibitor idasanutlin in patients with relapsed or refractory AML (Table 2) [97]. Encouraging results were observed with both drug combinations, with 18% (venetoclax plus cobimetinib) or 20% (venetoclax plus idasanutlin) responding patients [97]. Interestingly, the combination of venetoclax with a MDM2 inhibitor was supported by a preclinical study showing that p53 activation through pharmacological inhibition of MDM2, inducing a reduction of Ras/Raf/MEK/ERK, with consequent activation of GSK3β, MCL-1 phosphorylation and degradation, and, through this mechanism, overcoming resistance to Venetoclax [98].

Other preclinical studies have supported the synergistic interaction between venetoclax and PI3K inhibitors in inducing BAX-dependent mitochondrial apoptosis of AML cells [44]; interestingly, this drug combination was also very active in killing CD34^+^/CD38^−^/CD123^+^ leukemic stem cells [44]. Importantly, this study showed also that the venetoclax-induced apoptosis of AML cells involves and requires BAX activation [99]. The fundamental role of BAX activation in inducing apoptosis of AML cells is supported also by another recent study the identification of a molecule, BAX trigger site activator 1 (BTSA1) able to bind and activate BAX; this compound induced in vitro apoptosis of AML cell lines and prolonged AML xenograft recipients’ survival [100]. Finally, a strong synergistic effect for the combined treatment of BTSA1 and venetoclax was observed [100]. In this context, another recent study provided evidence that BAX cellular localization is a major determinant of apoptotic response of primary AML cells: cytosolic BAX localization is associated with decreased predisposition to apoptosis, while high BAX mitochondrial levels correlate with improved AML patient survival [101]. The cellular BAX localization in primary AML blasts is highly variable between different primary AML samples and may be variable at the level of subpopulations of a single sample.

Among the various drug combinations under evaluation, venetoclax is being evaluated in phase 1 studies in association with a TRAIL receptor agonist fusion protein, ABBV-621, active against hematologic tumors, including AMLs.

In conclusion, evidences in favor of clinical utility for selective inhibition of BCL-2 are clear, but future studies are required to improve the efficacy of these therapies and to better define the AML subtypes better responding to BCL-2 targeting. Furthermore, future studies will be required to better define combination therapies with venetoclax, able to deepen responses to this BCL-2 inhibitor and to bypass the mechanisms of resistance. This conclusion is strongly supported by the observation of the consistent effectiveness of several combinations of targeted agents including venetoclax and a kinase inhibitor in AMLs [102]. The high sensitivity of AML cells to these combinations in the ex in vivo assay supports the notion that coupling a kinase-derived antiproliferative signal inhibitor with a drug blocking an antiapoptotic agent greatly improves efficacy [102]. These observations strongly support the evaluation of Venetoclax in combination with other agents in AML.

Other studies indicate that another protein of the BCL-2 family, MCL-1, could represent a suitable target for developing an anti-AML therapy. MCL-1 exerts its anti-apoptotic activity by blocking pro-apoptotic proteins, such as BAK and BAX and, through this mechanism, acts as an inhibitor of apoptosis. Various lines of evidence indicate a deregulation of MCL-1 in AMLs: (a) MCL-1 was found to be expressed at high levels at both mRNA and protein levels in virtually all primary AMLs; interestingly, MCL-1 was more uniformly expressed at high levels in primary AMLs than BCL-2 or BCL-XL [103]; (b) pre-treatment MCL-1 levels in primary AML blasts directly correlated with BCL-2 levels and increased at disease recurrence [104]; (c) patients with particularly elevated MCL-1 levels have a poor prognosis and/or response to standard chemotherapy [105]; (d) MCL-1 is highly expressed in leukemic populations (CD34^+^/CD38^−^) enriched in leukemic stem cells and its expression is required for the survival of these cells [106].

Importantly, other studies have shown that MCL-1 is critical for the survival of mouse and human AMLs resulting from various types of oncogenic lesions [105]. These observations have strongly supported the development of pharmacologic inhibitors targeting MCL-1 or regulators of its expression for treatment of AMLs. MCL-1 inhibitors were shown to be able to induce apoptosis of AML cells. Thus, maritoclax, one of the first MCL-1 inhibitors developed, induces in AML cells a reduction of MCL-1 levels by targeting the protein for proteosomal degradation and activates a caspase-dependent apoptosis process, with consequent cell death [107]. These effects were shown both in leukemic cell lines and primary AML blasts [107]. However, subsequent studies have shown that this compound lacks of molecular specificity and induces cell death through multiple mechanisms.

Subsequent studies have reported the synthesis and characterization of several small-molecule MCL-1 inhibitors, including UMI-77, A-1210477, S63845 [108]. Among the various MCL-1 inhibitors particularly interesting was the S63845 inhibitor. S63845 was developed as a potent, selective MCL-1 inhibitor exhibiting low nanomolar cytotoxic activity in multiple cancer models and particularly in hematologic cancers, including AML [109]. Particularly, this compound was active in inducing the death of both AML leukemic cell lines and primary leukemic blasts, with a preferential cytotoxic activity against leukemic cells, sparing normal hematopoietic progenitors/precursors [109]. The properties of this compound were particularly suitable for clinical pharmacology development and one its derivative, S64315 was introduced in phase I clinical trials in patients with AMLs and myelodysplastic syndromes [110]. UMI-77 is a small molecule inhibitor of MCL-1 which binds to the BH3-binding groove of MCL-1 with Ki of 490 nmol/L, showing selectivity over other BCL-2 family members [111]. A-1210477 is a small-molecule inhibitor of MCL-1 binding with high-affinity this protein (Ki 545 nmol/L): the inhibitor blocks the binding of MCL-1 with BIM [112].

AMG 176 is a potent and selective MCL-1 inhibitor currently in phase I clinical evaluation; this compound binds with high-affinity and selectivity to the BH3-groove of MCL-1, disrupting the interaction between MCL-1 and BAK [113]. This inhibitor was active in inducing the cell killing of various tumor cell types, including AML cell lines [113]. Treatment of tumor cell lines with AMG176 elicited an increase of MCL-1 half-life [113]. Interestingly, molecular profiling of tumor cells treated with AMG176 showed an inverse correlation between BCL-XL expression and sensitivity to MCL-1 inhibition [113]. With the use of a human MCL-1 knock-in mouse, it was provided evidence that MCL-1 inhibition induced by active doses of the MCL-1 inhibitor is tolerated and collates with evident pharmacodynamic effects, such as reductions in B cells, monocytes and neutrophils [114]. A phase I (NCT 02675452) multicenter, non-randomized, open-label and dose-exploratory study is evaluating AMG176 administered IV in patients with multiple myeloma and AML (Table 3). AZD5991 is a macrocycle inhibitor of MCL-1, with sub-nanomolar affinity for MCL-1 that induces apoptosis of leukemic cell lines and primary leukemic cells at nanomolar concentration in vitro and induces in vivo tumor regression in various mouse xenograft models [115]. A phase I study (NCT 03218683) evaluated the safety and the anti-tumor activity of AZD5991 in relapsed or refractory hematologic malignancies (Table 3). 

Five recent studies highlight the great therapeutic potentialities of the association of an MCL-1 inhibitor with a BCL-2 inhibitor. Thus, Moujhalled and coworkers have shown that the combination of BH3-mimetics able to target both BCL-2 (S55746) and MCL-1 (S63845) was able to induce a pronounced cytotoxic effect on AML cells, active against a broad spectrum of poor risk genotypes, including primary chemoresistant AMLs [116]. Furthermore, the combined BCL-2 (Venetoclax) and MCL-1 (S63845) inhibitor treatment bypassed the relative resistance of FLT3-ITD-mutated AMLs to Venetoclax alone [91].

A second study showed that idarubicin at higher-doses was able to suppress MCL-1 and to synergize with venetoclax to induce the death of AML cells, including BAX-deficient leukemic cells [91]. Another advantage of this drug combination is that idarubicin at higher doses induces cell death independently of TP53 [91]. In line with this study, the MCL-1 inhibitor AMG176 and its related analogue AM-8621 synergize with Idarubicin and AraC to induce killing of primary AML blasts [114]. A third study showed that the combination of AMG176 and venetoclax is synergistic in primary AML blasts, as well as in AML animal models [114]. A fourth study showed that VU661013, a novel potent MCL-1 inhibitor destabilizing BIM/MCL-1 association, in combination with venetoclax was very active in inducing apoptosis of AML cells, including those resistant to venetoclax [117]. Finally, a fifth study showed that the A-1210477 MCL-1 inhibitor was found to synergistically induce apoptosis of primary AML cells with the BCL-2 inhibitor ABT-199. Biochemical studies showed that sequestration of BIM by MCL-1, a mechanism of ABT-199 resistance, is abrogated by combined treatment with the MCL-1 inhibitor A-1210477 [118]. 

Elevated MCL-1 and BCL-2 levels are often observed in AML blasts at relapse and contribute to the chemoresistance of these cells [104]. In MLL-AF9 leukemia models overexpressing BCL-2 and MCL-1 it was provided evidence that the combination of BCL-2 and MCL-1 inhibitors resulted in a synergistic anti-leukemic effect, when used alone or in combination with standard anti-leukemic drugs [119].

These studies strongly support a therapeutic strategy based on the combination of BCL-2 and MCL-1 inhibitors in sequence or in combination in AML clinical studies. The development of more efficacious anti-MCL-1 drugs will require both the synthesis of new more active inhibitors and the definition of biological assays able to better define the anti-tumor activity of these inhibitors and to predict their clinical efficacy. An effective strategy for inhibiting MCL-1 consists in the synthesis and use of designer peptides able to competitively engage the binding groove of this molecule, mimicking as much as possible the molecular mechanism of action of native sensitizer BH3-only proteins. The transformation of MCL-1 binding peptides into α-helical allowed the synthesis of cell-penetrating constructs that are selectively cytotoxic to MCL-1-dependent cancer cells and have biophysical properties (such as in vivo stability and high cell-penetrating capacity) suitable for drug development [120]. Concerning the problem of improving the assessment of the anti-tumor activity of MCL-1 inhibitors, it is particularly interesting a recent study by Brennan and coworkers reporting the development of a humanized MCL-1 mouse strain in which murine MCL-1 was replaced with the human homolog [121]. The valuation of these mice in the context of some models of hematological malignancies support their capacity to predict efficacy and tolerability for clinic al translation [121].

Strategies to reduce MCL-1 expression include not only direct targeting of MCL-1, but also indirect targeting by disruption of transcription/translation. Thus, the second possible strategy is targeting the synthesis of MCL-1, which could involve multiple components, including the promoter sequence, transcription/translation machinery/regulators. MCL-1 transcription is controlled by the positive transcription elongation factor b (P-TEFb) complex. P-TEFb, which is made up of CDK9 and cyclin T proteins. Thus, CDK9 inhibition was used as a strategy to directly inhibit MCL-1 expression.

Several CDK9 inhibitors are under evaluation in AML patients. Alvocidib, also known as flavopiridol, was evaluated in a randomized, phase II trial in combination with cytarabine and mitoxantrone, compared with cytarabine plus daunorubucin in 165 AML patients with core binding factor-negative leukemias [122]. Age subgroup analysis showed that patients younger than 50 years had better benefit from the former than from the latter treatment; however, no significant differences at the level of overall survival or progression-free survival were observed between the two groups [122]. The only difference observed was the higher complete remission rates observed among patients treated with alvocidib [122,123]. In a recent phase II trial, that used BH3 profiling as laboratory support to predict response to therapy, 17 patients with refractory/relapsing AML and a median MCL-1 dependency of 61% (measured by the BH3 profiling assay) received treatment with alvocidib administered before cytarabine and mitoxantrone [124]. A complete remission rate with incomplete hematologic recovery rate of 59% was observed; 75% of AML patients with refractory AML achieved CR and 50% of them were able to proceed to allogeneic stem cell transplantation [124]. A more recent update of this trial presented at the December 2018 ASH Meeting showed that among 163 AML patients screened, 29% were shown to be MCL-dependent; of these, 21 of the 25 enrolled patients were evaluable for response: 52% of these patients were resistant to standard frontline therapy; 62% of these patients showed a CR or Cri to therapy with alvocidib, cytarabine and mitoxantrone [125]. Given these findings, stage 2 of this trial was initiated, randomizing the patients to alvocidib, cytarabine and mitoxantrone versus cytarabine, mitoxantrone alone in MCL-1-dependent refractory AMLs [125]. Furthermore, a phase Ib trial evaluating alvocidib followed by 7 + 3 induction chemotherapy in newly diagnosed AML is ongoing [125].

CDK9 inhibitors, such as voruciclib, were shown to repress MCL-1 [126] and are under active investigation at both preclinical and clinical level [127]. Interestingly, dinaciclib, another CDK9 inhibitor, exerts potent apoptotic and antitumor effects in preclinical models of *MLL*-rearranged myeloid leukemias through inhibition of MCL-1 expression [128]. It is important to note that CDK9-targeted inhibitors described to date are not strictly-specific for CDK9 and suffered from adverse events, such as dose-limiting toxicity in the bone marrow and gastrointestinal tract in clinical trials [129]. The only selective CDK9 inhibitor under evaluation in clinical trials is BAY 1251152, which is currently in phase I in patients with advanced blood cancer [129].

Bone marrow (BM) stromal cells mediate a protection on leukemic blasts, reducing the sensitivity of these cells to various cytotoxic agents. The most studied mechanism by which BM stromal cells induce drug resistance is the activation of various pro-survival signals, leading to upregulation of antiapoptotic proteins, such as BCL-2 and BCL-XL [130]. A recent study showed that BM stromal cells exerts a protective effect on the leukemic stem/progenitor population (CD34^+^/CD38^−^ cells) against anti-leukemic drugs, such as cytarabine and daunorubicin, and BH_3_ mimetics [131]. Inhibition of BCL-2 and BCL-XL with ABT-737 is not sufficient to revert BM stromal cell-mediated drug resistance [131]. In contrast, MCL-1 inhibition with A1210477 or repression with the CDC 7/CDK9 inhibitor PHA-767491 sensitized CD34^+^/CD38^−^ cells to anti-leukemic cytotoxic drugs, thus offering a strategy to eradicate this leukemic cell population [131].

Bromodomain extra-terminal protein-inhibitors (BET-is) disrupt the binding of BETP Bromodomain 4 (BRD4) chromatin and reduce the synthesis of various oncoproteins stimulating the survival and proliferation of AMNL cells. Some of these inhibitors are under evaluation in phase I/II clinical trials. BET-is reduce the expression of CDK6, MYC, BCL-2 and MCL-1 and stimulate the expression of BIM, HEXIM1, CDKN1A and induce apoptosis of AML cells [132]. Given these effects of BET-is, it is not surprising that these drugs synergize with BCL-2 and MCL-1 inhibitors in inducing apoptosis of AML cells [132]. In immunodepleted mice engrafted with AML cells, the co-administration of a BET-I with Venetoclax was very efficacious in reducing tumor burden [132].

The casein kinase 1A1 gene (CK1α) is a tumor suppressor gene located in the common deleted region for del (5q) myelodysplastic syndrome (MDS). CKIα inhibition induces p53 activation, representing the proposed mechanism of action of thalidomide derivative lenalidomide in human MDS [133]. A recent study reported the identification of new specific CK1α inhibitors endowed with the multiple property of stabilizing β-catenin and TP53, of inhibiting the expression of MYC and MDM2 and inhibiting CDK7 and CDK9 [134]. As a consequence of these multiple biologic effects, resulting in TP53 activation and MDM2 downregulation, in combination with transcriptional silencing of leukemia oncogenes, such as *MYC*, *MYB* and *MCL-1*, resulted in a rapid and pronounced anti-leukemia effects, confirmed in patient-pronounced anti-leukemia effects, confirmed in patient-derived xenografts of various AML subtypes, including TET-2^−/−^ and FLT3-ITD driven leukemias [134]. These observations strongly support the preclinical and clinical development of CKIα inhibitors.

A recent study showed that targeting sphingosine kinase 1 (SPHK1) is another strategy to induce MCL-1-dependent cell death in AML cells [135]. SPHK1 generates the lipid sphingosine-1-phosphate (SIP) that promotes cell proliferation and survival through its action as a ligand for a family of 5 S1P-specific G protein-coupled receptors (S1PR1-5) or an intracellular second messenger. SPHK1 and SPHK2 are two sphingosine kinase isoenzymes with opposing functions in sphingolipid metabolism: SPHK1 exerts a pro-survival function, while SPHK2 inhibits cell growth and enhances apoptosis [136]. Interestingly, sequence analysis showed that SPHK2 contains a 9-amino acid motif similar to that present in BH3-only proteins; furthermore, SPHK2 was shown to be able to interact with BCL-XL [137]. SPHK1 is overexpressed in AMLs and its targeting induces caspase-dependent apoptosis of leukemic cells, including the population of leukemic progenitors [135]. Administration of SPHK1 inhibitors to mice xenografted with AMLs resulted in an improvement of animal survival [135]. SPHK1 inhibition in leukemic cells resulted in reduced survival signaling from SIP receptor 2 and in selective downregulation of MCL-1 protein; the combination of SPHK1 inhibitors and BH3 mimetics targeting BCL-2 resulted in a synergistic induction of leukemic cell death [135]. However, the potency of this SPHK inhibitor is in the low μM range and this precludes its use in human clinical trials [138]. Nevertheless, these observations are promising and support the development of studies aiming to identify more potent ATP-competitive SPHK inhibitors [138]. These observations clearly support that SPHK1 is a potential therapeutic target for the treatment of AML. Interestingly, SKI-178, a multitargeted inhibitor of SPHK1 and 2, functions also as a microtubule network disrupting agent; this molecule acts as a potent inducer of apoptosis of leukemic cell lines and inhibits leukemia growth in mouse AML models [139]. 

## 4. Targeting the Energetic Metabolism in AML cells

### 4.1. Abnormalities of the Glycolytic Pathway

Metabolic reprogramming, such as the Warbug effect (a key metabolic change consisting in the capacity of cancer cells to consume more glucose than non-proliferating cells, but to preferentially ferment glucose to lactate regardless of oxygen availability) is a common phenotypic feature of cancer cells, allowing their rapid adaptation in response to the high energetic requests necessary to support their high rate of proliferation. The consumption of glucose by cancer cells is required for purposes beyond ATP generation, such as the generation of anabolic intermediates necessary for macromolecule biosynthesis (nucleotides, amino acids and lipids). Compared with mitochondrial respiration, aerobic glycolysis is more efficient in allowing the production of a more sustained metabolic flux for anabolism and maintaining a more favorable redox balance to fuel active cell proliferation [140].

Enhanced glycolysis and tricarboxylic acid gene expression was frequently observed in AMLs and correlated with poor prognosis [141]. The evaluation of serum level of metabolites involved in glucose metabolism showed decreased values of lactate, citrate and glycerol-3-phosphate and increased levels of pyruvate, 2-oxo-glutarate and 2-hydroxyglutarate [141]. Importantly, enhanced glycolysis contributed to decreased sensitivity of leukemic blasts to Ara-C and its inhibition restored cytotoxicity of Ara-C [141]. Patients with a high glycolysis and TCA (tricarboxylic acid cycle) cycle gene expression level displayed a negative clinical outcome [141].

Given the high rate of glucose utilization by proliferating leukemic progenitors, this sugar may become insufficient in the bone marrow microenvironment; thus, it is surprising that AML cells are prone to fructose uptake via the upregulated expression of the fructose transporter GLUT5, whose elevated expression in leukemic blasts is associated with poor outcome [142]. Pharmacological inhibition of fructose uptake improves the leukemic phenotype and increases the sensitivity of leukemic cell to Ara-C [142]. Bone marrow microenvironment plays an essential role in promoting activation of the glycolytic pathway in AML blasts, as supported by in vitro co-culture of leukemic cells with stromal cellular elements [143]. The CXCL 12/ CXCR4 pathway seems to play a key role via mTOR activation [143]. mTOR blocking via rapamicyn inhibited the stimulating effect of stromal cells on the rate of glycolysis in AML blasts [143].

It is important to underline that mTOR signaling is frequently activated in cancer cells and its activation is involved in the control of cancer cell metabolism, altering the expression/activity of many key metabolic enzymes [144]. On the other hand, some metabolic alterations, such as those occurring at the level of increased amino acid or glucose uptake modify mTOR signaling [144]. mTOR is a serine/threonine kinase, composed by two subunits mTOR complex 1 and mTOR complex 2, and, through its activity, promotes anabolic processes, such as protein synthesis, ribosome biogenesis, fatty acid and lipid synthesis and inhibits catabolic processes, such as autophagy [144].

An essential role of mTOR in the glycolytic mechanisms of AML cells is supported by other studies. The serine/threonine kinase mTOR complex 1 is overactivated in the majority of AMLs and promotes glycolysis in these cells and drives glycolysis addiction; mTOR inhibition induces metabolic reprogramming in AML cells; metabolic analysis showed an important role of glucose-6-phopshate dehydrogenase (G6PD) and of the pentose phosphate pathway in sustaining the energetic metabolism of leukemic cells [86]. High G6PD expression in AML cells was associated with reduced patient survival. G6PD is a potential target of vulnerability of AML cells [145].

Other studies have addressed the problem of the occurrence of glycolysis abnormalities present in some AML specific subsets. Recurring mutations of the transcription factor ZBTB7A gene were observed in 23% of AML (8;21) patients [86]. The transcription factor ZBTB7A is important for hematopoietic lineage fate decisions and for regulation of glycolysis. ZBTB7A mutations induce derepression of some glycolytic genes, thus favoring the proliferation of AML cells [146].

Importantly, studies in experimental models of mouse leukemogenesis support a major role of glycolytic components in the leukemogenetic process. Thus, deletion of either pyruvate kinase M1 isoform (PKM1) or lactate dehydrogenase A (LDHA) genes, the most expressed glycolytic genes in murine bone marrow cells, induces HSCs and progenitor cells defects at the level of normal hematopoiesis and impairs leukemia development in mouse leukemia models [147].

To avoid a decrease in intracellular pH due to increased metabolic activity through the glycolytic pathway, lactate molecules and protons are exported out of the cell by an efficient transport system. This function is mediated by proton-lined monocarboxylate transporters (MCTs); the MCT family comprises various members, including MTC1-MTC4, involved in the transport pyruvate, lactate and ketone bodies: the direction in which each of these substrates is transported is dictated by their chemical gradients. MCT1 and MCT4 are highly expressed in various cancers and their expression is often a prognostic factor in these tumors. MTC1 and MTC4 are expressed in AMLs, MTC1 more than MTC4 [148]. MTCs could be exploited as transporters of drugs, such as bromopyruvic acid, to kill AML cells [89]. Alternatively, MTC inhibitors, such as AR-C155858 [149] or syrosingopine [150] may be used as anti-leukemic drugs. Interestingly, syrosingopine induces intracellular lactate accumulation, eliciting synthetic lethality with metformin [150]. 

### 4.2. Abnormalities of the Mitochondrial Energetic Metabolism

Few studies have explored the effect of BCL-2 inhibitors at the level of the compartment of leukemic progenitors/stem cells. The majority of leukemic stem cells were shown to produce low levels of reactive oxygen species and to possess elevated BCL-2 levels: BCL-2 inhibition in these cells reduced oxidative phosphorylation and selectively eradicated leukemic stem cells [151]. Interestingly, studies carried out on progenitors/leukemic stem cells in myelodysplastic syndromes have shown that these cells highly express and are sensitive to BCL-2 inhibitors [152,153]. These observations strongly support the idea that the targeting of oxidative phosphorylation could represent an important vulnerability of AML cells and, particularly of the leukemic stem cell compartment. Various studies have supported the existence of an altered mitochondrial energetic metabolism in AML cells, as well as in the majority of malignant tumors [154]. Mutations of mitochondrial genes, such as *IDH1* or *IDH2* [155,156] or mitochondrial encoded *electron transport chain* genes [157] are frequent in AMLs and contribute to the leukemogenetic process.

IDH proteins are polymeric enzymes, involved in various biologic processes, such as energetic metabolism, histone demethylation and DNA modification and adaptation to hypoxia. The IDH1 enzyme is localized in the cytoplasm, while IDH2 is localized in the mitochondria where is a critical component of the tricarboxylic acid cycle (citric acid or Krebs cycle): both these enzymes catalyze the oxidative decarboxylation of isocitrate to α-ketoglutarate (α-KG) through a two-step reaction, involving first the conversion of isocitrate to oxalosuccinate and the conversion of oxalosuccinate to α-KG (Figure 2). A third IDH enzyme, IDH3, is a heteroterameric NAD^+^-dependent enzyme that, as well as IDH1/IDH2, decarboxylates isocitrate and produces α-KG, NADH and CO_2_ (Figure 3). The different IDH isoforms have overlapping, but not redundant metabolic functions. Thus, IDH1 promotes the activity of several cytoplasmic and nuclear doxygenases that require α-KG as a co-substrate, the generation of nonmitochondrial NADPH and promotes also the carboxylation of α-KG to isocitrate, in turn promoting lipid generation after conversion to Acetyl-CoA. IDH3 promotes the generation of α-KG required for succinate generation and promotes the generation of ATP. Finally, IDH2 regulates the concentration of α-KG and isocitrate in the mitochondria (Figure 3). mutations in IDH catalyze the reduction of α-KG to ®-2-hydroxyglutarate (2HG), an oncometabolite, inducing DNA and histone hypermethylation, altered gene expression and a block of hematopoietic cell differentiation (Figure 2). The oncogenic effects of 2HG are mediated though its capacity to inhibit some demethylases and several dioxygenases, including TET2, a demethylating enzyme. The IDH mutations determine also a reduction of α-KG, leading to reduced levels of prolyl hydroxylases and upregulation of HIF-1α. The *IDH1/IDH2* genes are mutated in about 15–20% of AMLs (7–9% *IDH1* mutations and 12–14% *IDH2* mutations): the large majority of *IDH1* mutations occur at the level of R132 (R132H or R132C being the most frequent mutations), while the *IDH2* mutations more frequently occur at the level of R140 (R140Q) or less frequently at the level of R172 (R172K); *IDH1* and *IDH2* mutations are mutually exclusive [155,156]. 

In the large majority of IDH-mutated patients, IDH mutations are founder mutations; however, in a minority of these patients, IDH mutations are “progressor mutations” [157]. IDH1/IDH2 mutations are mainly associated with DNMT3A, SRFS2, NPM1, ASXL1 and RUNX1 when IDH mutations act as founder mutations, while are mostly associated with TP53 and FLT3-ITD mutations when IDH mutations act as progressor mutations [158].

Molecular analyses and immunohistochemistry studies provided evidence that the IDH1 R132H and R132C mutants show a different distribution pattern among AML genotypes. IDH1-mutated AMLs showed in 62% of cases also a *NPM1* mutation, in 48% a *DNMT3A* mutation, 23% a *FLT3-ITD*, 16% a *NRAS* mutation, and 12% a SRSF2 mutation [159]. Interestingly, 89% of *IDH1-R132H* patients showed a *NPM1* mutation, while in only 33% of *IDH1-R132C* patients a *NPM1* mutation occurred [159]. *IDH1-R132H* was mutually exclusive for *RUNX1, SRSF2* and *ASXL1* [159]. According to these findings, it was proposed that *IDH1-R132H* shows a typical de novo AML pattern, while *IDH1-R132C* shows a more s-AML like genetic pattern [159]. Other studies have shown remarkable differences between IDH2 and IDH1-mutant AMLs, at the level of genetic landscapes [160]. Cytogenetic alterations were observed only in 26% of IDH-mutant AMLs; however, the frequency of an aberrant karyotype was higher in *IDH2-R172*-mutated patients (42%) than in *IDH1-R132* (21%) and *IDH2-R140* (23%)-mutated AMLs [160]. Interestingly, NPM1 mutations were absent among ADH2-R172 patients, while they were frequent among *IDH1-R132* (63%) and *IDH1-R140* (50%) patients [160]. *FLT3* was mutated rarely in patients with *IDH2-R172* (5%), compared to those with mutations in *IDH1-R132* (28%) or *IDH2-R140* (31%) [160]. SRSF2 was much more frequently mutated in *IDH2-R140*-mutated patients (43%) than in patients mutated in *IDH1-R132* (13%) and IDH2-R172 (8%). Finally, *ASXL1* mutations occurred less frequently in *IDH1-R132*-mutated AMLs (5), compared to 22 and 16% in *IDH2-R140*- and *IDH2-R172*-mutated AMLs [160]. 

There are conflicting data about the prognostic impact of *IDH1/IDH2* mutations in AML; a meta-analysis of the literature data indicates that the presence of *IDH* mutations does not change AML prognosis [161]. The mutational association between *IDH2* and *NPM1* seems to confer a prognostic benefit [161]. *IDH* mutations determine the accumulation of the oncometabolite (*R*)-2-hydroxyglutarate (2HG) which induces epigenetic changes by inhibiting demethylases; the increase of 2HG, with the concomitant decrease of α-ketoglutarate, determine decreased prolyl hydroxylases and upregulation of HIF-1α and deregulation of response to hypoxia; high 2HG levels are also responsible for the differentiation blockade [155,156]. Interestingly, *IDH*-mutant AML cells depend on BCL-2 for their survival and are sensitive to the pro-apoptotic effect induced by the BCL-2 inhibitor venetoclax [86]. In line with this observation, in the clinical study that evaluated the response of relapsed/refractory AMLs to single-agent venetoclax, the overall response rate was 19% for the whole AML cell population, but 33% of *IDH*-mutant AMLs displayed a CR/Cri [92]. In a recent study, relapsing/refractory AML patients received venetoclax administration in combination with low-dose Cytarabine or hypomethylating agents: of 11 *IDH*-mutated AML patients, 27% responded to this treatment, with 1 CR, 1 CRi and 1 with morphologic leukemia-free state [162]. The treatment of IDH-mutant AMLs was recently revolutionized by the discovery of potent *IDH1* and *IDH2*-mutant inhibitors. Durable remissions (30% of complete remissions or complete remissions with partial hematologic recovery) were observed among *IDH1*-mutated relapsed or refractory AMLs treated with the mutant IDH1 inhibitor ivosidenib (AG-120) [103]. Importantly, a part (21%) of the responding patients had no residual detectable *IDH1* mutations [163]. Data presented at the December 2018 ASH Meeting, reported the results observed in 258 patients (78 in dose-escalation, 180 in expansion), including 34 patients with untreated AML: 41% of the treated patients achieved either complete remission (CR) or CR with incomplete hematologic recovery [164]. A part of mIDH1 untreated AML patients achieved durable remissions and transfusion independence [164]. 40% of IDH2 relapsed or refractory AMLs achieved a response (19% of complete responses) to treatment with the mutant IDH2 inhibitor enasidenib (AG-221) [165]. Median overall survival was 8.8 months; response and survival were comparable among patients with IDH2-R140 and IDH2-R172 mutations; furthermore, response rates were similar among patients who, at study start, were in relapse or were refractory to standard therapies [166]. Importantly, clearance of mutant-IDH2 clones was associated with occurrence of complete responses [166]. Most of AML patients initially responding to IDH inhibitors relapse. The molecular mechanisms underlying relapse are complex: in the majority of patients, the disease relapse was not associated with the acquisition of new IDH mutations, but is related to a clonal evolution or selection of terminal or ancestral clones [167]; in a minority of patients, new *IDH2* mutations were acquired at relapse in trans, at the level of the *IDH2* allele not bearing the original *IDH2* mutation [168]; finally, in another minority of patients *IDH*-mutant AMLs develop resistance to isoform-selective IDH mutation by isoform switching from mutant IDH1 to mutant *IDH2* and viceversa [169]. One of the most frequent toxicity caused by both Ivosidenib and Enasidenib is the so-called differentiation syndrome, elicited by the massive in vivo differentiation of leukemic blasts induced by these agents; this clinical syndrome may be fatal if not treated. A recent evaluation on the trials until now performed showed that the incidence of this syndrome is high, amounting at 40% of the treated AML patients [170]. 

As above stated, 8% of AMLs had a mutation in one or more of the mitochondrial encoded electron transfer chain genes; complex IV and I genes were the most frequently (4.5% and 2.5%, respectively) mutated [157]. Complex I and IV mutations were associated with TP53 mutations and were associated with worse overall survival [157].

Interestingly, also a part of non-*IDH* mutated AMLs display an IDH^mut^-like DNA hyper-methylation condition, related to the overexpression of the enzyme BCAT1, a cytosolic aminotransferase for branched-chain amino acids (BCAAs) [171]. A fraction of IDH^WT^ AMLs overexpress BCAT1 and these AMLs are associated with a poor prognosis compared to IDH^WT^ AMLs associated with normal/low BCAT1 levels [171]. Overexpression of BCAT1 in leukemia cells determines a decrease of α-ketoglutarate, mimicking a condition observed in IDH-mutant AMLs, and induces DNA hypermethylation via altered TET activity [171]. BCAT1 is particularly overexpressed in leukemic cell populations enriched in leukemic stem cells and BCAT1^high^ AMLs show a pronounced enrichment for leukemic stem cell signatures [171]. Interestingly, a BCAT1 overexpression plays a significant role also in the pathogenetic mechanisms underlying blast crisis in chronic myeloid leukemia [172].

Various studies indicate that in AML many properties of mitochondria, including mitochondrial translation, mitochondrial DNA copy number and basal oxygen consumption rates are differentially regulated in comparison with normal hematopoietic cells. Recent studies have further shown the presence in leukemic cells of peculiar abnormalities of mitochondrial metabolism. AMP-activated protein kinase (AMPK) is a key metabolic checkpoint kinase, tuning cellular metabolic state with proliferation by promoting catabolism and inhibiting anabolism. AMPK is a heterotrimeric serine/threonine kinase promoting catabolism, including glucose uptake, glycolysis, fatty acid oxidation and autophagy; activation of AMPK inhibits mTORC1-induced protein biosynthesis and fatty acid biosynthesis through inactivation of the phosphorylation of acetyl CoA carboxylase. The studies carried out on AMPK in AML cells have shown that either an activation or an inhibition of this enzyme exerts an inhibitory effect on these leukemic cells. Thus, some studies have documented an inhibitory effect induced by AMPK activation. An initial study has shown that the AMPK pathway is functional in AML cells and when activated by metformin blocks the catalytic activity of mTOR [173]. As a consequence of the inhibitory effect on mTOR, metformin decreases AML cell proliferation [173]. The level of AMPK activation induced by either glucose deprivation or metformin in AML blasts is controlled by ERK: thus, AMLs showing a strong constitutive ERK activation exhibit a lower Amp activation [174]. This negative crosstalk between AMPK and ERK could diminish the antileukemic activity of metformin [174]. Specific activation of AMPK by the AMPK agonist GSK621 induces a cytotoxic effect in AMLs, but not in normal hematopoietic cells. AMPK-mediated cytotoxicity requires mTORC1 activation that is unique to AML cells and requires the EIF2α/ATF4 signaling pathway [175]. Other studies have shown that AMPK loss induces apoptosis in AML cells. Studies on murine models of MLL-AF9-induced AML revealed an essential role for AMPK at the level of leukemic stem cells; in fact, AMPK deletion inhibited leukemogenesis and depleted LSCs by reducing the expression of the glucose transporter GLUT1, reducing glucose flux and inducing oxidative stress in the hypoglycemic bone marrow microenvironment [176]. In contrast to AML cells, normal HSCs were not dependent upon AMPK during homeostasis or following transplantation or in stress conditions [176]. Particularly, some studies have unveiled the existence of a peculiar mitochondrial dynamics in leukemic stem cells [91,117]. In LSCs, it is observed the specific activation of the metabolic stress activator AMPK that induces upregulation of FIS1 which, in turn, regulates mitophagy activity (a process through which damaged mitochondria are degraded) that functions to sustain leukemic stem cell homeostasis [177,178]. The blockade of this pathway leads to cell cycle arrest, cell differentiation and a reduction of leukemic stem cell self-renewal [178].

Targeting mitochondrial oxidative phosphorylation seems to be an effective strategy for the eradication of therapy-resistant chronic myeloid leukemia stem cells [179]. In fact, an increased mitochondrial oxidative metabolism at the level of CML LSCs was observed and treatment with imatinib plus tigecycline, an antibiotic that inhibits mitochondrial protein translation eradicates therapy-resistant chronic myeloid leukemia stem cells. [179].

Very interestingly, the analysis of the mechanisms underlying the response of elderly AML patients to the therapy based on decitabine or azacytidine in combination with venetoclax indicate a major role at the level of the targeting of the oxidative metabolism in the LSC populations. The combination of venetoclax with azacytidine or decitabine demonstrated remarkably high rates of durable complete remissions in previously untreated elderly patients with AML [92,180]. A study carried out on a group of responding patients treated with azacytidine and venetoclax provided fundamental information about the mechanisms undergoing the significant anti-leukemia effects induced by this drug combination.

Analysis of leukemic cell populations enriched in LSCs provided evidence of disruption of the tricaboxylic acid cycle, as evidenced by decreased α-ketoglutarate and increased succinate levels, a finding compatible with an inhibition of electron transport chain complex II [168]. All these metabolic perturbations determine a suppression of oxidative phosphorylation, efficiently targeting LSCs [181]. These observations strongly support the view that the disruption of the metabolic machinery driving energy metabolism is an efficient therapeutic strategy to attempt to eradicate LSCs [181]. The pronounced effects elicited by azacytidine and venetoclax administration at the level of the LSC compartment through a specific targeting of the metabolic properties of leukemic cells certainly help to understand the rate, the kinetics and the durability of responses achieved by this drug regimen; however, the very pronounced effect of this drug combination cannot be ascribed only to the effect on LSC-specific metabolic properties, but is seemingly related also to the capacity of azacytidine to induce a cellular condition that makes AML cells more susceptible to BCL-2 inhibition [90].

In a parallel study, the same investigators have further characterized the metabolic targeting elicited by azacytidine and venetoclax in AMLs [182]. In this study, Jones and coworkers have isolated the LSC population sorting AML cells according to their endogenous ROS levels: AML blasts with low ROS production are enriched in cells with functional properties of LSCs, while cells with high ROS levels are representative of the bulk AML blasts [182]. LSCs were shown to display an enhanced capacity of uptake of amino acids, particularly of glutamine and proline [182]; the LSC population resulted also to be dependent upon amino acids for its survival [182]. In contrast, amino acid depletion does not affect the normal HSC population, thus indicating that the dependency displayed by LSCs is a leukemia-specific property [182]. In contrast to amino acid depletion, glucose depletion was inhibitory for ROS-high, but not ROS-low AML cells [122]. OXPHOS in LSCs, but not in bulk AML blasts, was selectively sensitive to loss of amino acids, seemingly because LSCs are less metabolically flexible and cannot compensate for amino acid loss [182]. Interestingly, the analysis of AML cells of patients undergoing therapy with azacytidine and venetoclax showed and rapid inhibitory effect on LSCs, associated with a reduction of amino acid levels and metabolism in these cells [182]. Furthermore, it was shown that azacytidine and venetoclax decrease amino acid uptake into LSCs, this effect being associated with a concomitant reduction of OXPHOS in an amino acid-dependent manner [170]. These observations were made in primary AML cells; however, in relapsing AML cells, the administration of azacytidine and venetoclax failed to induce any significant effect on amino acid metabolism and this escape mechanism of relapsing AML blasts seems to be related to an increased fatty acid metabolism: in these leukemic cells, targeting both fatty acid uptake and amino acid metabolism may represent a valuable therapeutic strategy [182]. The facts that relapsing LSCs do not respond to azacytidine and venetoclax in terms of metabolic inhibition is in line with preliminary clinical data showing that venetoclax-based regimens are much less effective in relapsed patients than in therapy-naïve AML patients [163]. Furthermore, the metabolic diversity of relapsing LSCs is in line with other studies showing that LSCs at relapse are markedly increased in their frequency and are phenotypically different [183]. 

55% of primary AMLs display upregulated mitochondrial DNA synthesis; cytoplasmic nucleoside kinases are required to sustain this increased mitochondrial DNA synthesis in leukemic cells and the inhibition of their activity reduces the mitochondrial DNA synthesis [184]. AML cells have increased cytidine nucleoside kinase activity and can be leveraged to selectively target oxidative phosphorylation in AML [184].

AML cells exhibit in addition to dependence on glycolysis, rely on oxidative phosphorylation (OXPHOS) for bioenergetic and biosynthetic processes. Glycolysis is under the negative control of OXPHOS and is induced by tricarboxylic acid cycle (TCA)-mediated allosteric inhibition of glycolytic enzymes (“Pasteur effect”); therefore, genetic or pharmacological OXPHOS inhibition should result in different tumor cell responses depending on the capacity to activate compensatory glycolysis: thus, tumor cells with a reduced capacity for compensatory glycolysis would be more sensitive to OXPHOS inhibition. Interestingly, a compound of recent identification, IACS-010759, acting as an inhibitor of complex I of the mitochondrial electron transport chain, exerted a potent apoptotic effect on AML cells reliant on OXPHOS, through a mechanism related to energy depletion and reduced aspartate production [185]. In models of AML, leukemia growth was potently inhibited following IACS-010759 treatment at tolerated doses [185]. These observations indicate that AML cells are dependent on OXPHOS and its disruption creates an environment of energy and macromolecule depletion that leads to cell cycle arrest, apoptosis and cell differentiation [185]. IACS-010759 is currently being evaluated in phase I clinical trials in relapsed/refractory AML [185]. Interestingly, a recent report showed a pronounced activity of IACS-010759 in NOTCH1-mutated T-ALL models; in these leukemic cells, pharmacological inhibition of complex I with IACS-010759 induces catastrophic changes in mitochondria, with induction of mitochondrial reactive oxygen species, DNA damage, and activation of the compensatory mTOR pathway [186]. In addition to IACS-010759, other Complex I inhibitors were reported and their pharmacodynamic profile was assessed through analysis of the modulation of oxygen consumption rate, aspartate and specific transcriptional changes [187]. Very interestingly, an enhanced OXPHOS activity by leukemic blasts was associated with chemoresistance. In fact, Farge and coworkers have explored the properties of AML cells residual after Ara-C treatment and have shown that these chemoresistant leukemic cells are not enriched in leukemic cells with progenitor/stem phenotype, but by leukemic cells exhibiting high levels of ROS, high mitochondrial mass and active polarized mitochondrial mass and active polarized mitochondria, all features consistent with a high OXPHOS status [188]. Targeting mitochondrial protein synthesis, electron transfer or fatty-acid oxidation induced a metabolic shift to low-OXPHOS and increased the sensitivity of these cells to Ara-C [188]. These observations strongly support a new strategy to treat AML residual disease.

In line with these findings, Lim and coworkers observed that in pediatric AML patients the abundant expression of miR-106a was associated with treatment resistance (induction failure) [189]. Since several candidate targets of this mRNA were involved in oxidative phosphorylation (such as ATP5S, ATP5S2-PTCD1, NFUDA10, NDUFC2 and UQCRB), it was concluded that treatment resistance occurs via modulation of genes that are involved in oxidative phosphorylation [189]. miR-106a is a component of the miR-106a-363 cluster, including in addition to miR-106a also miR-20b, miR-19b, miR-92a and miR-363 and this cluster was recently shown to be upregulated in adult AML patients, particularly those with adverse cytogenetics [190]. Furthermore, it was confirmed that mitochondrial processes, such as oxidative phosphorylation and electron transport chain components are targets of miR-106a: in line with this finding, miR-106a-overexpressing AMLs display increased numbers of mitochondria [190].

The increased content of mitochondria in AML leukemic blasts seems to be related to a peculiar mechanism of transfer of mitochondria from bone marrow stromal cells to leukemic blasts through AML-derived tunneling nanotubes; this organelle transfer process requires the NADPH oxidase-2 (NOX2) activity generating superoxide, which stimulates stromal cells to release mitochondria [191]. Inhibition of NOX2 activity induces AML apoptosis and exerts a therapeutic effect on AML mouse models [191]. A recent study better defined the pathogenetic role played by NOX2 activation in AML development. AML blasts induce a senescent phenotype in the stromal cells within the bone marrow microenvironment, driven by p16INK4a expression; in turn, the p16INK4a-expressing senescent stromal cells promote blast AML survival and proliferation, a phenomenon mediated through AML-induced NOX2-derived superoxide [192]. Targeting of NOX2 elicited a reduction of bone marrow stromal cell senescence and a reduced AML proliferation [192]. As above stated, NOX2 plays a key role in superoxide production. Hole and coworkers have measured superoxide production by primary AML blasts, showing that these cells display overproduction of NOX-derived ROS [133]. Leukemic cells can produce ROS either via mitochondria and/or NOX oxidases: mitochondrial superoxide production in AML blasts is moderately reduced compared to normal hematopoietic cells; in contrast, extracellular superoxide release by leukemic blasts is markedly increased compared to normal hematopoietic cells; in contrast, extracellular superoxide release by leukemic blasts is markedly increased compared to normal hematopoietic cells (65% of primary AMLs display this property) [193]. NOX2 was shown to be the main mediator of the enhanced superoxide production in AML blasts; in a minority of cases, the enhanced superoxide production was not related to NOX2, but to other mechanisms [193]. High superoxide production in AML cells was associated with reduced anti-oxidant expression, a condition that is expected to increase the expression of stress signaling mediated by p38^MAPK^; however, contrary to this expectation, low or undetectable constitutive p38^MAPK^ activation was observed in AML blasts, thus indicating that AML blasts can evade the oxidative stress responses originated from high levels of ROS by inactivating or attenuating the activation of p38^MAPK^ [193]. Finally, it was provided evidence that ROS released by AML cells promotes their proliferation [193].

The development of peculiar abnormalities of mitochondrial metabolism may create metabolic vulnerabilities in AML subtypes. *EVI1* proto-oncogene is deregulated by chromosomal translocations in some cases of AML; EVI1 overexpression in hematopoietic cells alters cellular metabolism and promotes overexpression of genes related to purine and pyrimidine metabolism, amino acid metabolism (alanine, aspartate, glutamate, arginine and proline), the pentose phosphate pathway and glycolysis [194]. A screen using short hairpin RNAs identified mitochondrial creatine kinase CKMT1 as necessary for survival of EVI1-positive leukemic cells in AML samples EVI1-positive [194]. CKMT1 promotes the conversion of creatine to phosphocreatine and drives in EVI1-transformed AMLs mitochondrial function and ATP production [194]. EVI1 promotes CKMT1 expression by repressing RUNX1 [194]. Small molecule CKMT1 inhibitors selectively inhibited the viability of EVI1-mutated AMLs [194]. According to these findings it was proposed that the targeting of this enzyme may represent a promising therapeutic strategy for this AML subtype [194]. Suppression of arginine-creatine metabolism by the small molecule cyclocreatine altered the mitochondrial respiration and ATP production of EVI1-positive AML cells, thereby inhibiting their proliferation [135]. Gene sets related to the GSK3 and WNT pathways were the most inhibited by cyclocreatine [195]. These findings allowed to demonstrate that the blockade of the creatine pathway inhibits GSK3 and silences WNT pathway, thus blocking the physiological control of GSK3 on the WNT pathway through CTNNB1 (catenin-β). Thus, inhibition of creatinine kinase pathway allows the anti-leukemic effects and the induction of leukemic differentiation elicited by GSK3 inhibition, without activation of the pro-leukemogenic WNT signaling pathway [195].

Since AMLs are reliant on OXPHOS, OXPHOS inhibition could represent an effective therapeutic strategy to inhibit these cells. Several drugs, including metformin, atovaquone, and arsenic trioxide, are used at clinical level for non-oncologic indications, but growing evidences indicate their potential use as OXPHOS inhibitors [196]. CPI-613, a lipoate analogue that blocks pyruvate dehydrogenase (PDH) and α-ketoglutarate dehydrogenase (KGDH), inhibiting mitochondrial metabolism: CPI-613 induces PDH inactivating phosphorylation via stimulation of PDK, inducing a consequent collapse of mitochondrial function and the activation of multiple tumor cell death pathways; in parallel, CPI-613 induces a large, tumor-specific production of mitochondrial ROS from the E3 subunit of KGDH, inducing an inhibition of this enzyme [197]. This drug was evaluated in phase I clinical studies in patients with refractory/relapsed AML [198,199]. The results of these studies showed an acceptable safety profile and preliminary promising efficacy results, particularly in patients with poor cytogenetics (with an overall response rate of 50% and a median survival of 6.7 months) [199]. The analysis of the response of AML patients stratified according to age showed that while in the group of patients treated with chemotherapy alone there was a clearly better survival for younger patients (<60 years) compared to older patients, in the group of patients treated with chemotherapy plus CPI-613 the survival of older patients was similar to that achieved for younger patients [200]. These observations have supported the development of a phase III randomized study comparing younger and older refractory/relapsing AML patients for their response to their response to standard chemotherapy supplemented or not with CPI-613. The U.S. Food and Drug Administration has granted orphan drug designation for the treatment of peripheral T-cell lymphoma.

The study of some AML subtypes showed the complexity of the abnormalities of the energetic metabolism of leukemic cells. Thus, some recent studies have characterized the peculiar features of the energetic metabolism in *FLT3^ITD^*-mutated AMLs. Thus, it was shown that FLT3-ITD causes an increase in aerobic glycolysis through AKT-mediated upregulation of mitochondrial hexokinase and renders these leukemic cells dependent on glycolysis and, thus, sensitive to glycolysis inhibitors, able to potentiate the cytotoxic effect induced by FLT3 inhibitors [201]. A second study explored the effect of FLT3 inhibitors on glycolytic activity of *FLT3^ITD^*-mutated AML cells, showing that these inhibitors exert a more pronounced inhibitory effect on the glycolytic activity, but a less marked inhibition of TCA cycle activity and respiratory function [202]. This reduced inhibitory activity on mitochondrial metabolism is due to the continuous uptake of glutamine, supporting oxidative metabolism: glutamine supports the TCA cycle and glutathione production following FLT3 inhibition; in line with this observation, combined inhibition of FLT3 and of glutamine metabolism determined an increased cell death of FLT3^ITD^ leukemic cells [202]. The effects of combined FLT3-TKIs and glutaminase inhibitors can be rescued by the glutamine downstream product α-ketoglutarate [202]. Several other genes associated with metabolic pathways demonstrated significant synergistic lethality with FLT3 inhibitors, such genes involved in fatty acid synthesis and phosphatidylinositol metabolism [202]. Interestingly, inhibitors of glutaminase, catalyzing the conversion of glutamine to glutamate, inhibit OXPHOS and induce arrest of proliferation of leukemic cells; glutaminolysis inhibition activates mitochondrial apoptosis and synergistically sensitizes leukemic cells to priming with BCL-2 inhibitors [203,204]. Other studies on the cellular metabolome showed that FLT3 inhibition causes marked alterations in central carbon metabolism, resulting in impaired production of the antioxidant factor reduced glutathione; this metabolic condition is further impaired by ATM or G6PD inactivation [205]. Studies in various *FLT3^ITD^* AML models provided evidence that FLT3 inhibition induces mitochondrial oxidative stress that determines the triggering of an apoptotic response, whose extent is exacerbated by ATM/G6PD inactivation [205]. In conclusion, these studies showed that the use of an agent that intensifies mitochondrial oxidative stress in combination with a specific FLT3 inhibitor may represent a useful strategy to improve elimination of AML cells to treat FLT3 mutated AMLs. A recent study provided evidence that distinct metabolic features, such as higher abundance of nucleosides, reduced abundance of tryptophan and increased levels of L-carnitine, differentiate *FLT3-ITD* AMLs from FLT-WT childhood AMLs [206]. A recent study clarified the mechanism through which *FLT3-ITD* AML cells prevent the induction of mitochondrial apoptosis. In fact, Hospital and coworkers showed that the serine/threonine kinase Pim2, overexpressed in many cancers including AMLs, is essential for the survival of *FLT3-ITD* AML cells [207]. Pim2 inhibition induced upmodulation of Bax and apoptosis. Ribosomal protein S6 kinase A3 (RSK2), a serine/threonine kinase involved in the mitogen-activated protein kinase cascade encoded by the RPS6KA3 gene is a Pim2 target and mediates cell survival downstream of Pim2 [207].

Glutamine, either directly or indirectly through ammonia production acts as a nitrogen source for pyrimidine biosynthesis (Figure 4). Glutamine stimulates orotate synthesis, through an effect on mitochondrial carbamyl phosphate synthesis [208] (Figure 4). Orotate is synthesized from dihydroorotate through an enzymatic reaction catalyzed by dihydroorotate dehydrogenase (DHODH), a mitochondrial protein localized on the outer surface of the inner mitochondrial membrane (Figure 4). DHODH is the rate-limiting enzyme in the *novo* synthesis of pyrimidines, and as such may control the rate of cell division. This enzyme links the electron transport chain to pyrimidine production and thus to the maintenance of cell viability. A high throughput analysis carried out by Sykes and coworkers provided evidence that DHODH is a key metabolic regulator in pyrimidine synthesis and a metabolic target in differentiation therapy of AML [209]. This study was based on the screening of more than 330,000 compounds and 11 of the 12 compounds acting as inducers of differentiation of AML blasts pertain to the group of DHODH inhibitors [209]. In this study, it was shown that AML blasts when starved of pyrimidines, displayed signs of induction of cell differentiation and of apoptotic cell death; furthermore, cells surviving to this treatment were more differentiated, as shown by their morphology, immunophenotyped and the loss of their leukemia-initiating capacity [149]. These studies have led to the identification of DHODH inhibitors, such as brequinar [210] or ML390 [211], whose inhibitory activity was not sufficiently strong for acceptable clinical activity. Importantly, studies with other DHODH inhibitors more potent than brequinar for their anti-leukemic activity, confirmed their capacity to induce differentiation of AML blasts [212]. 

As of 2018, clinical trials with two DHODH inhibitors, BAY240234 (Bayer AG, Leverkusen, Germany) and ASLAN003 (Aslan Pharmaceutical, Singapore), are currently underway in AML patients. BAY240234 is a selective low-nanomolar inhibitor of human DHODH enzymatic activity, inhibiting proliferation of AML cell lines in the range and showing in vivo efficacy in monotherapy in patient-derived AML xenograft models [213]. Parameters of pharmacodynamic activity were shown by induction of AML cell differentiation and by increased plasmatic and tumor levels of dihydroorotate levels, consequence of DHDOH inhibition [213]. Based on this preclinical background, a phase I study (NCT 03404726) was designed to evaluate the safety profile and the anti-leukemic activity in myeloid malignancies (Table 3). ASLAN003 is a novel, potent small molecule DHDOH inhibitor, developed in AML by Aslan Pharmaceuticals. This inhibitor was more active than brequinar, considered the reference standard of DHODH inhibitors, in inducing differentiation of AML blasts [214]. The large majority of primary AML cells is highly or moderately sensitive to ASLAN003 [214]. This inhibitor displayed clear anti-leukemic activity in various in vivo AML models [214] and is under evaluation in a phase IIa study in patients with AML. A preliminary evaluation of this ongoing clinical study showed early signs of clinical activity and a favorable safety profile [215].

DHODH inhibitors represent a new therapeutic tool in AML differentiation therapy. A recent study showed that isobavachalcone acts as a potent DHODH inhibitor, induces differentiation of AML cells and in combination with adriamycin exerts a marked anti-leukemia effect in AML mouse models [216]. Mechanistically, the cell differentiation and apoptosis induced by this DHDOH inhibitor is in part mediated by c-Myc suppression, by directly reducing c-Myc gene transcription and inducing c-Myc protein degradation via proteasome-mediated degradation [216]. Interestingly, AMLs with high DHODH activity have a poor prognosis [216].

The prodrug leflunomide and its active metabolite teriflunomide are clinically approved DHDOH inhibitors used for treatment of rheumatoid arthritis and multiple sclerosis, respectively. However, the clinical use of these drugs is limited by off-target effects and long in vivo half-life. Although these drugs display some anti-tumor activity, their clinical utility is strongly limited by their low-potency [216]. Interestingly, some studies have shown that leflunomide elevates p53 protein levels in HeLa cells, suggesting that pyrimidine biosynthesis links mitochondrial respiration to the p53 pathway [217]. Interestingly, a recent study, reporting the identification of chiral tetra-hydroindazoles as potent DHODH inhibitors (HZ00 and HZ05), showed that these new DHODH inhibitors, as well as brequinar, act as potent inducers of TP53 expression [218]. Particularly, HZ compounds accumulate cancer cells, including leukemic cells, in S-phase, increase TP53 synthesis, and synergize with an inhibitor of TP53 degradation to reduce tumor growth in vivo [218]. Another study reported the characterization of another DHODH inhibitor, PTC299, initially described as a VEGF inhibitor; this compound induced apoptosis and differentiation of AML cells [219]. PTC299 induced inhibition of cell proliferation and of VEGF production, both linked to a decrease in uridine nucleotides by targeting DHODH [219]. Interestingly, PTC299 is a more potent DHODH inhibitor compared to other previously reported DHODH inhibitors, as assayed on isolated mitochondria [219]. PTC299 had a potent anti-leukemic activity in AML models [159]. PTC299 seems to have several advantages on previously reported DHODH inhibitors, derived from its greater potency, good oral bioavailability, lack of off-target kinase inhibition and myelosuppression [219]. In another recent study Hsu and coworkers have evaluated the effects of DHODH inhibitors on cellular metabolism showing that Brequinar treatment not only leads to a rapid depletion of pyrimidines and downstream metabolites, but also to changes in TCA cycle metabolites, including succinate and fumarate [220]. The metabolites act as regulators (inhibitors) of TET enzymes; these enzymes convert 5-methyl-cytosine to 5-hydroxymethylcytosine (5-hmC) in DNA and high 5-hmC levels are associated with high transcription rate [221,222]. An increase in global 5hmC, suggesting an increased TET-enzyme activity, paralleled the cellular differentiation induced by brequinar in AML cells [220]. In conclusion, these studies suggest that DHODH, an enzyme essential for de novo biosynthesis of pyrimidine-based nucleotides, is a promising therapeutic target for AML. The absolute relevance of the DHODH-driven pyrimidine pathway for tumorigenesis is further supported by very recent studies. In fact, the time-resolved analysis of tumor formation by mitochondrial DNA-depleted cells or with genetic manipulations of OXPHOS did led to the conclusion that DHODH-driven pyrimidine biosynthesis, rather than OXPHOS-mediated production, is essential for tumorigenesis [223].

Tumor cells often develop extensive reprogramming of cellular energy metabolism to meet the high metabolic requests of proliferation. A frequent strategy adopted by cancer cells consists in developing auxotrophy to specific amino acids required to meet protein biosynthesis. Thus, metabolic inhibitors of amino acid-depleting enzymes have been explored for their biologic effects and as potential cancer therapeutics. From this point of view, four amino acids are mostly involved in this metabolic reprogramming process: glutamine, asparagine, arginine and cysteine. Glutamine is the most abundant amino acid in plasma and participates in many energy-generating and biosynthetic processes for the proliferation of cancer cells by entering the TCA cycle. Glutamine enters into cells via transporter (Solute Carrier family 1 neutral Amino Acid transporter 5, SLC1A5) and in converted to glutamate and ammonia by glutaminase (GLS) in the mitochondria; subsequently, glutamate is further oxidatively deaminated into α-ketoglutarate (α-KG) through two different mechanisms: glutamate dehydrogenase (GDH); aminotransferases (Figure 5). The aminotransferases are involved in maintaining the stability of intracellular amino acid pools to provide nonessential amino acids for cell growth: phosphoserine aminotransferase (PSAT1) for synthesis of serine; glutamic pyruvate transaminase (GPT) for synthesis of alanine; glutamic oxaloacetate transaminase (GOT) for synthesis of aspartate. In addition, glutamine can also directly provide nitrogen for asparagine by Asparagine Synthetase (ASNS, Figure 5). α-KG enters the TCA cycle and can provide energy and in involved in the production of nucleotides, ATP, some amino acids, lipids and glutathione in mitochondria [163,164,165]. Glutaminase controls the first step of the glutaminolysis pathway and represents a promising target for developing antitumor drugs [224,225,226]. Several recent studies have provided evidence that leukemic cells can utilize glutamine as a carbon source for energy production through TCA cycle and redox homeostasis [203,227,228]. While wild-type IDH1/IDH2 convert isocitrate to α-KG in cytosolic and mitochondrial compartments, mutant IDH1/IDH2 generates a neo-enzymatic activity, catalyzing the reduction of α-KG, with production of 2-hydroxyglutarate (2-HG); glutamine represents a source of α-KG converted to 2-HG by mutant IDH1/IDH2 [229,230]. Given this function of glutamine in providing an essential metabolite for the oncogenic activity of mutant IDH1/IDH2, it is not surprising that BTES, a glutaminase inhibitor, reduced the growth of IDH-mutant AML cells [231].

Matre and coworkers explored the expression of the two isoforms of GLS1: KAG, kidney isoform; GAC, glutaminase C isoform. GAC transcript was significantly more expressed in complex or del 5/7 cytogenetics and in core-binding factor AML (t(8;21) and inv(16)) than in normal karyotype AMLs [232].

As discussed above, FLT3-ITD-mutated AMLs represent another AML subtype in part dependent on glutamine for their energetic metabolism. Particularly, it was shown that these leukemias depends on glycolysis and glutamine supply for their energetic metabolism: FLT3-TKI treatment blocks glucose uptake and mostly glycolysis, rendering the cells dependent on glutamine metabolism [200]. The combined treatment with a FLT3-TKI and a glutaminase inhibitor induces a pronounced anti-leukemic effect [200]. Furthermore, another recent study, as above discussed, provided evidence that AML stem cells are uniquely reliant on amino acid metabolism, and particularly on glutamine metabolism, for oxidative phosphorylation and survival [182]. These observations strongly support the use of glutaminase inhibitors, together with anti-leukemia-specific drugs for AML treatment.

Calithera Biosciences Inc. (South San Francisco, CA, USA) is developing CB-839, a leading candidate from a program of glutaminase inhibitors, for the potential oral treatment of multiple cancers, including AML. Phase 1 trial (CX-839-003) was carried out in relapsed/refractory AML patients [233]. CB-839 was well tolerated and showed some signs of clinical activity [233]. Other studies showed that CB-839 synergizes with azacytidine in inhibiting the growth of AML cell lines and primary AML cells [234]. These studies have promoted the ongoing clinical study evaluating CB-839 in association with azacytidine in AML and myelodysplastic patients (Table 3).

In conclusion, targeting glutamine metabolism represents an effective and potentially widely applicable therapeutic strategy for treating multiple types of leukemia. Future studies will evaluate the therapeutic impact of glutaminase inhibition, particularly in combination with adjuvant drugs that further perturb redox mitochondrial state [235].

Asparagine targeting represents another important strategy for metabolic targeting of AML cells. Asparagine is synthesized in mammalian cells through a reaction involving the enzyme asparagine synthetase (ASNS) that catalyzes the synthesis of asparagine and glutamate from aspartate and glutamine in ATP-dependent aminotransferase reaction (Figure 6). Based on animal growth studies, asparagine is usually defined as a “nonessential” amino acid. At the clinical level, inhibition of asparagine uptake by cancer cells is achieved using the enzyme L-asparaginase, an enzyme catalyzing the conversion of L-asparagine to aspartic acid and ammonia (Figure 6). Asparaginases are currently approved only for the treatment of acute lymphoblastic leukemia (ALL). However, asparaginase showed also a significant anti-leukemic activity in preclinical studies of AML. These studies have shown that L-asparaginase plays an important role in the control of glutamine metabolism in AML cells [227]. Thus, Willems and coworkers have shown that glutamine contributes to leucine import into the cells, which controls the Target of Rapamycin Complex 1) (mTORC1); in line with this finding, knockdown of the SLC-145 high-affinity transporter for glutamine induced apoptosis of AML cells [227]. Interestingly, the anti-cancer drug L-asparaginase harbors an intrinsic glutaminase activity and induces, through this glutaminase activity, a marked inhibition of mTORC1 and produces a pronounced apoptotic response in primary AML cells [227] (Figure 6). Thus, these studies have shown that glutaminase deletion by asparagine indirectly inhibits mTOR activity via decreased leucine uptake in AML cells [227]. The intrinsic glutaminase activity of various types of Asparaginases used for clinical purposes is supported also by clinical studies on leukemic patients, showing that L-ASP administration caused a clear decline of plasma glutamine levels [236]. Chan and coworkers have studied leukemic cells that express or not ASNS and, through this analysis, have reached the conclusion that the glutaminase activity of asparagine preparations is required for their anti-leukemic activity in ASNS-positive cells, but not in cells not expressing ASNS [237]. Since the clinical toxicity of L-ASP is thought to derive from its glutaminase activity, these findings suggest the hypothesis that glutaminase-negative variants of L-ASP could exert a stronger therapeutic activity than wild-type L-ASP for ASNS-negative leukemias [237]. In line with this hypothesis, novel low-glutaminase asparaginase variants have been developed on the backbone of the FDA-approved *Erwinia chrysantemi* L-asparaginase: the variants displayed strong anti-T-ALL and -B-ALL activity, while exhibiting reduced acute toxicity features [238]. However, a recent study on a preclinical AML model showed that among various L-ASP preparations, only those exhibiting glutaminase coactivity induce durable anti-leukemic responses [239].

Clinical experience concerning the use of L-asparaginase in AML is limited to the use of Ecoli-derived Asparaginase in association with multi-agent chemotherapeutic regimens in patients with relapsed/refractory disease and as first-line therapy in patients who do not accept blood transfusions; more recently, some patients with relapsed/refractory AML were treated with L-ASP from Erwinia chrysanthemi in the context of the clinical trial NCT02283190. The clinical studies carried out on ALL patients treated with L-ASP have shown that leukemic blast sensitivity to this drug is related to low/absent ASNS enzyme in ALL cells. Interestingly, a recent study showed that AML patients exhibiting ASNS haploinsufficiency due to chromosome 7 monosomy, exhibit a high sensitivity to L-ASP [240]. A recent study showed a potential strategy to improve the response of cancer cells to asparaginase. This strategy was based on the lowering of the activity of general control nonderepressible 2 (GCN2), a kinase playing a key role in the cellular response to amino acid deprivation; in a preclinical study, it was shown the capacity of small-molecule inhibitors of GCN2 to induce or to increase the sensitivity of leukemic cells to asparaginase, by preventing the induction of ASNS [241]. Importantly, AML cells were shown to be highly sensitive to the combined treatment with GCN2 inhibitors and asparaginase [241]. 

Targeting arginine metabolism has been considered another potential therapeutic strategy to act on another metabolic vulnerability of AML cells. Arginine is involved in several metabolic pathways, such as synthesis of nitric oxide, polyamines and other amino acids, including glutamate and proline (Figure 7). Cells may obtain arginine from the external milieu or can synthesize it from citrulline, through two enzymatic steps involving argininosuccinate synthetase-1 (ASS1) and arininosuccinate ligase (ASL). ASS1, the rate-limiting enzyme in arginine synthesis, forms argininosuccinate from aspartate and citrulline (Figure 7). ASS1 is not expressed in many tumor cells and its absence confers a proliferative advantage to these cells. Because ASS1-deficient tumors rely on extracellular arginine for their survival, arginine deprivation is a therapeutic strategy for these cancers. The absent expression of ASS1 is a feature of many AMLs, while ASL is normally expressed in AML blasts [225]. ASAS1 is suppressed in the majority of AMLs through an epigenetic mechanism involving promoter methylation [225]. Arginine deaminase (ADI) represents a tool to deplete ASS1-deficient cells of arginine: a pegylated form of ADI (ADI-PEG) was used to deplete ASS1-deficient tumor cells of arginine. ADI-PEG administration to mouse AML xenografts depletes arginine in vivo and reduces leukemia burden, while normal hematopoiesis is not inhibited [242]. In vitro studies indicate that about 50% of AMLs are sensitive to the pro-apoptotic effect of ADI-PEG: sensitive AMLs are characterized by low ASS1 expression [242]. A second study confirmed the existence of an arginine dependency of AML blasts, showing that the leukemic cells express the arginine transporters CAT-1 and CAT-2B and that in most cases deficiencies in the arginine-recycling pathway enzymes arginine succinate synthase and ornithine transcarbamylase were detected [243]. In line with these findings, ADI-PEG induced arginine deprivation and consequent proliferation arrest and apoptosis of AML blasts [243].

Arginine starvation causes cell death of cancer cells targeting mitochondrial function, as supported by three lines of evidence: arginine starvation modulates mitochondrial electron transport chain gene expression; arginine induces ATF4-ASNS, thus diverting cellular aspartate toward increased asparagine and suppresses the aspartate-malate shuttle; mitochondria deficient cancer cells are resistant to the effect of arginine starvation [244]. This metabolic shift induced by arginine starvation reduces mitochondrial OXPHOS and inhibits nucleotide biosynthesis [244]. The efficacy of ADI-PEG was evaluated as single agent in 43 relapsed/refractory/high-.risk AML patients in the context of a phase II clinical trial (NCT 01910012) [244]. The majority of treated AML patients displayed ASS deficiency [245]. The results of this study showed that single-agent ADI-PEG therapy resulted in a CR rate of 4.7% and a stable disease rate of 16.3% in the 43 AML patients, with response durations of 7.5 and 8.8 months for two patients achieving CR [245]. The duration of arginine depletion correlated with disease control and the tumors depleted of arginine for the longest time had the best responses [245]. In conclusion, the treatment based on weekly ADI-PEG20 monotherapy resulted in a low response rate in AML patients, with a low toxicity profile [245]. Further studies are required to better explore the therapeutic potentialities of arginine deprivation in AML patients and to define potential drug combinations [245]. It is important to note that in some studies a recombinant preparation of the enzyme human arginase, catalyzing the conversion of arginine to ornithine and urea, leading to arginine depletion was used. Particularly, BCT-100, a clinical grade pegylated recombinant human arginase was under evaluation in AML patients, as well as in some solid tumors [243]. Interestingly, AML blasts were shown to alter the immune microenvironment though enhanced arginine metabolism. Particularly, AML blasts have an arginase-dependent ability to inhibit T-cell proliferation and hematopoietic stem cells and to modulate the polarization of monocytes [246].

An accumulating body of evidence indicates that branched-chain amino acids (BCAAs) are essential nutrients for leukemic cell growth and are used in various biosynthetic pathways and as an important source of energy. Leucine, isoleucine and valine are BCAAs and constitute a group of essential amino acids (Figure 8). These amino acids have many important biological functions including protein synthesis, glucose homeostasis, nutrient-sensitive signaling pathways (such as PI3K, mTOR), energy production [247]. Cells control the levels of BCAAs via uptake and amino acid catabolism. The main BCAA transporter is SLC7A5 which imports BCAAs and other essential amino acids in exchange of glutamine. BCAAs can be catabolized to generate nitrogen and carbon groups. The enzymes responsible for the early steps of BCAA catabolism are shared for the various BCAAs and include the cytosolic BCAT1 catalyzing the conversion of BCAAs into branched-chain chetoacids by transferring the BCAA amino group onto α-KG, generating glutamate (Figure 8) and the mitochondrial BCAT2 enzymes [247]. It has been recently demonstrated that BCAT1 is overexpressed in a subset of AMLs and in CML [86,162]. BCAT1 was shown to be overexpressed in *TET2/IDH* wild-type AMLs at the level of the leukemic stem cell population, where its enhanced activity leads to e depletion of its substrates BCAAs and of α-KG [86]. Acting through this mechanism, BCAT1 overexpression determines in a part of IDH-mutant AMLs a condition of low TET2 activity, caused by a reduction of α-KG and thus similar to that observed in *TET2*-mutant AMLs [86]. The levels of BCAT1 expression in *TET2/IDH*-WT AMLs have a negative prognostic impact in IDH-mutant AMLs: this finding is not surprising in that IDH-mutant AMLs already display reduced activity of α-KG-dependent dioxygenases. At the variance with the observations made in AMLs, in CML BCAT1 has been shown to increase cellular concentrations of BCAAs to stimulate mTOR activity, to drive protein synthesis, growth and survival [162]. These findings support the development of strategies attempting to increase α-KG levels by inhibition of BCAT1 to inhibit leukemic stem cells and thus to improve the outcome of AML patients, expressing high levels of BCAT1, not associated with TET2 or IDH mutations. Other recent findings support the existence of a link between BCAAs and normal and leukemic stem cells. BCAA levels are regulated through catabolic steps, mainly controlled through the branched-chain α-keto acid dehydrogenase (BCKD) complexes. The activity of the BCKD holoenzyme is determined by the regulation of its phosphorylation status, controlled by a mitochondrial-targeted 2C-type Ser/Thr protein phosphatase (PPM1K): this enzyme promotes BCKD dephosphorylation and through this mechanism activates BCAA catabolism [248]. This function is essential to sustain glycolysis and quiescence in HSCs, as well as in LICs and is mediated by a molecular mechanism preventing MEIS1/p21 ubiquitination/degradation; in contrast, BCAAs enhance ubiquitination of MEIS1 and p21 [248]. PPM1K was shown to be particularly important to maintain the population of leukemia stem cells [248]. These observations are relevant in view of the development of strategies for leukemia treatment through metabolic manipulation.

Most mitochondrial proteins are encoded in the nucleus, translated at the level of cytoplasmic ribosomes and then imported into mitochondria. A peculiar subset of mitochondrial proteins characterized by cysteine rich domains and located at the level of mitochondrial membrane interspace and oxidized and folded by the so-called mitochondrial inter-membrane space assembly (MIA) pathway [249]. Interestingly, genes encoding substrates of the MIA pathway are overexpressed in LSCs, compared to bulk AMLs [249]. Targeting the FAD-linked sulfhydryl oxidase ALR, an integral part of the MIA machinery, using the small molecule inhibitor MitoBloCK-6 induced differentiation and apoptosis of primary AML cells and reduced their clonogenic activity [249]. Cox17, a copper chaperone that promotes respiratory chain complex IV assembly by loading copper into the respiratory complex, was identified as the primary target of ALR inhibition [250]. According to these findings, it was suggested that inhibitors of ALR or Cox17 may represent a potential new tool for AML therapy [250].

In addition to glucose and glutamine, fatty acids are an additional key energy source. They can be incorporated from the extracellular media or can be potentially obtained from hydrolysed triglycerides. De novo synthesis of fatty acids is required for membrane synthesis and therefore for cell growth and proliferation. Fatty acid synthesis is an anabolic process that starts from the conversion of acetyl CoA to malonyl CoA by acetyl CoA carboxylase; malonyl CoA is then involved in the elongation of fatty acids through fatty acid synthase (FASN). Additional modifications of fatty acids can be carried out by elongases and desaturases. Fatty acids are catabolized by the fatty acid oxidation (FAO) pathway. Recent studies have shown the relevance of FAO for leukemia cell function. FAO is a complex metabolic pathway and is composed of a cyclical series of reactions that result in the shortening of fatty acids (two carbons per cycle) and that generate in each round NADH, FADH_2_ and acetyl CoA, until the last cycle when two acetyl CoA molecules are originated from the catabolism of a four-carbon fatty acid. NADH and FADH_2_ that are generated by FAO enter the electron transport chain (ETC) to produce ATP. The first step of FAO is fatty acid activation, producing long-chain acyl-CoA catalyzed by the long-chain acyl-CoA synthetase, which is the initial essential prerequisite for subsequent long-chain fatty acid catabolism. There are 26 genes encoding Acyl-CoA with different affinities for activating fatty acids of various length. Long Acyl-CoAs are unable to penetrate the inner mitochondrial membrane and necessitate of a specific transport system provided by the Carnitine Palmitoyl Transferase system, comprising the components Carnitine Palmitoyl Transferase I (CPT1), Carnitine Acylcarnitine Translocase (CPCT) and CPTII. The CPTI family of proteins, shutting long-chain fatty acids into mitochondria, represent the rate-limiting step of FAO.

Multiple abnormalities of FAO pathway were reported in AMLs. Some studies were focused on CPT1, a key enzyme catalyzing the rate limiting step of FAO pathway. CPT1A was found to be overexpressed in primary AMLs, compared to normal BM; This finding was confirmed also through the comparison of leukemic and normal CD34^+^ cells [251]. The stratification of AML patients according to CPT1A levels showed that CPT1A-high expression was associated with adverse outcomes (including survival and molecular risk parameters) [251]. Other studies showed that pharmacologic inhibition of CPT1 either using Etomoxir [252] or ST1326 [252] induced a dose- and time-dependent apoptosis in primary AML cells. CPT1 inhibitors synergized with anti-leukemic drugs (Cytarabine) or with a BCL-2 inhibitor (ABT-737) to induce apoptosis of AML cells [252,253].

Another abnormality of FAO in AML is observed at the level of its regulation by the post-translational modifying enzyme PHD3. PHD enzymes form a class of enzymes devoted to regulating metabolism in response to a modification of the environmental conditions; these enzymes are a conserved family of oxygen- and α-KG-dependent enzymes involved in the regulation of the glycolytic metabolism through HIF. Acetyl-CoA Carboxylase 2 (ACC2), the gatekeeper of FAO, was identified as a substrate of PHD3: through ACC2 activation, PHD3 represses the oxidation of long-fatty acids. In normal conditions, PHD3 represses FAO activity in conditions of nutrients abundance and cells with low PHD3, such as the majority of AMLs, have persistent FAO activity independently of external nutritional conditions [254]. Low PHD3 expression may be considered as a biomarker to identify AML patients, candidate for treatment with FAO inhibitors [255].

Other studies have addressed the problem of the interaction between the stromal microenvironment and the metabolic status of AML blasts. Thus, it was shown that bone marrow stromal cells promote AML cell survival via a metabolic shift from pyruvate oxidation to fatty acid oxidation, which determines mitochondrial uncoupling the induces a decreased mitochondrial ROS production and reduced intracellular oxidative stress [256]. Avocatin B, a potent FAO inhibitor, cooperates with Cytarabine to induce the killing of chemoresistant AML cells co-cultured in the presence of bone marrow stromal adipocytes [256].

Several recent studies have attempted to elucidate the molecular mechanisms responsible for the reprogrammation of metabolic pathways observed in AML cells. In this context, particularly interesting was a recent study exploring the effects on metabolic pathway of HLX (H2.0-like homeobox transcription factor), a homeobox gene overexpressed in 85% of AMLs [257]. HLX overexpression in HSCs and HPCs leads to a myeloid differentiation block [256]. HLX overexpression leads to downregulation of genes encoding electron transport chain components and upregulation of PPARδ (Perixosome Proliferator Activated Receptor delta) gene expression and results also in AMPK (AMP-activated Kinase); pharmacological modulation of PPARδ signaling relieves the HLXL-induced myeloid differentiation block, while AMPK inhibition reduces the viability of AML cells [257]. It is important to note that PPARδ is a regulator of fatty acid metabolism, plays an essential role in HSC stemness regulating mitochondrial mitophagy and promoting HSC asymmetric divisions; during hematopoietic differentiation, PPARδ is downregulated, leading to progressive increase in mitochondrial mass [258]. Therefore, PPARδ and HLX that controls its expression represent a molecular system to adapt the energetic metabolism to HSC fate. Interestingly, PPARδ is overexpressed, as well as HLX, in a subset of monoblastic AMLs [259].

Interestingly, a recent study explored the metabolic impact of leukemic development not at single cell level or tissue level, but at the level of the whole organism [260]. The limitation of previous studies based on metabolomic analysis of leukemic cells is that do not consider leukemia from an organismal perspective, where leukemic cells represent only a small proportion of host body mass, and the existence of a competition between leukemic cells and normal tissues for limited amounts of systemic glucose could represent a critical component of leukemic growth [260]. The systemic analysis in appropriate leukemic models showed that leukemic cells gain a competitive advantage over normal tissues by developing multiple mechanisms to induces a diabetes-like physio-pathologic condition in the host (impairment of both insulin sensitivity and insulin secretion to provide leukemic cells with increased glucose; induction of high-level production of IGFBP1 from adipose tissue to stimulate insulin sensitivity) and, thus, inducing a subversion of systemic glucose metabolism to allow leukemic progression [260]. These observations offer the opportunity to develop an additional therapeutic strategy to try to limit leukemia development/progression acting at the level of systemic metabolism.

## 5. Metabolomics and BH3-Profiling Studies: Two Precious Tools for the Discovery and Prediction of Drug Sensitivity of Leukemic Cells

The omic sciences of systemic biology, including genomics, transcriptomics and proteomics have given a great and fundamental contribution to the study of human AMLs, as well as of other cancers. The omic spectrum was recently complemented with metabolomics, a global quantitative assessment of endogenous metabolites within a tumor at the level of the tissue tumor and of plasma [261,262]. Metabolomics involves the global detection of several small molecule metabolites and the identification of metabolic reprogramming, assessed as changes at the level of metabolic pathways and phenotypes [261,262]. Metabolomics has an emerging importance in cancer diagnosis, prognosis and recurrence, in identifying new cancer biomarkers and developing new cancer therapeutics [263,264]. The discovery of numerous metabolic abnormalities in AMLs strongly supports the importance of complementing the characterization of AMLs with the analysis of metabolic profiling. Metabolomics requires the application of analytical techniques, such as nuclear magnetic resonance spectroscopy and mass spectrometry to identify and assess metabolite levels in biological samples from tissues and represents the most useful platform to detect metabolic abnormalities in plasma, urine or tissue samples [261,262,263,264]. Metabolomic studies have contributed to the discovery and characterization of metabolic abnormalities of AML cells associated with genomic alterations in metabolic enzymes: this was the case of *IDH1* and *IDH2* mutations leading to the generation of the oncometabolite D-2-hydroxyglutarate [263]. The evaluation of AML metabolic profiling has given also a fundamental contribution to the identification of various metabolic abnormalities in some AML subsets: identification of distinct metabolic features differentiating *FLT3-ITD* AMLs from *FLT3-WT* childhood AMLs [206]; identification of the peculiarities of amino acid metabolism of human AML stem cells [182]; identification of the creatine kinase pathway as a metabolic vulnerability in EVI1-positive AML [194]; analysis of the enhanced fatty acid and lipid metabolism in AMLs, particularly in *IDH*-mutant AMLs [265]. All these observations support the inclusion of the analysis of metabolic profiling among the omic studies to improve the assessment methods of AML patients for precision medicine.

Future developments, using inhibitors of the BCL-2 family will take advantage on the development of specific functional assay, such as the BH3 profiling assay. This kind of assay could contribute to predict clinical response to BH3-targeting agents and the to optimize the initial therapy and to functionally define the mechanisms of the resistance developing during anticancer therapy with these agents. BH3 profiling is a functional assay that can be used to predict cellular responses to stimuli based on measuring the responses of mitochondria to perturbations by a panel of BH3 domain peptides [266]. This assay is able to measure the spectrum of pro-survival proteins on which a tumor cell depends for its survival and the tendency of a tumor cell to undergo apoptosis under deprivation of a specific BCL-2 family member [266,267,268].

A new technique was recently developed; this technique was called dynamic BH3 profiling and measures the early changes in net pro-apoptotic signaling at the mitochondrion level induced by chemotherapeutic agents in cancer cells, requiring only short incubation times [269]. Practically, in the standard BH3 profiling, mitochondria are exposed to proapoptotic BH3 peptides and then the amount of mitochondrial outer permeabilization is measured; in the dynamic profiling, the intact cells are first exposed to one or more agents to be evaluated, and then is evaluated whether the treatment causes an increase in apoptotic priming, corresponding to an increased sensitivity of the mitochondria to the BH3 peptide [269]. Thus, BH3 profiling measures the relative interactions of pro-apoptotic and anti-apoptotic proteins to determine whether a tumor cell is near the threshold to activate apoptosis. Cells are considered “primed” or “unprimed” based on whether they are near or far from the apoptotic threshold. Thus, dynamic profiling offers a functional approach to predict clinical response by measuring death signaling induced by specific drugs in tumor cells from patients [270]. BH3 profiling significantly correlated with progression-free survival after treatment with carboplatin and paclitaxel in patients with ovarian adenocarcinoma [269,270]. The dynamic BH3 profiling was used to assess the sensitivity of various hematopoietic malignancies to Venetoclax, including AMLs, and this assay predicted a high sensitivity to this drug of Blastic Plasmocytoid Dendritic Cell Neoplasms [271]. These observations have supported the development of a clinical trial assessing the safety and therapeutic efficacy of venetoclax in patients with blastic plasmocytoid dendritic cell neoplasm (NCT03485547).

A recent study showed the great potentialities of BH3 profiling in helping to define optimal therapeutic strategies in some AML subsets. Thus, Saito and coworkers, combining various experimental approaches, integrating population- and single-cell-level genomics and in vivo functional assessment reached the conclusion that in FLT3-ITD mutated AMLs the treatment with kinase inhibitors was efficacious in removing the large majority of leukemic cells [272]. This finding is in line with clinical studies showing that midostaurin, a FLT3 inhibitor, improved 5-year event-free survival and overall survivals in FLT3-ITD-positive AML patients when administered in combination with standard chemotherapy [273]. Although targeting FLT3-ITD alone resulted in the elimination of the large majority of AML cells, few kinase-resistant leukemic cells survived to the treatment; Dynamic BH3 profiling showed that kinase-resistant leukemic cells are sensitive to BCL-2 inhibitor: co-treatment with kinase inhibitor and BCL-2 inhibitor resulted in complete eradication of human AML cells resistant to kinase inhibition [272]. These observations strongly support the concomitant administration of FLT3-inhibitors and BCL-2-inhibitors in FLT3-ITD AMLs.

The co-operative dynamic BH3 profiling was recently used at experimental pre-clinical level to identify effective pro-apoptotic anti-leukemic drug combinations. Using this technique, it was shown that the MCL-1 downregulator TG02 sensitizes to the BCL-2 inhibitory BAD-BH3 peptide, while the BCL-2 antagonist ABT-199 sensitizes to the MCL-1 inhibitory NOXA-BH3 peptide, and the two agents synergize in dual-sensitive cells to induce apoptosis [274]. According to this assay, drugs were dichotomized as either agents sensitizing to BCL-2 antagonism or agents sensitizing to MCL-1 antagonism and it was shown the efficacy of combining an agent from each category in apoptosis assays [275].

Other studies, comparing AML sensitivities to various apoptogenic compounds with BH3 profiling and prosurvival BCL-2 family protein expression have confirmed that BH3 profiling is a valuable tool to predict apoptosis induction by some chemotherapy agents and targeted therapies. However, the additional assessment of prosurvival BCL-2 family proteins is required to assess resistance to MCL-1 and BCL-XL [276]. Interestingly, dynamic BH3 profiling showed that statins, inhibitors of 3-hydroxy-3-methylglutaryl coenzyme A reductase, primed leukemic cells to venetoclax-mediated apoptosis [277].

## 6. The Extrinsic Apoptotic Pathway: TRAIL-Rs, Caspase-8 and Their Abnormalities in AMLs

The extrinsic pathway of apoptosis is triggered by activation of death receptors (DRs) through binding of their specific ligands. Death receptors are a subset of the Tumor Necrosis Factor Superfamily (TNFRS) that includes various members, such as Tumor Necrosis Factor 1 (TNFR1, TNFRSF1a), CD95 (TNFRSF6, Apo-1 and Fas), TNF-Related Apoptosis-Inducing Ligand (TRAIL) receptor-1 and -2 (TRAIL-R 1-2; DR 4-5, and TNFRSF10 a-b) DR3 (TNFRSF 25) and DR6 (TNFRSF 21). All these receptors are characterized by the common property of inducing apoptosis when triggered by their cognate ligands or by agonistic antibodies [278].

DRs are key and fundamental mediators of two essential processes, extrinsic apoptosis and necroptosis, that decide whether a cell lives or dies. These two death pathways are strictly interwinded and are regulated by a complex cascade of events determining the final death event. Necroptosis is a form of programmed necrosis. Necroptic stimuli promote the interaction of RIP1 and RIP3 kinases under conditions in which caspase-8 in not active [4]. This complex, known as complex IIb, mediates necroptosis [4]. Interestingly, there is a bridge between apoptosis and necroptosis in normal and pathological conditions. 

The initial step of DR-mediated activation of both apoptosis and necroptosis is a molecular complex consisting of various proteins: Fas Associated via Death Domain (FADD), Aspartate-Specific Cysteine Protease (Caspase-8) or Capsase-10 and FLICE-Like Inhibitory Protein (cFLIP) [279]. The importance in the cell death, regulated by apoptosis or necroptosis is crucial is crucial since induction of apoptosis implies the death of a cell, without alarming surrounding cells, not inducing in these cells an inflammatory response; in contrast, when necroptosis is induced, dying cells release pro-inflammatory cytokines and other alarmins, promoting an inflammatory response in the surrounding microenvironment [280].

TRAIL (Apo2 Ligand, CD253, TNF-SF10) induces apoptosis through binding a series of death receptors. TRAIL binds to DR4 (TRAIL-R1) or DR5 (TRAIL-R2) and induces caspase-8-dependent apoptosis. TRAIL is able to bind to other two TRAIL-Rs: decoy receptor 2 (DcR1, also known as TRAIL-R3) and decoy receptor 2 (DcR2, also known as TRAIL-R4). DcR1 functions as a TRAIL-neutralizing decoy receptor that binds TRAIL but is unable to trigger apoptotic cell signaling. DcR2 contains a cytoplasmic domain, lacking a functional death domain, thus being unable to induce apoptosis upon TRAIL binding. However, DcR2 has been found to be capable of transducing signaling pathways, such as NF-kB and AKT. Thus, the induction of apoptosis is a property restricted solely to TRAIL-R1 and TRAIL-R2, due to the presence of a death domain within their cytoplasmic tail. The death domain is necessary and sufficient for the recruitment of the adaptor protein FADD which, in turn, enables the association of the initiator caspases.

TRAIL-R1 and TRAIL-R2 are O-glycosylated and N-glycosylated and their glycosylation at both sites regulates tumor-cell sensitivity to the TRAIL ligand [281,282]. Furthermore, TRAIL-R glycosylation plays also a prominent role in regulating receptor/receptor interactions and trafficking [283].

Apart from inducing cell death, the binding of TRAIL to its receptors TRAIL-R1, TRAIL-R2 and TRAIL-R4 results in the activation of NF-kB, a transcription factor mediating pro-inflammatory immune responses. RIPK1, in addition to be involved in the induction of necroptosis, is also involved in TRAIL-mediated NF-kB induction by activating the Inhibitor of kB Kinase complex (IKK), which is involved in phosphorylation of IkB, with consequent degradation and activation and nuclear translocation of NF-kB. TRAIL-induced NF-kB activation may mediate TRAIL resistance and TRAIL-induced cell proliferation, a phenomenon frequently observed in apoptosis-resistant tumor cells [283]. These observations are important because explain why the simple TRAIL activation is a limited anti-cancer strategy because many cancers, in addition to being TRAIL resistant, use the endogenous TRAIL-TRAIL-R system to fuel their survival/proliferation [283].

Many studies have explored the sensitivity of AML blasts to the pro-apoptotic effects of TRAIL, showing that the large majority of these cases are resistant to this death ligand [223,224]. The study of the pattern of TRAIL-R expression provided evidence that TRAIL-R3 and TRAIL-R4 are frequently expressed in AML, while TRAIL-R1 expression is usually observed in AMLs with monocytic differentiation phenotype [284,285,286]. The presence of TRAIL-R1/TRAIL-R2 expression is associated with TRAIL decoy receptor expression and does not confer sensitivity to TRAIL [283]. More recent studies have shown that AMLs characterized by high FLT3 expression, either associated or not with FLT3-ITD mutations, display high TRAIL-R expression [287]. The expression of TRAIL-R2 and TRAIL-R3 was significantly increased in AML unfavorable risk groups; furthermore, high co-expression of TRAIL-R2 and TNF-R1 on leukemic blasts is an independent predictor of poor prognosis [288]. Finally, another study showed that TRAIL-R3 expression on leukemic AML blasts is associated with poor outcome and induces apoptosis-resistance which can be bypassed by targeting TRAIL-R2 with a specific agonistic antibody [289]. Various agents, such as anti-leukemic drugs [283], synthetic triterpenoids [290], triptolide [291], the multikinase inhibitor sorafenib [292], AKT inhibitors [293], the poly (ADP-ribose) polymerase inhibitor olaparib [294] are able to render AML blasts sensitive to TRAIL-induced apoptosis. In the majority of cases, this sensitizing effect is mediated through TRAIL-R1/TRAIL-R2 up-modulation and/or c-FLIP down-modulation [290,291,292,293,294]. Interestingly, phorbol esters downmodulate Protein Kinase C epsilon (PKCε) and, through this effect, sensitizes primary AML cells to both the apoptogenic and the differentiative effects of TRAIL [295]. A more recent study showed that PKCε is a key regulator of mitochondrial function, modulating mitochondrial ROS production through a regulation of the expression of regulatory proteins of redox homeostasis, electron transport chain flux, outer mitochondrial membrane potential and transport [296].

Multiple mechanisms are responsible for resistance of leukemic cells to TRAIL monotherapy through various mechanisms involving up-regulation of TRAIL decoy receptors, increased expression of anti-apoptotic genes, such as the NF-kB target genes C-FLIP, BCL-2, MCL-1. The low expression of the PU.1 transcription factor may contribute to mediate TRAIL resistance in AML cells through a lack of inhibitory effect on NF-kB-mediated up-regulation of anti-apoptotic genes such as c-FLIP, MCL-1, BCL-2 and BCL-XL upon TRAIL treatment [297]. Furthermore, the PU.1 deficiency represents a condition of low stimulation of *TRAIL-R2* gene transcription, positively controlled by PU.1 level [297]. 

Inhibitors of apoptosis (IAPs) family of genes encode a family of proteins with antiapoptotic function; important members of this family are XIAP and c-IAP-1 and c-IAP-2 [298]. These proteins are characterized by RING and UBC domains and act by binding to proapoptotic factors and by ubiquitylating and promoting degradation of these factors [298]. XIAP protein is overexpressed in about 43% of primary adult AMLs and is associated with poor prognosis [299] and in 25% of childhood AMLs associated with an immature blast phenotype and with a negative prognosis [300]. Furthermore, particularly elevated XIAP levels were observed in acute mixed lineage leukemias, higher than those observed in other AML subtypes [301]. Implication of XIAP in ineffective induction of cell death by TRAIL in leukemia was shown in various resistant leukemic cell lines [302]. The study of the effect of various XIAP inhibitors (dequalinium chloride or embelin) supported a major role for XIAP in protecting AML cells from apoptosis; particularly, these agents induced differentiation and impaired clonogenic capacity of primary AML cells [303] and synergized with TRAIL in inducing apoptosis of AML blasts [304]. These observations have led to develop various strategies to inhibit XIAP function to induce killing of AML cells and to restore their sensitivity to the apoptotic effects of TRAIL. Early phase I/II clinical trials evaluating the effect of a XIAP antisense oligonucleotide (ASO), AEG35156, in combination with chemotherapy demonstrate a better outcome in patients with AML refractory as compared to a single induction regimen [305]. However, these optimal results could not be reproduced in the randomized phase II study [306]. Probably the lack of clinical results could be ascribed to the incapacity of ASO drugs to achieve long-term efficacious suppression of its target. Small-molecule inhibitors of more recent identification, such as RO-BIR2, may circumvent this drawback, eliciting a rapid and durable inhibition of their IAP targets. In fact, this XIAP inhibitor developed by the Roche Co. (Basel, Switzerland) targets the BIR2 domain, involved in multiple molecular contacts with caspase-3 and -7: these caspases are released, following treatment with RO-BIR2, and induce PARP activation and apoptosis [307]. RO-BIR2 clearly synergize with TRAIL or with Ara-C in inducing cell death of primary AML cells, supporting phase I clinical studies evaluating these drug associations [307]. Another promising inhibitor is CDKI-73: this is an orally CDK9 inhibitor, able to induce apoptosis of AML cells through targeting of the anti-apoptotic proteins BCL-2, MCL-1 and XIAP [308], particularly active against MLL-rearranged AMLs [129]. The inhibitory effect of XIAP on TRAIL-induced apoptosis may be abrogated by molecules like SMAC, a mitochondrial component of the apoptotic machinery promoting apoptosis by binding to and degrading multiple IAPs, including XIAP, c-IAP-1 and c-IAP-2 [309]. SMAC mimetics represent a group of engineered analogues of SMAC for pharmacological use that work in a similar manner to native SMAC, inducing cell death. Preclinical studies have shown that SMAC mimetics induce cell death in a large proportion of primary AML samples: 51% of cases are clearly sensitive; 21% are moderately sensitive and 28% are resistant [310]. Interestingly, most of AML samples resistant to Ara-C, are sensitive to the SMAC mimetic BV6 [310]. Sensitivity of AML blasts to SMAC mimetic correlated with low XIAP expression and presence of NPM1 mutations [310]. Interestingly, the study of the mechanisms responsible for the induction of cell death in leukemic cells showed that AML blasts resistant to apoptosis are induced to cell death by induction of necroptosis via RIP1, RIP3 and MLKL activation [311]. Necroptosis inhibitors rescue AML cells from SMAC mimetic-induced cell death [311]. AML cells exhibit the autocrine release of TNF-α, further increased by SMAC mimetics [311]. 

Other studies have shown that AML blasts secrete TNF-α and other inflammatory cytokines: particularly, M3, M4 and M5 (promyelocytic, myelomonocytic and monocytic leukemias) AMLs secrete TNF-α [312], with preferential secretion by cells with an immature phenotype corresponding to leukemia stem/progenitor cells, while more mature leukemic cells preferentially secrete Interleukin-1β (IL-1β) [313]. This autocrine production of inflammatory cytokines is responsible for constitutive NF-kB signaling, a factor antagonizing TRAIL-induced signaling. In line with these observations, a recent study reported that dexamethasone, a potent anti-inflammatory agent, improves the outcome of hyperleukocytic AML patients receiving intensive chemotherapy [314].

Monovalent and bivalent SMAC mimetics have been developed and some of them are under clinical evaluation as anti-cancer drugs [315]. Usually these drugs have demonstrated suppression of their target, induction of an apoptotic response and in some cases anti-tumor activity [315]. Although the SMAC mimetics are under evaluation in some hematological malignancies, at the moment no clinical trial involves the study of these drugs in AML patients. LCL161, an oral SMAC mimetic/IAP antagonist in under clinical evaluation in various tumors, including myelofibrosis. Significantly increased levels of inflammatory cytokines, including TNF-α, have been observed in patients with myelofibrosis; cIAP protein, including XIAP, expression is increased in CD34^+^ myelofibrosis cells and cIAP/TNF-α-induced NF-kB activation is stronger in myelofibrosis compared to normal CD34^+^ cells [316]. These findings have supported the evaluation of LCL161 in myelofibrotic patients who have failed or are intolerant/ineligible for JAK inhibitor therapy; an objective response rate of 30% was observed in these patients [317]. Interestingly, response to treatment with LCL161 was associated with reduction of c-IAP-1, while resistance to treatment or relapse are associated with high baseline XIAP or increasing levels during treatment [317].

The induction of autophagic mechanisms could represent an additional mechanism of TRAIL resistance operating in leukemic cells. In fact, treatment of tumor cells with autophagy inducers (i.e., trehalose) exerted an inhibitory effect on TRAIL-induced apoptosis, markedly delaying its timing [318]. Other studies have shown the existence of a connection between autophagy and necroptosis induced by TRAIL. Inhibition of early/mid-stages of autophagy prevented cell death; in contrast, inhibition of autophagy at later stages did not inhibit TRAIL-induced death [319]. These observations can be interpreted that some components of the autophagy machinery contribute to cell death by providing a scaffold for necroptosis signaling [319]. In line with this interpretation, necrosome was shown to form a complex with the autophagy receptor p62 [319]. In agreement with these studies, it was shown that various apoptosis defects that block TRAIL-mediated cell death, operating at the level of different steps of the apoptotic pathway, induce a shift from apoptotic signaling to cytoprotective autophagy [320]. Inhibition of TRAIL-induced response promotes an effective apoptotic response, caspase-8 dependent [320]. 

The initial finding that TRAIL can induce toxicity of cancer cells by induction of apoptosis without causing toxicity in mice has strongly stimulated the development of drug candidates, mainly agonistic anti-TRAIL-R1 and anti-TRAIL-R2 that activate TRAIL apoptotic signaling [278]. The outcome of clinical trials with these pharmacologic agents acting as TRAIL-R agonists has been, however, disappointing [278]. The major factors contributing to this lack of clinical activity are related to (a) the suboptimal TRAIL receptor clustering by these agonists; (b) the resistance of cancer cells to TRAIL-mediated apoptosis related to various mechanisms; (c) the use by many cancer cells of the endogenous TRAIL-TRAIL-R system to maintain proliferation/survival [278]. To bypass these limitations, a novel, second-generation of TRAIL receptor agonists comprising a human IgG1-Fc linked to native single-chain TRAIL-receptor binding domain (scTRAIL-RBD) monomers that are covalently connected by glycosylated linkers, resulting in two sets of trimeric receptor binding domains [321,322]. One of these molecules, ABBV-621, was synthesized: each ABBV-621 molecule possesses six death receptor binding sites and this maximize receptor clustering [321,322]. This compound exhibited potent single agent activity in AML and lymphoma cell lines following 24 hr treatments; importantly, the induction of apoptosis by ABBV-621 in targeted leukemic cells was rapid, with induction of major apoptotic-related events (caspase activation, mitochondrial depolarization, phosphatidylserine exposure) [321,322]. ABBV-621 was very active in xenograft models of various human solid tumors and human leukemias/lymphomas [321]. ABBV-621 strongly synergized with venetoclax or navitoclax in inducing apoptosis of leukemia cell lines in vitro and in vivo [322]. These observations support the conclusion that targeting the extrinsic apoptotic pathway with ABBV-621 alone or in combination with venetoclax to harness the intrinsic apoptotic pathway may be of potential therapeutic benefit in hematologic malignancies. These observations have supported the development of a phase I clinical study to evaluate the safety, tolerability and efficacy of ABBV-621 in previously treated advanced solid tumors and hematologic malignancies, including AMLs (NCT 03082209).

Procaspase-8 is expressed under form a zymogen, encompassing a N-terminal pro-domain, containing two DED domains (DED1 and DED2), a large protease subunit containing the catalytic site (p18), a short linker region, and a small subunit (P10). Procaspase-8 binds to the DED domain of death receptor-associated FADD through a pocket present in its DED1, thus giving rise to the formation of DISC. FADD is an adapter protein, containing a C-terminal death domain (DD) and an N-terminal Death Effector domain (DED). After the initial formation of this heterodimer, additional procaspase-8 molecules are recruited to the DISC and interact with DED2 of the prior procaspase-8, through a second pocket present at the level of DED1 [280,281]. The ensemble of this process results in a DED-mediated procaspase-8 oligomerization, resulting in an oligomer that contains on average six procaspase-8 molecules for a single FADD molecule [280,281]. The oligomer formation allows homodimerization of the proteolytic domains of procaspase-8 and consequent autoproteolytic cleavage of an aspartate residue (D834), located between the linker and small subunit, thus allowing the initial activation of procaspase-8, completed by cleavage of an aspartate connecting DED2 and the large subunit [280,281].

The main targets of activated caspase-8 are represented by the effector caspase-3 and -7, present as dimers activated by caspase-8-mediated cleavage between their small and large subunits. Activated caspase-8 can also cleave the protein BID and cleaved BID then activates the effector proteins BAX and BAK and this represents a mechanism to circumvent the inhibition of effector caspases by X-linked Inhibitor of Apoptosis (XIAP) [280,281]. XIAP is inhibited by SMAC/DIABLO, which is released from the mitochondrial intermembrane space through BAX/BAK channels in OMM [280,281].

Two recent studies have provided evidence that the Caspase-8 gene is frequently mutated in AMLs and its mutation may contribute to the malignant leukemic phenotype. Both these studies were carried out by Li and coworkers [323,324]. In a first study, these authors have reported the occurrence in 58% of primary AMLs of four different types of mutations at the level of the P10 small subunit of Caspase-8: pAsn775 Thr (26.7%), Y465S Stop (9.6%), Q482Stop (5.8%), Phen491Thr (16.4%) [325]. The presence of these mutations does not seem to be preferentially associated with any recurrent mutation typically observed in AMLs [325]. Functional studies have shown that AMLs with mutant Caspase-8 display resistance to some chemotherapics, including etoposide, and confer resistance to TRAIL-induced apoptosis [323]. The lack of cytotoxic response to TRAIL triggering was related to the incapacity of mutant Caspase-8 to dimerize under TRAIL stimulation [325]. In another study, the same authors reported the occurrence of the Caspase-8 Q482H mutant in 7.5% of AML patients [324]. The functional properties of this Caspase-8 mutant are similar to those reported for the other P10 Caspase-8 mutants [326].

The identification of frequent Caspase-8 mutations in AMLs impairing the response to TRAIL agonistic agents is an important finding that must be carefully evaluated in pre-clinical and clinical studies aiming to induce apoptosis of AML cells through TRAIL-R triggering.

Interestingly, an experimental study evaluated the effect of the pan caspase inhibitor emricasan in combination with the SMAC mimetic Birinapant [327]. The AML cells are sensitive to birinapant-induced cell death and this effect is potentiated, and not prevented, by the pan caspase inhibitor emricasan [327]. In line with these findings, deletion of Caspase-8 sensitized AML cells to birinapant [327]. Induction of necroptosis by these two agents warrants clinical investigation as a potential opportunity in AML patients [327].

## 7. ONC201: A TRAIL—Inducing Small Chemical Compound

ONC 2011 in the founding member of a novel class of anti-cancer compounds called imipridones, that is currently in evaluation in phase II clinical trials in multiple advanced cancers. This compound was identified as a potential new anti-cancer agent in a luciferase reporter screening for small molecule p53-independent inducers of TRAIL transcription in BAX-null HCT116 human colorectal cancer cells [328,329]. The initial preclinical studies with ONC 201 have supported a selective cytotoxic effect of this compound against tumor cells [328,329]. The study of the mechanism of action of ONC 201 indicated that this compound inactivates AKT and ERK at late time points post-treatment and consequent Fox 3a activation that precede downstream TRAIL production and induces a gene signature ATF4-related. ATF4 upregulation promotes apoptosis by upregulating the expression of two pro-apoptotic proteins, CHOP and TRAIL-R2 and the expression of both these proteins exhibited a time-dependent increase following ONC1 administration [330,331]. ONC1 induced cell death of leukemic cells through an integrated stress response involving the transcription factor ATF4, the transactivator CHOP, and the TRAIL receptor TRAIL-R2 [330,331]. The activation of ATF4 in response to ONC1 requires the kinases HRI and PKR, which phosphorylate and activate the translation initiation factor eIF2α [330,331]. Other studies have shown that ONC 201 antagonizes the dopamine receptor D2-like subfamily of G protein-coupled R receptors [332]. The anti-tumor efficacy of ONC 201 was demonstrated in a large number of cancers, including patient samples that are refractory to standard or to targeted therapies. Thus, ONC 201 is being to be evaluated in phase I/II clinical studies in several advanced types of cancers [333]. A large in vitro screening on more than 1,000 cancer cell lines showed that ONC 201 is maximally active against lymphoma and myeloma [333]. Furthermore, ONC 201 strongly induces apoptosis in AML cell lines and in primary AML patient samples; ONC 201 was also active against in vivo models, reducing the engrafting of AML blasts into NOD/SCID mice and prolonging their survival [329]. Interestingly, ONC 201 induced the apoptosis also of the leukemic cell population with progenitor/stem cell features (CD34^+^/CD38^−^/ CD45^dim+^) [329].

In a phase I clinical study, patients with various types of solid tumors were treated every three weeks with increasing drug doses, showing that drug administration was well tolerated and showed promising clinical activity [334]. The recommended dose for phase II studies of ONC 201 was determined to be 625 mg given orally every three weeks to patients with advanced cancer [334].

Prabhu et al. have performed an extensive screening of the anti-tumor activity of ONC 201 in a large set of preclinical models of hematological malignancies. ONC 201 demonstrated dose- and time-dependent efficacy in AML, ALL, CML, CLL, diffuse large B-cell lymphoma and multiple myeloma [335]. INC 201 induced caspase-dependent apoptosis that involved activation of the ATF4/CHOP pathway, inhibition of AKT phosphorylation, Foxo3a activation, downregulation of Cyclin D1, IAP and BCL-2 family members [335]. Interestingly, ONC 201 displayed synergistic combinatorial activity with other anti-leukemic drugs and, particularly, synergized with cytarabine and 5-Azacytidine in reducing the viability of AML blasts [335]. 

A recent study provided evidence that dose-intensification of ONC 201 may improve anti-tumor efficacy of this drug [336]. Thus, in various experimental models, it was provided evidence that with dose-intensification, ONC 201 promoted a potent anti-metastasis effect and inhibition of cancer cell migration and invasion and promoted accumulation of activated NK and CD3^+^ lymphocytes within treated tumors [336]. According to these findings, a change in ONC 201 dosing in all open clinical trials was adopted [336].

## 8. Autophagy Pathway: Abnormalities a Therapeutic Opportunities in AML

Autophagy is a lysosomal pathway of degradation of intracellular materials, such as damaged organelles. It appears to be an adaptive survival mechanism triggered by conditions of cellular stress, such as nutrient deprivation or caloric restriction. Various phases in the autophagy process have been identified: a first step consists in the autophagy initiation and formation of the phagophore. Signals of the nutrient status are sensed by ULK1 (like autophagy activating kinase 1) initiation complex, which then activates the autophagic process, recruiting a second complex, known as VPS34 complex, determining the formation of a unique flat membrane known as phagophore; phagophores then elongate, resulting in autophagosomal maturation; finally, the last step is represented by the fusion of autophagosomes with lysosomes, with demolition of phagosome content by lysosomal proteases [337,338]. Recent studies have shown that autophagy plays an important role in the control of normal and leukemic hematopoiesis [337,338].

Autophagy seems to be a mechanism essential for the homeostasis of the normal hematopoietic system, acting particularly at the level of the hematopoietic stem cell compartment. Autophagy regulates mitochondrial homeostasis of HSCs, ROS production and cell differentiation [339]. In fact, deletion of *ATG7*(*autophagy-related protein 7*), an essential constituent of the autophagy machinery resulted in the loss of HSC main biologic functions and development of severe myeloproliferation [339]. Importantly, ATG7 was shown to interact with p53 and can modulate p53 checkpoint in cell cycle exit in response to metabolic stress [340]. 

Another autophagy-related gene, *ATG12* (*autophagy-related protein 12*), was found to be an important modulator of HSC self-renewal: mice with hematopoietic-specific deletion of ATG12 displayed increased numbers of circulating blood elements, but exhibited a markedly reduced regenerative capacity of these HSCs in transplantation, due to severe engraftment impairment [341]. Finally, FIP 200 (200 kDa family kinase-interacting protein), a constituent of the ULK1 (unc-51 like autophagy activating kinase) autophagic initiation complex is required for HSC homeostasis since its deletion resulted in increased mitochondrial mass and proliferation of HSCs [342].

Mitophagy, a peculiar mechanism of autophagy, operates at the level of HSCs and plays a fundamental role in maintaining a low mitochondrial activity in this cell compartment. A key regulator of mitophagy is PINK1 (PTEN-induced putative kinase 1) that interacts with the outer mitochondrial membrane and Parkin, a E3 ubiquitin ligase, driving mitochondria to autophagosomal degradation. A major role in the process of autophagy is played by the ATAD3A protein: this protein acts as a molecular bridge between the translocase of the outer membrane complex and the translocase of the inner membrane complex, thus facilitating the import of PINK1 into mitochondria. Deletion of ATADA3A determines enhanced mitophagy, with mitochondria depletion and blockage in cell differentiation [343].

The role of autophagy in leukemia development is controversial in that both leukemia-promoting and leukemia-inhibitory roles have been suggested. Watson and coworkers have explored the role of autophagy in AML development using leukemia animal models and exploring expression of ATGs in primary AML cells [283]. Monoallelic loss of ATG7, a key autophagic gene, in a mouse AML model elicited an increase of leukemic cell proliferation under conditions of metabolic stress in vitro, dependent upon enhanced glycolytic activity and leukemia progression in vivo [344]. The study of primary AML blasts showed the reduced expression of autophagy genes and decreased autophagic flux, with accumulation of damaged mitochondria [344]. Multiple key autophagic genes are comprised within regions which are commonly heterozygously deleted in AML. Among these genes, ATG12, which is involved in phagosome maturation, is frequently decreased in AML patients [344]. 

At variance with this study, Folkerts and coworkers reported variable levels of autophagy flux in primary AML blast with higher levels among AMLs classified as poor risk AMLs, compared with AMLs with an intermediate/favorable profile risk [345]. CD34^+^ AML cells were more sensitive than normal CD34^+^ cells to autophagy inhibitors: however, only AMLs with *TP53^WT^*, but not those with *TP53^mut^* were inhibited by the autophagy inhibitor hydroxychloroquine [345]. Inhibition of the autophagic machinery reduced the in vivo survival of AML CD34^+^ cells in immunodeficient mice [345].

An analysis of the whole exome sequencing of 223 cases of myeloid neoplasms, including AMLs, showed that in 14% of these patients, missense mutations or copy number alterations of at least 1 gene involved in autophagy was observed [346]. Marconi and coworkers have explored abnormalities of autophagy machinery in AML patients and found that autophagy alteration (*ULK1 CHR11, ULK1 CHR17, BECN1, ATG14, AMBRA1, UVRAG, ATG14, ATG9A, ATG9B, PIK3C3, PIK3R4*) was associated with poor prognosis and with complex karyotype and TP53 mutation [347].

Other observations have shown that the antileukemic drug cytarabine, in spite of its antileukemic effects, promotes also mTOR-dependent cytoprotective effects in primary AML blasts and that blocking autophagy by treatment with bacilomycin A1 or the lysosomotropic agent chloroquine elicited a marked increase of the cytotoxic effects of cytarabine on leukemic blasts [348]. Concomitant silencing of *ATG7* in both AML and mesenchymal stromal cells elicited a more pronounced sensitization to the cytotoxic effects of cytarabine than depletion of ATG7 only in AML cells [349]. Supported by these observations, a clinical trial (NCT 02631252) is evaluating a chloroquine derivative, hydroxychloroquine, in association with anti-leukemic chemotherapy (mitoxantrone + etoposide) in relapsing/refractory AML patients.

Other studies have shown the existence of a link between specific molecular abnormalities and autophagy abnormalities. Thus, it was shown that *FLT3-ITD* mutations induce an increase in basal autophagy in leukemic cells, through a signaling cascade involving the ATF4 transcription factor [350]. Inhibition of autophagy or *ATF4* expression in *FLT3-ITD*-mutated AMLs induced cell proliferation in vitro and tumor burden in murine xenograft models [350]. Importantly, autophagy inhibition was able to overcome FLT3 inhibition resistance related to FLT3-TKD mutation [350]. Another study showed a link between FLT3 mutations and mitophagy. In fact, Rudat and coworkers observed the frequent activation of the RET receptor tyrosine kinase in several AML subtypes [351]. Particularly, it was shown that key RET effectors are represented by mTORC1-mediated suppression of autophagy and subsequent stabilization of leukemogenetic drivers, such as mutant FLT3 [351]. Genetic or pharmacologic RET inhibition impaired the growth of FLT3-dependent AML cells and was accompanied by upregulation of autophagy and FLT3 depletion [351].

Differently from canonical autophagy, mitophagy selectively targets mitochondria which have been marked by mitophagy receptors; these receptors include PINK1 receptor system, microtubule-associated proteins 1 light chain 3 (LC3)-interacting region (LIR)-containing receptor system and lipid-mediated system (ceramide) [352]. Several observations suggest that abnormalities of mitophagy could play a relevant role in some AMLs. 

One of the few studies addressing the role of mitophagy in leukemia was a recent study by Pei and coworkers showing that primitive AML leukemic stem cells possess a high expression of the mitochondrial dynamic regulator FIS1 and a unique mitochondrial morphology; FIS1 loss attenuates mitophagy and impairs AML LSC potential [178]. Using valinomycin as an agent inducing mitochondrial stressing, it was shown that AML leukemic stem cells overexpressing FIS1 have higher levels of mitophagy than non-leukemic stem cells [178]. FIS1 deletion in AML-leukemic stem cells impairs mitophagy, and induces several secondary effects consisting in induction of myeloid differentiation, block in cell cycle and reduced self-renewal activity [178]. Furthermore, these studies showed that the central metabolic stress regulator AMPK is activated in leukemic stem cells and cats as an inducer of FIS1: AMPK inhibition recapitulates the biologic effects induced by FIS1 loss [178]. These observations support the view that the high activity of AMPK/FIS1-mediated mitophagy is an efficacious way to maintain stem cell properties that could be otherwise compromised by the numerous stresses induced by oncogenic transformation. Other studies have shown that FIS1 is overexpressed in a part of AMLs, particularly in refractory AMLs, and its overexpression is associated with a reduced rate of complete remissions to induction therapy [353]. 

A recent study of experimental leukemogenesis supports the role of mitophagy in leukemia progression. Thus, Nguyen and coworkers have explored the leukemogenetic role of the autophagy receptor p62, a protein bridging the ubiquitinated cargos into autophagosomes, showing that: this protein is overexpressed in some AMLs, in association with a poor prognosis; loss of p62 impaired expansion and colony-forming capacity of AML cells [354]. A cellular analysis showed that loss of p62 impaired the removal of damaged mitochondria, increased mitochondrial superoxide production and impaired mitochondrial respiration [354]. These observations support an essential role for p62 in mitophagy in leukemic cells.

The study of AMLs bearing FLT3-ITD mutations showed that FLT3 signaling suppresses generation of ceramide, one of the mediators of mitophagy [355]. Molecular or pharmacologic inhibition of FLT3-ITD reactivated ceramide synthesis, inducing ceramide accumulation on the outer mitochondrial membrane, which then bound autophagy-inducing light chain (LC3), to recruit autophagosomes for the execution of lethal mitophagy [355]. Interestingly, FLT3-inhibitor resistance is attenuated by LCL-461, a mitochondria-targeted ceramide analog drug, which induces lethal mitophagy in human AML blasts with FLT3-ITD mutations [355].

## 9. TP53-Mutated AMLs

Tumor suppressor TP53 is the most frequently mutated gene in human cancer (about 50% of cases). TP53 is activated by various cellular stresses, such as DNA damage and oncogene activation, and, in turn, regulates a wide number of cellular processes, mediated through an anti-proliferative transcriptional program [356]. Thus, TP53 is a key regulator of many fundamental cellular processes, such as cell proliferation, senescence and apoptosis; however, recent studies indicate a key role of TP53 also in cell metabolism, stem cell division and cell death by ferroptosis [356].

TP53 plays a key role in the control of apoptotic processes. In response to irreparable DNA damage, TP53 is activated and triggers the induction of an apoptotic response. A cytosolic and a mitochondrial TP53 apoptotic pathways have been identified [296]. In the cytosolic p53 apoptotic pathway, nuclear TP53 induces PUMA and NOXA expression, which in turn promote the release of cytosolic TP53, maintained in an inactive state in the cytoplasm through binding with BCL-XL. Activated TP53 induces BAX oligomerization and mitochondrial translocation. Cytosolic TP53 is the main source of mitochondrial TP53. At the level of mitochondria, TP53 induces BAX and BAK oligomerization, antagonizes the anti-apoptotic activity of BCL-2 and BCL-XL, and forms a molecular complex with cyclophilin D in the mitochondrial inner membrane. These changes determine mitochondrial outer membrane permeabilization, which results in the release of cytochrome c and other pro-apoptotic proteins from the intermembrane space to cytosol and induction of apoptosis [357]. TP53 activation may stimulate an apoptotic response also through another mechanism, involving upmodulation of death receptor. Such as Fas or TRAIL-R2, expression [358]. Recent studies have implicated TP53 also in the regulation of autophagy through the AMPK/TCS2-mTOR signaling pathway; particularly, two TP53-inducible signaling molecules, DRAM (damage-regulated autophagy modulator) and 14ARF are involved in the autophagy pathway and may activate also apoptosis [359]. 

The effects of P53 on the autophagic process are complex and depend on the cellular localization of this protein: thus, nuclear TP53 acts as a pro-autophagic factor, whereas cytoplasmic p53 inhibits the induction of autophagy [360]. TP53 activates autophagy through a negative regulation of the mTOR pathway: in fact, several TP53 target genes, such as *PTEN*, *TSC2* and *AMPK*, induced through sestrin activation, lead to the inhibition of mTORC1 [361]. At variance with normal TP53, mutant *TP53* enhances cancer cell survival under oxidative and genotoxic stress conditions [362]. Thus, mutant *TP53* may directly block Caspase-9 activity [363], as well as Caspase-9-mediated activation of mitochondrial Caspase-3 [364].

In a large screening carried out in 858 primary AMLs it was estimated a frequency of TP53 amounting to 13% (13% total; 5% mutated+deleted; 7% mutated-only; 1% deleted-only) [304]. The presence of *TP53* mutations in AML strongly correlates with the presence of complex karyotype, observed in 95% of these patients [365]. The incidence of TP53 mutations in AML patients is strongly influenced by patient’s age, being much more frequent among old than young patients [365]. The presence of TP53 mutations negatively impacted AML survival, representing one of the AML subtypes associated with the most negative prognosis [365]. TP53 mutations are particularly frequent in AML patients corresponding to the M6 FAB subtype, acute erythroid leukemia [366,367]. *TP53* mutations are stable during disease progression and are considered as an event occurring within founding clone of the leukemic process [368]. The analysis of co-occurring mutations has strongly supported the view that TP53-mutant AMLs makes part of a unique, distinct subgroup [369]. *TP53*-mutated AMLs are characterized by a paucity of other common single-nucleotide mutants (particularly *NPM1, FLT3, DNMT3A, IDH1, IDH2* and *TET2*); in contrast, *TP53*-mutated AMLs are associated with recurrent co-occurring karyotypic structural alterations, including abnormalities of chromosomes 5, 7 and 17 and with events involving chromothripsis ( a one-step genome shattering catastrophic event resulting from disruption of one or few chromosomes into multiple fragments and consequent random rejoining and repair) [370,371].

Surprisingly, *TP53*, as well as *DNMT3A* mutations, are conspicuously absent from virtually all pediatric cases [41]. Usually, TP53 protein overexpression in AML blasts correlates with the presence of *TP53* mutations and with the presence of complex karyotype abnormalities. A recent study explored TP53 expression in AML blasts with normal karyotype showing that 24% of these leukemias display TP53 protein overexpression, associated with absent CD34 expression on leukemic blasts and presence of *FLT3-ITD* mutations [311]. The presence of TP53 overexpression predicted for shorter leukemia-free survival in patients who underwent allogeneic stem cell transplantation [372]. Using bioinformatic approaches, Huang and coworkers have tried to identify the key pathways and genes associated with *TP53* mutations [373]. The analysis of differentially expressed genes showed that these genes are enriched in pathways associated with cell adhesion biological processes, Ras-associated protein 1, PI3K-AKT pathway and cell adhesion molecules [373]. These pathways could play a key role in *TP53* mutation AMLs and could help to identify potential therapeutic targets [373].

*TP53* mutations are particularly frequent in therapy-related AMLs (tAMLs). tAMLs usually develop 1-5 years after exposure to chemotherapy or radiotherapy; these AMLs are distinguished from de novo A>MLs for a higher incidence of *TP53* mutations, abnormalities of chromosomes 5 or 7, complex cytogenetics and a reduced response to chemotherapy [374]. The mechanisms by which TP53 mutations are selectively enriched in tAML are mainly related to the presence in these patients of pre-existing HSCs/HPCs carrying *TP53* mutations, resistant to chemotherapy and expanding preferentially after treatment [374]. The early acquisition of *TP53* mutations in the founding clone contributes to the progressive acquisition of cytogenetic abnormalities and to the poor response to chemotherapy [374]. 

As above discussed, *PPM1D* mutations are also considerably enriched in tAMLs. In a recent report, Hsu and coworkers observed *PPM1D* mutations in about 20% of AMLs (compared to a frequency of <1% in de novo AMLs) [13]. Only *TP53* mutations (28% of tAMLs) are more frequent than *PPM1D* mutations in tAMLs [13]. *PPM1D* mutations in tAMLs do not co-occur with *TP53* mutations and, at variance with *TP53* mutations, are not associated with karyotype abnormalities (complex cytogenetics or deletions in chromosomes 5 opr 7) [13]. PPM1D is a serine/threonine phosphatase that is transcriptionally upregulated by TP53 in response to DNA damaging agents; PPM1D in turn negatively regulates TP53 and several proteins involved in the DNA damage response (DDR). Truncating *PPM1D* mutations occurring in AML inhibit DDR and provides a selective advantage to mutant-PPM1D hematopoietic cells in the presence of chemotherapy [375]. Small-molecule inhibigtors of PPM1D activity reverse a normal DDR sensitivity and may represent a tool to prevent expansion of hematopoiesis induced mu mutant *PPM1D* [375].

As above mentioned, *TP53* mutations in AML are associated with clearly lower responses to standard chemotherapy and are considered as a strongly negative prognostic marker associated with poor treatment outcomes, including with allogeneic stem cell transplantation [376].

*TP53*-mutated AMLs have a very negative outcome. The treatment of these leukemias should adopt the use of drugs that do not involve TP53 activation in their mechanism of anti-leukemic activity. Recent studies have provided the hope for an improvement of the therapeutic response of this AML subgroup. These clinical studies were based on the use of drugs, such as decitabine, that do not cause direct cytotoxicity, but alter the epigenetic response of leukemic cells. The use of this drug in the treatment of *TP53*-mutated AMLs was supported by the observation that the presence of adverse-risk karyotypes does not seem to affect the clinical response to decitabine. Thus, Welch and coworkers have shown that 10-day cycles of decitabine induced clinical responses in 21 of 21 patients with *TP53* mutations, and these patients had equivalent outcomes compared with *TP53* non-mutated AMLs: in both AML groups, the median overall survival was around 500 days [377]. More recently, using a 5-day schedule of decitabine administration it was observed a 62% response rate among TP53-mutated AML patients [378]. More recently, it was evaluated a new therapeutic regimen based on cladribine (a purine nucleoside inducing DNA hypomethylation by inhibition of S-adenosylhomocysteine hydrolase) and low-dose cytarabine alternating with decitabine as front-line therapy for elderly patients with AML [379]. This regimen was shown to be effective, with a median overall survival of 13.8 months; 40% of *TP53*-mutated AMLs displayed an objective response to this treatment [316]. It is important to note that this treatment did not lead to an extinction of the leukemic clones: in fact, responders, including those in complete remission, have persistent tumor burden for at least 1 year without disease progression [380].

Recently, it was explored also another drug combination based on vosaroxin (an anti-cancer quinolone-derived DNA topoisomerase II inhibitor) and decitabine administration to older patients with AML. At optimal doses of vosaroxin (70 mg/m^2^) a median overall survival of 14.6 months was observed [318]. The response of *TP53*-mutated AMLs remained poor and apparently not improved by this treatment regimen [381]. 

The conclusion of these studies was that hypomethylating agents may induce clinical responses in TP53-mutated AMLs. Randomized studies will be required to compare a day-5 regimen versus a day-10 regimen of decitabine in these patients [382]. A recent randomized study compared a 5-day versus 10-day schedule of decitabine in older patients with newly diagnosed AML [383]. However, the results of this study showed that 10-day schedule does not lead to better outcomes than the usual 5-day schedule [381]. Interestingly, in clinical trials involving the use of hypomethylating agents in combination with venetoclax, 20% of clinical responses were observed among *TP53*-mutated AML patients [180]. It is of interest to note that responding patients had concurrent RUNX1 mutations; similarly, patients with adverse cytogenetics responding to venetoclax, all have concurrent RUNX1 mutations [180].

Very interesting are the first clinical results obtained using pevonedistat (PEV), a small-molecule inhibitor of the NEDD8-activating enzyme (NAE), which processes NEDD8 for binding to target substrates. PEV administration determines reduced NAE activity, leading to Cullin-RING E3 ubiquitin ligase substrate accumulation, with consequent inhibition of cell proliferation [384]. Initial studies have shown anti-leukemic activity; in a recent study, PEV was administered in combination with azacitidine for the therapy of naïve AML patients with ≥60 years who cannot be treated with standard induction therapy [384]. Interestingly, in TP53-mutated AMLs the CR/PR rate was 80% [384]; two (of a total of five) of these patients stayed on study for >10 cycles [384].

In spite these progresses, the outcome of *TP53*-mutated AMLs remains very poor. In an analysis carried out by Alliance for Clinical Trials in Oncology no complete responses were observed among *TP53*-mutated AMLs, while some complete responses were observed among high-risk AML patients with non *TP53*-mutated AMLs [65]. Patients with complex atypical karyotype, less frequently associated with TP53 mutations have higher complete remission rates and longer overall survival than those with complex typical karyotype, frequently associated with TP53 mutations [65]. 

TP53 mutations occur in less than 10% of de novo AMLs; however, other non-mutational TP53 abnormalities are more frequently observed in AMLs and strongly support a greater significance of TP53-pathway inactivation in the biology of these leukemias [323]. TP53 mutations in AMLs result in loss of trans-activating function and stabilization of the mutant TP53 protein due to the repression of TP53-inducible proteins, such as MDM2 that target TP53 protein for degradation. However, TP53 stabilization can be found also in the absence of TP53 mutant alleles. In normal cells, TP53 protein is maintained at low levels by several regulators including MDM2, which acts as a TP53 ubiquitin ligase to facilitate TP53 protein degradation. Some cellular stresses, such as DNA damage, promote TP53 phosphorylation, blocking MDM2-mediated degradation [324]. In contrast, oncogenic mechanisms may induce the ARF tumor suppressor to inhibit MDM2 [385]. Thus, TP53 function in AMLs can be suppressed not only by mutations, but also via overexpression of negative regulators, such as MDM2 and/or its homolog MDM4 or through p14^ARF^ inactivation.

A recent study based on the analysis of 511 de novo AMLs provided strong evidence that the TP53 pathway dysfunction is highly prevalent in AML independent of TP53 mutational status [386]. In fact, proteomics, mutational profiling and network analyses showed that: (a) TP53 stabilization is a constant feature of mutant *TP53* samples and it is frequent (>15% of cases) in AML samples wild-type *TP53* and both these conditions are associated with a poor prognosis; (b) the *TP53* negative regulator MDM2 is frequently (35.8% of cases) overexpressed in AML samples retaining wild-type *TP53* alleles, associated with absence of p21 expression and negative prognosis; (c) *MDM4* is overexpressed in 38.5% of AMLs and its presence is associated with a less negative prognosis; (d) AML samples display unique patterns of TP53 pathway protein expression, allowing the segregation into various cohorts with different prognostic impact and with some potential vulnerabilities exploitable at therapeutic level [386].

The discovery of non-mutational *TP53* dysfunction aims to rescue WT-*TP53* inactivation, acting at the level of a specific TP53-inactivating mechanism [385]. Thus, a number of MDM2 inhibitors have been developed, under investigation in monotherapy and in combination with molecularly targeted agents, such as venetoclax, decitabine and trametinib [385].

## 10. Conclusions

The approval of venetoclax for the treatment of chronic lymphocytic leukemia and the submission of a request of approval to FDA for venetoclax in association with low-dose cytarabine or with the hypomethylating agent dDecitabine for the treatment of AML patients who are ineligible for the treatment with standard chemotherapy represents a new era in targeting the apoptotic pathway for leukemia therapy. Very recently (end of November 2018), the FDA has granted an accelerated approval to venetoclax for use in combination with azacitidine or decitabine or low-dose cytarabine for the treatment of newly-diagnosed adult patients with AML who are aged 75 years or older, or who have comorbidities that preclude use of intensive induction chemotherapy. 

Other studies recently presented at the last ASH Meeting (December 2018) strongly support the great potentialities of venetoclax in AML therapy. Wei et al reported the results of an ongoing phase Ib clinical study performed in elderly AML patients treated with venetoclax plus a modified intensive chemotherapy schedule. The results of the first 44 treated patients, including de novo treated sAML patients, patients with secondary AMLs and patients pretreated with hypomethylating agents, showed an overall CR rate of 71% (95% among patients with de novo AMLs); response rates were 88% among intermediate risk cytogenetic AMLs and 46% in adverse risk AMLs [387]. The analysis of the clinical response in function of the mutations showed 100% of response in *NPM1*-mutated AMLs, 90% in *RUNX1*-mutated AMLs, 90% in *RAS*-mutated AMLs, 89% in IDH-mutated AMLs and 33% for *TP53*-mutated AMLs [387]. The overall response rate for untreated patients was 84%, but 43% for patients previously treated with hypomethylating agents [325]. In another ongoing study, Wei and coworkers have reported the results of an ongoing phase I/II study involving the administration of venetoclax, together with low cytarabine, to 82 elderly AML patients [387]. 54% of these patients achieved CR: 35% and 71% in patients with secondary and de novo AMLs. The responses consistently changed in function of the main genetic features of treated AMLs, with the following rates of CRs: NPM1 89%, IDH 1–2 72%, FLT3 44%, TP53 30% [388]. These observations show that venetoclax, in combination with low-dose citarabine is an effective therapeutic option for AML patients who are not suitable for standard induction therapy [388]. Another study explored the potential mechanisms of resistance of AML blasts to venetoclax and azacytidine. In fact, although about 80% of AML patients are responding to this combination therapy, a significant proportion of them experience disease progression. The analysis of the potential mechanisms of drug resistance was based on two observations already available on the mechanism of the anti-leukemic effects of venetoclax: the mutational background of AML blasts; the targeting of the energy metabolism. The analysis of these two key parameters led to the conclusion that the presence of mutations of *PTPN11*, a tyrosine phosphatase SHP2, present in 4–6% of AMLs represents a major mechanism of resistance to venetoclax. These AML patients displayed frequent *TET2* (52%), *NPM1* (31%) and *FLT3-ITD* (26%) mutations and in about 52% of cases displayed cytogenetic abnormalities (complex in 19%, chromosome 3 in 14% and chromosome 5 or 7 in 20% of cases) [389]. The analysis of the energetic profile of AML cells showed that resistance of AML patients to with *PTPN11* mutations is related to compensatory mechanisms that restore sufficient levels of OXPHOS, related to the presence in these leukemic cells of elevated MCL-1 levels [390]. In AML patients treated with venetoclax+azacytidine the presence of PTPN11 predicted shorter response duration [328]. In vitro studies showed that *PTPN11*-mutant AMLs are particularly sensitive to MCL-1 inhibitors [390].

Thus, targeting BCL-2 has emerged as an efficacious and tolerated clinical strategy for AML patients. BCL-2 inhibitors, as well as other targeted agents need to be administered together with standard anti-leukemic drugs. Furthermore, these inhibitors can be combined with other targeted agents, with activity in some AML subgroups and preclinical studies support the rationale of these drug associations. Among these various drug associations, particularly promising could be the association of BCL-2 inhibitors with MCL-1 inhibitors, given the biological function of BCL-2 and MCL-1 to act as resistance factors for inhibition of the other. Future studies will be required to assess the safety and clinical efficacy of this strongly rational drug association. Although these drug combinations seem to be very promising, they can raise some relevant safety issues, that could be bypassed though selective targeting of tumor cells with these inhibitors. Another promising drug combination is represented by BCL-2 inhibitors and HDAC inhibitors, showing significant anti-leukemic activity in in vitro and in vivo models of high-risk and *TP53*-mutated AMLs [391]. 

As above mentioned, IDH-targeting inhibitors exhibited promising activity as anti-leukemic drugs and received approval for use as anti-leukemic drugs in refractory-relapsing IDH-mutant AMLs. In these studies, the IDH1, ivosidenib, and IDH2, enasidenib, inhibitors were evaluated as single agents. A recent study reported the initial results of a phase I trial involving the administration of ivosidenib or enasidenib, in combination with standard induction and consolidation chemotherapy, to the de novo diagnosed AML patients bearing either *IDH1* or *IDH2* mutations [392]. The initial results based on the analysis of 47 AML patients treated with ivosidenib and 87 AML patients treated with enasidenib showed 93% and 73% of complete responses among primary AML patients [392]. A significant proportion of responding patients displayed a negative measurable disease [392]. These results strongly support the evaluation of IDH inhibitors in combination with standard induction/consolidation therapy in the context of a randomized phase III study. An ongoing clinical trial (NCT02719574) is evaluating FT-2102, a mutant IDH1 inhibitor in combination with azacytidine in AML patients previously pretreated with standard antileukemic therapies. Preliminary results of the first 27 treated patients showed an overall responding rate of 42%, with some durable complete responses [393]. These observations further support the evaluation of this drug combination.

As above mentioned, in a phase II clinical trial of venetoclax monotherapy, *IDH*-mutated relapsed/refractory AML patients had a response rate of 33% compared to 10% of *IDH*-wild-type AML patients [394]. Based on these findings it was hypothesized that combination therapy based on enasidenib and venetoclax may exhibit superior anti-leukemic activity compared to single agents in the treatment of *IDH2*-mutant AMLs. A preclinical study based on xenograft models of IDH2-mutated human AMLs supported this hypothesis, showing enhanced anti-leukemic activity of the drug combination, compared to monotherapy [394]. These observations support future clinical investigations of this drug combination.

The group of AMLs bearing *IDH1* or *IDH2* mutations is heterogeneous and a better definition of biologically and clinically relevant subgroups among these leukemias is very important to better define the spectrum of sensitivity to IDH-targeted therapies. In this context, particularly relevant was a recent study based on the analysis of 3898 newly diagnosed AML patients, including 329 patients with IDH1 mutations and 423 with *IDH2* mutations [395]. The most relevant finding of this study was that the R132C variant of *IDH1* mutation was associated with increased patient age, lower WBC (white blood cell) and lower *NPM1* and/or *FLT3* co-mutation rate: these patients displayed a lower CR rates and shorter OS [395]. Furthermore, the IDH2 R172K mutation, mutually exclusive with *NPM1* and/or *FLT3-ITD* mutations displayed better overall survival than patients with *IDH2 R140Q* [395]. Another study showed that AMLs with mutations in *IDH1* and *DNMT3A* display a distinct epigenetic signature, with poorer overall survival, while *IDH2/DNMT3A* mutated AMLs conferred similar mortality risk compared to DNMT3A alone [396]. Thota and coworkers have analyzed the clinical outcomes for 425 AML patients with a diagnosis of *IDH1/2* mutant AML [397]. IDH1 and IDH2 mutant AMLs have a similar response to intensive chemotherapy; non-responders to intensive chemotherapy were enriched for *RUNX1, SRSF2* and *ASXL1* mutations [397]. Finally, another recent study explored the mechanisms of response and resistance to IDH inhibitors. This study showed that IDH inhibitors induce differentiation and differentiation syndrome through upregulation of PU.1 and C/EBP target genes, associated with demethylation of their binding motifs [398].

The important developments related to the use of BCL-2 and IDH inhibitors in leukemia treatment could represent only the first step in the introduction into cancer therapeutics of drugs targeting the apoptotic machinery, with future developments possibly involving other members of the BCL-2 family and members of the TRAIL-family. Mitochondria are judges and executioners of the cell death sentences to tumor cells [399]. Frequently, AML cells display mitochondrial abnormalities, related either to mutations of some mitochondrial genes, such as IDH or electron transport chain genes, or to abnormalities of the mitochondrial oxidative metabolism, with a consequent reprogrammation of the oxidative phosphorylation pathway to meet the challenges of high energy demand. Often leukemic cells can acquire a hybrid glycolysis/OXPHOS phenotype in which both metabolic pathways can be utilized for energy production and biomass synthesis.

This approach is particularly important in view of the growing evidence that there is an absolute need to better characterize relapsing AML cells. Recent studies have shown that therapy-resistant cells are already present at diagnosis and in some patients, relapse originates from rare leukemic stem cells, while in other patients, relapse originates from larger leukemic subclones of phenotypically less immature cells, retaining transcriptional stemness signatures [45]. Accordingly, stemness targeting would represent an important strategy to prevent leukemic relapse. These conclusions are supported laso through the study of patients developing AML from lympho-myeloid clonal hematopoiesis. In the large majority (>80%) of AML patients with lympho-myeloid clonal hematopoiesis, the preleukemic clone was refractory to chemotherapy and was the founding clone for relapse [400]. Long-term survival of these patients was achievable only after allogeneic stem cell transplantation [400]. Other current studies have shown that leukemia-regenerating cells (LRCs) arise post-chemotherapy and are distinct from LSCs identified at diagnosis: LRCs emerge following chemotherapy treatment and are characterized by a peculiar pattern of gene expression, with high expression of dopamine receptor D2 (DRD2) and human serotonin receptor 1B (HTR1B) and of angiopoietin-1 [401]. In experimental models targeting of LRCs suppresses accelerated re-growth of relapsing leukemic cells [401]. In line with these observations, inhibition of HTR1 elicited an impairment of LSC functionality, further supporting the rationale of targeting this receptor for anti-leukemia therapy [402]. Furthermore, the initial results of a phase I trial in 13 relapsing/refractory AML patients using thioridazine, a DRD2 antagonist, showed signs of anti-leukemic activity [403]. Importantly, another recent study provided evidence that chemotherapy-resistant leukemic stem cells present after induction chemotherapy are characterized by a high phosphorylation status: targeting OXPHOS markedly enhanced AraC sensitivity of these cells, thus indicating that oxidative mitochondrial metabolism may represent an important target to try to eradicate these chemoresistant leukemic stem cells [188].

In conclusion, studies carried out in these last years strongly support the view that targeting defective apoptotic cellular death and deregulated metabolism represent new key strategies for the treatment of leukemias, including AMLs. Future studies need to be based on the rational design of drug combinations and identification of predictive biomarkers to identify leukemic patients most likely to benefit from therapies containing BH3-mimetic drugs or other drugs inducing apopotosis or metabolic inhibitors, taking into due account AML subtype, oncogenic drivers and AML clonal heterogeneity.

## Figures and Tables

**Figure 1 cancers-11-00260-f001:**
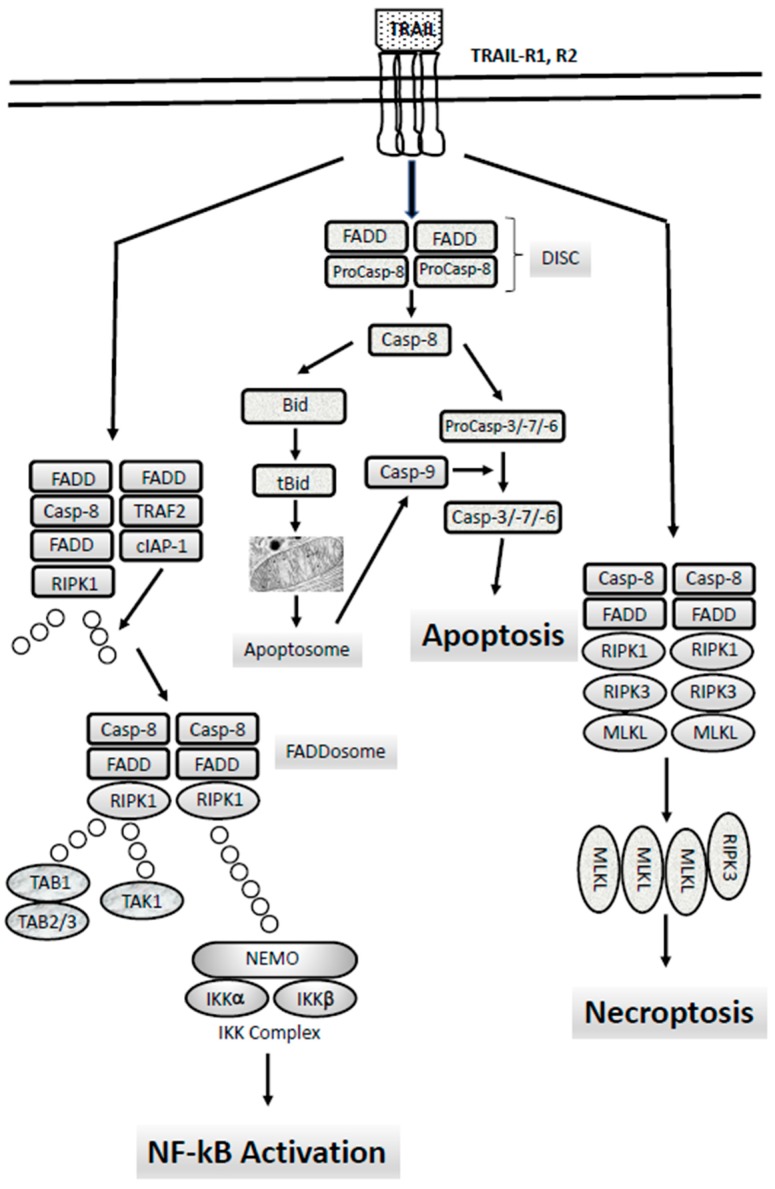
TNF-related apoptosis-inducing ligand (TRAIL)-induced signaling pathways. The interaction between TRAIL and its membrane receptors TRAIL-R1 and TRAIL-R2 may activate three different signaling pathways: non-death, leading to nuclear facotr κ-light-chain-enhancer of activated B cells (NFκB) activation; death leading either to apoptosis or to necroptosis. In the absence of inhibitory signals, the apoptotic signaling pathway is activated and triggers apoptosis. In the apoptotic signaling pathway, binding of TRAIL to its membrane receptors TRAIL-R1 or TRAIL-R2 induces the recruitment of FADD (fas-associated protein with death domain) and Caspase-8 at the level of the cytoplasmic tail of these receptors, forming DISC (death-inducing signaling complex) or Complex 1. In this complex pro-Caspase-8 is activated to Caspase-8, determining the release of active Caspase-8 in the cytosol initiating the induction of the Caspase apoptotic cascade. In some cells, the mitochondrial amplification loop is required to induce the activation of Caspase-3, through activation of Bid, mitochondrial activation with release of the apoptosome, containing activated Caspase-9, required for Capsase-3 activation. In some cells, TRAIL stimulation may result in the activation of the NF-kB signaling pathway, with production of inflammatory cytokines. NF-kB activation via TRAIL-R activation is promoted through the formation a FADDosome complex, containing FADD, Caspase-8 and RIPK1 (receptor-interacting serine/threonine protein kinase 1); the ubiquitin linkages created by the c-IAPs assembled to the FADDosome lead to the recruitment of linear ubiquitin chains; modified RIPK1 then induces the recruitment of two protein complexes, the TAB/TAK (TGFβ-activated binding/TGFβ-activated kinase) complex and the IKK (I Kappa B kinase) complexes, with subsequent NF-kB activation and induction of inflammatory cytokines production. Under conditions where Caspase-8 activity is inhibited, the dissociated DISC complex can interact with RIPK1 and RIPK3 to form a molecular complex, which determines RIPK3 and MLKL (mixed lineage kinase domain-like) activation and then necroptosis.

**Figure 2 cancers-11-00260-f002:**
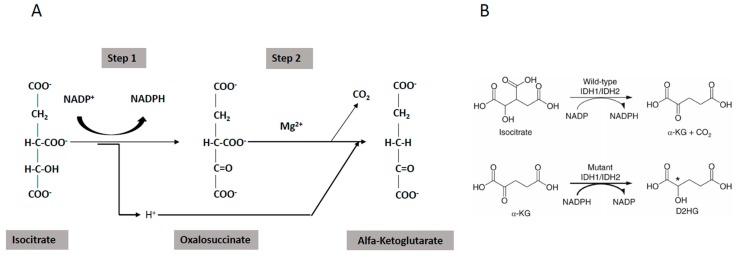
Reactions catalyzed by the enzymes isocitrate dehydrogenase (IDH). (**A**) Two-step conversion of isocitrate to α-ketoglutarate (α-KG) catalyzed by IDHs, through a first step involving conversion of isocitrate to oxalosuccinate and a second step consisting in the conversion of oxalosuccinate to α-KG. (**B**) Top panel: the reaction catalyzed by WT-IDH1/2 enzymes implies the conversion of Isocitrate to α-KG, with concomitant reduction of NADP to NADPH and production of CO_2_. Bottom panel: the reaction catalyzed by mutant IDH1/IDH2 implies the conversion of α-KG to (R)-2-Hydroxyglutarate, with concomitant oxidation of NADPH to NADP.

**Figure 3 cancers-11-00260-f003:**
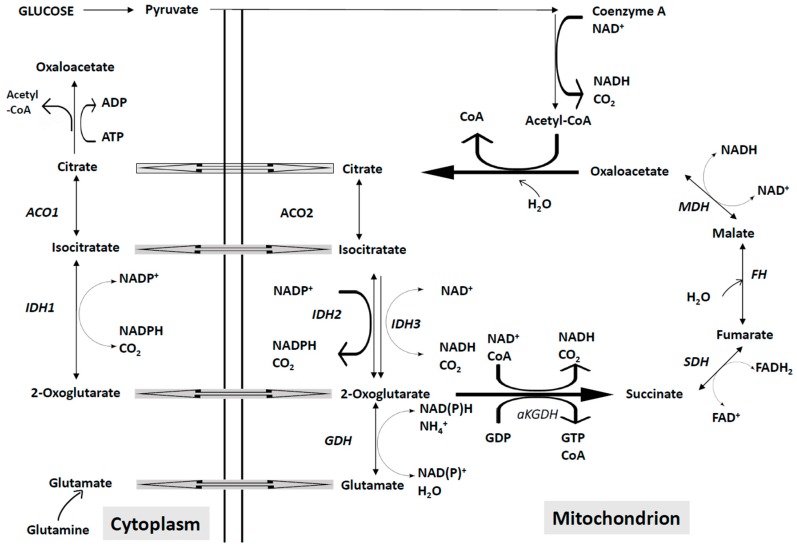
Role of IDH family members in the energetic metabolism. The main mitochondrial and cytoplasmic reactions involving the three IDH enzymes (IDH1, IDH2 and IDH3) are shown; mitochondrial IDH2 and IDH3 partecipate to the TCA cycle. IDH1 and IDH2 catalyze the NADP-dependent reversible conversion of isocitrate to α-KG, either in the cytoplasm (IDH1) or in the mitochondria (IDH2); IDH3 catalyzes the NAD^+^-dependent irreversible conversion of isocitrate to α-KG. The entry of glucose and of glutamine-derived carbon molecules in the TCA cycle is also shown. ACO. Aconitase; GDH: glutaminase; SDH: succinate dehydrogenase; αKGDH: alpha ketoglutarate dehydrogenase; MDH: malate dehydrogenase.

**Figure 4 cancers-11-00260-f004:**
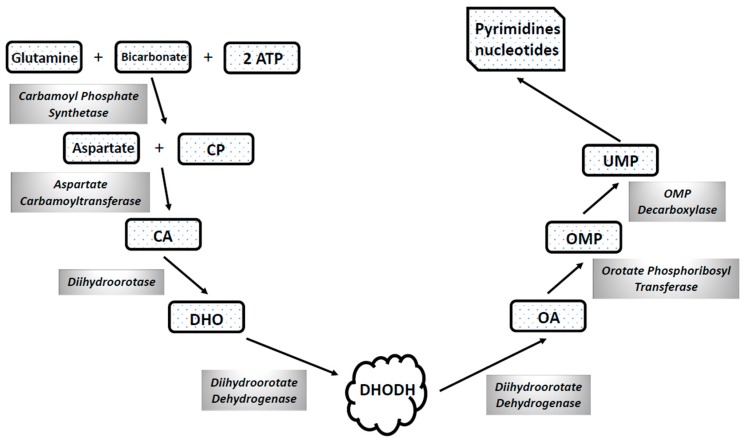
Schematic representation of the main steps of pyrimidine biosynthesis. The initial step is catalyzed by Carbamoyl Phosphate Synthetase catalyzing the generation of carbamoyl phosphate (CP), starting from one molecule of glutamine and bicarbonate and two molecules of ATP. The second step involves the generation of carbamoyl aspartate (CA) from one molecule of aspartate and CP, catalyzed by carbamoyl-transferase (CA). In the third step, dihydroorotase catalyzes the conversion of CA to diihydrooratate (DHO). In the fourth step, DHO is converted to oratate by the enzyme diihydrooratate dehydrogenase (DHODH). In the following step, orotate phosphoribosyltranferase converts OA to AMP (orothidine 5′ monophosphate). Finally, in the last step, OMP is converted to UMP and OMP decarboxylase (uridine monophosphate).

**Figure 5 cancers-11-00260-f005:**
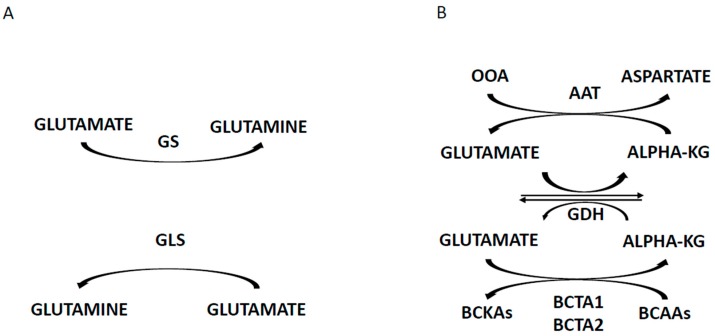
Main steps of glutamine metabolism. (**A**) Glutamine enters the cells via the SCL1A5 transporter (not shown) and contributes directly to nucleotide synthesis (not shown) or is converted to glutamate by glutaminase (GLS); the opposite reaction, the conversion of glutamate to glutamine, is catalyzed by glutamine synthetase (GS). (**B**) Glutamate is converted to α-ketoglutarate (alpha-KG) by glutamate dehydrogenase (GDH) or by aminotransferases (AAT).

**Figure 6 cancers-11-00260-f006:**
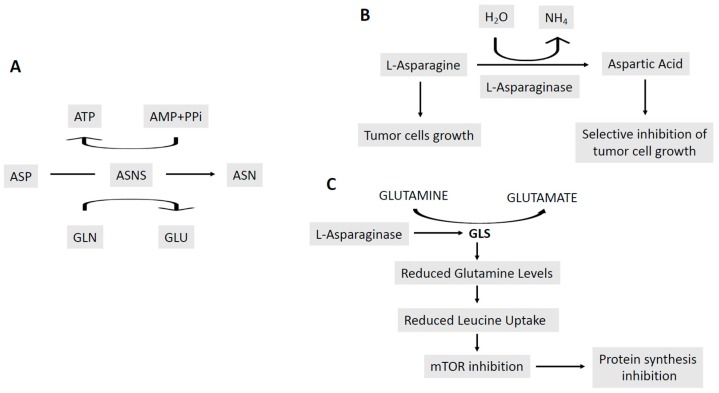
Asparagine metabolism. (**A**) Reaction catalyzed by the enzyme asparagine synthetase (ASNS): aspartate (ASP) and glutamine (GLN) are converted to asparagine (ASN) and glutamate (GLU). (**B**) The enzyme L-Asparaginase promotes the conversion of asparagine to aspartic acid, thus depriving tumor cells of L-asparagine and inhibiting tumor cell growth. (**C**) In AML cells, particularly those with low SANS activity, it was discovered that L-asparaginase inhibits leukemic cell proliferation for its glutaminase activity, leading to a depletion of glutamine intracellular levels, which in turn lower leucine uptake by leukemic cells, with consequent inhibition of mTOR activity and protein synthesis.

**Figure 7 cancers-11-00260-f007:**
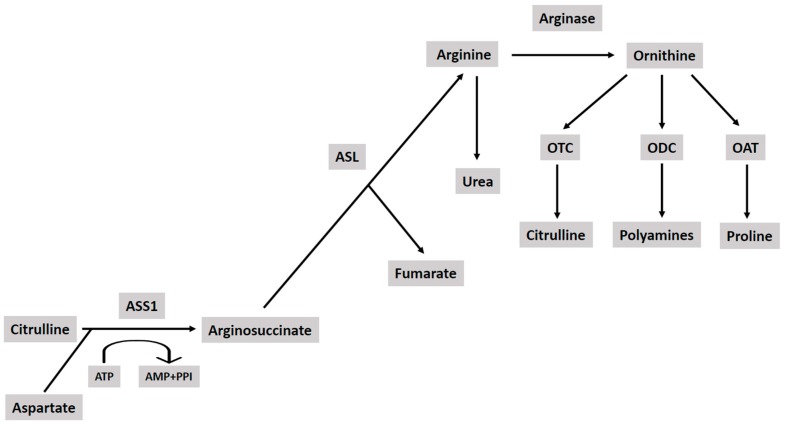
Main steps of arginine metabolism. Arginine may be derived from diet or endogenously synthesized from the amino acid citrulline as an immediate precursor. This de novo biosynthetic pathway involves the conversion of citrulline to arginine catalyzed: first by arginosuccinate synthase (ASS1) catalyzing the conversion of citrulline and aspartate to arginosuccinate, with the concomitant use of one molecule of ATP converted to AMP and PPi; then, by arginossuccinate lyase, catalyzing the conversion of one molecule of arginosuccinate into arginine and fumarate. The catabolism of arginine implies either the conversion of arginine to NO, catalyzed by nitric synthases (not shown) or the conversion of arginine to ornithine and urea catalyzed by arginase. Ornithine may generate citrulline through ornithine transcarbamylase (OTC), polyamines through ornithine decarboxylase (ODC) or proline through ornithine amino transferase (OAT).

**Figure 8 cancers-11-00260-f008:**
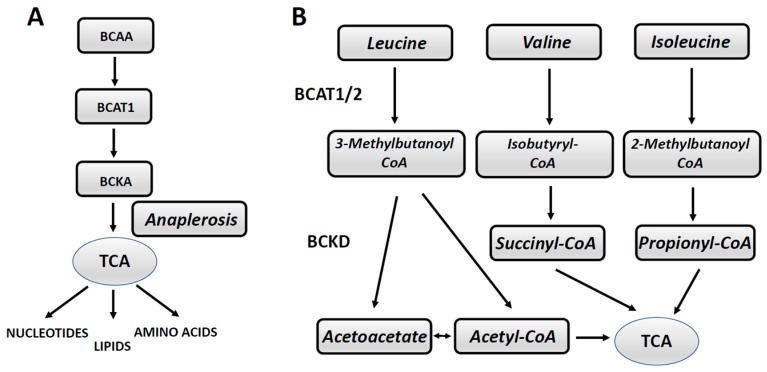
Main steps of branched chain amino acid (BCAA) metabolism. (**A**) The catabolism of BCAAs through various steps leads to the production of branched chetoacids (BKAs) that enter the TCA cycle and represents the source for nucleotide, lipids and amino acid biosynthesis allowing cancer cells to adapt to various and variable microenvironmental conditions. (**B**) Main steps of BCAA catabolism. BCAA catabolic reactions, starting from leucine, valine and isoleucine are catalyzed by a series of enzymes, starting with transamination catalyzed by BCAA aminotransferases (BCAT 1 and 2), then with decarboxylation by keto acid dehydrogenase (BCAD), with production of enter products succinyl-CoA and acetyl-CoA entering to TCA cycle.

**Table 1 cancers-11-00260-t001:** Prognostic classification of acute myeloid leukemias according to the European Leukemia Network. Co-occurring mutations and other genetic alterations are reported in parenthesis.

Prognostic Group	AML Subtypes (Concurrent Mutation Frequency)	Probability of CR (%)	Probability of Relapse (%)
Low Risk	**t(8;21)(q22;q22); RUNX1-RUNX1T1** 1–6% (KIT 25%, NRAS 20%, Cohesin 20%, ZBTB7A 20%, ASXL1 10%) **inv(16)(p13.1q22)/ t(16;16)(p13.1;q22)** **CBFB-MYH11** 1–6% (NRAS 40%, KIT 35%, FLT3-TKD 20%, KRAS 15%) **t(15;17)(q22;q12); PML-RARA** 5–13% (FLT3-ITD 35%, FLT3-TKD 15%, WT1 15%) **NK and NPM1^+^/FLT3-ITD^−/WT^** (DNMT3A 50%, FLT3-ITD 40%, cohesion 20%, NRSA 20%, IDH1 15%, IDH2 R140 15%, TET2 15%) **NK and CEBPA^+/+^** (biallelic) 1–5% (GATA2 30%, NRAS 30%, WT1 20%, CSF3R 20%, 9q^−^ 15%)	80–95	5–40
Intermediate Risk	**NK and NPM1^−^/FLT3-ITD^−/low^** (without adverse-risk genetic lesions) **NK and NPM1^+^/FLT3-ITD^high^** **t(9;11)(p21.3;q23.3); MLLT3-KMT2A** 1% (NRAS 20%, FLT3-TKD 10%, FLT3-ITD 5%, +8/+8q 30%) **Cytogenetic abnormalities not in the low or high risk groups** **t(8;21), inv(16) and t(16;16) with KIT mutations;BCFB-MYH11** 1–6% (NRAS 40%, KIT 35%, FLT3-TKD 20%, KRAS 15%, +8/+8q 15%)	50–80	50–80
High Risk	**t(6;9)(p23;q34.1); DEK-NUP214** 1% (FLT3-ITD 70%, KRAS 20%) **t(3;3)(q21.3q26.2), inv(3)(q21.3q26.2); GATA2, MECOM (EVI1)** 1% (NRAS 30%, KRAS 15%, PTPN11 20%, GATA2 15%, SF3B1 20%, ETV6 15%, PHF6 15%, RUNX1 10%, ASXL1 10%) **ASXL1^mut^** 5–15% (RUNX1 20%, IDH2-R140 13%, +8 15%) **t(9;22)(q34.1;q11.2); BCR-ABL1** **monosomy 7 (-7)** **monosomy 5 (-5)** **deletion of long arm (q) of chromosome 7 (−7q)** **abnormalities of 3q, 17p, 11q** **multiple cytogenetic abnormalities (≥3–5)** **NK and NPM1^−^/FLT3-ITD^high^** **RUNX^mut^** 5–20% (SRSF2 25%, ASXL1 20%, KMT2A 20%, IDH2-R140 12%, −7/7q^−^ 10% **TP53^mut^** (complex karyotype 68%, −5/5q 47%, −7/7q 44%, −12/12p 17%, +8/8q 16%, c-kit 15%, CEBPA 15%)	<50	>90

NK (normal karyotype); CBFB-MYH11: core-binding factor subunit beta/myosin 11; PML-RARA: promyelocytic leukemia-retinoic receptor alpha; CR: complete remission; KIT: kinase-inducing tumors; TKD: tyrosine kinase domain.

**Table 2 cancers-11-00260-t002:** Clinical trials involving the use of BCL-2 inhibitors.

Clinical Trial. Gov Identifier	Clinical Trial	Molecular Target	Phase	Relevant Preliminary Results
NCT01994837	A phase 2 study of ABT-199 in subjects with AML	BCL-2	II	ORR 38% CR 19%
NCT02287233	A phase 1/2 study of venetoclax in combination with low-dose cytarabine in treatment-naïve subjects with acute myelogenous leukemia who are ≥ 60 years of age and who are not eligible for standard anthracycline-based induction therapy	BCL-2	I/II	CR 62%
NCT03069352	A study of venetoclax in combination with low dose cytarabine versus low dose cytarabine alone in treatment naïve patients with AML who are ineligible for intensive chemotherapy	BCL-2	III	Phase II: ORR 64%
NCT02993523	A study of venetoclax in combination with azacytidine versus azacytidine in treatment naïve subjects with AML who are ineligible for standard induction therapy	BCL-2	III	Phase II: ORR 67%
NCT03404193	Venetoclax in combination with 10-day decitabine in newly diagnosed elderly or relapsed/refractory AML and relapsed high-risk myelodysplastic syndrome	BCL-2	II	
NCT03214562	Study of the BCL-2 inhibitor venetoclax in combination with standard intensive AML induction/consolidation therapy with FLAG-IDA in patients with newly diagnosed or relapsed/refractory AML	BCL-2	IB/II	
NCT0267044	A study of venetoclax in combination with cobimetinib and venetoclax in combination with Idasanutlin in patients aged ≥ 60 years with relapsed or refractory AML who are not eligible for cytotoxic therapy	BCL-2 MEK MDM2	IB/II	ORR (Ven + Cobi) 18% ORR (Ven + Ida) 20% CR (Ven + Cobi) 18% CR (Ven + Ida) 15%
NCT02203773	Study of ABT-199 in combination with azacytidine or decitabine (chemo combo) in subjects with AML Group A: venetoclax + decitabine Group B: venetoclax + azacitidine Group C:venetoclax + decitabine and posaconazole (CYP3A inhibitor)	BCL-2	IB	CR or CRi 61% (overall) CR or CRi 60% (group A 65%; group B 59%; group C 67%)
NCT03194932	Study of venetoclax in combination with chemotherapy in pediatric patients with refractory or relapsed AML or acute leukemia of ambiguous lineage	BCL-2	I/II	
ACTRN12616000445471	A phase IB clinical evaluation of venetoclax in combination with chemotherapy (cytarabine+idarubicin) in older patients with acute myeloid leukemia	BCL-2	IB	
NCT02391480	A phase 1 study evaluating the safety and pharmacokinetics of ABBV-075 in subjects with advanced cancer (expansion combination treatment cohort of ABBV-075 and venetoclax)	BET BCL-2	I/II	
NCT03537482	A phase i study of safety, tolerability, pharmacokinetic and pharmacodynamics property of orally administered APG-2575 in patients with hematologic malignancies	BCL-2	I	
CT03082209	TRAIL receptor agonist ABBV-621 in subjects with previously treated solid tumors and hematologic malignancies (group II: ABBV-621 and venetoclax)	BCL-2 TRAIL-R	I	

**Table 3 cancers-11-00260-t003:** Clinical studies involving MCL-1 inhibitors.

Clinical Trial. Gov Identifier	Clinical Trial	Molecular Target	Phase and Patients
NCT 02979366	Phase I study of S64315 administered intravenously in patients with acute myeloid leukemia and myelodysplastic syndrome	MCL-1 (BH3-groove)	I, relapsed/refractory AMLs or secondary MDSs
NCT o2675452	AMG 176 first in human trial in subjects with relapsed or refractory multiple myeloma and subjects with relapsed or refractory AML	MCL-1 (BH3-groove)	I, relapsed/refractory multiple myeloma and AML
NCT 03218683	Study of AZD 5991 in relapsed or refractory hematologic malignancies	MCL-1	I, relapsed/refractory hematologic malignancies

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
