# Peer review of "Emerging Therapies for Acute Myelogenus Leukemia Patients Targeting Apoptosis and Mitochondrial Metabolism"

_cancers, 2019, doi:10.3390/cancers11020260_

Round 1
Reviewer 1 Report
The authors have answered to all reviewer's comments and added new paragrahs to the manuscript as requested by reviewers.
No additional comments from my side.
Reviewer 2 Report
I do not have any further comments on this manuscript a I do recommend to accept it in its current state.
This manuscript is a resubmission of an earlier submission. The following is a list of the peer review reports and author responses from that submission.
Round 1
Reviewer 1 Report
The paper is of good quality. Pending a professional English proofreading, I recommend it for publication.
Reviewer 2 Report
In the manuscript entitled "Emerging therapies for AML patients targeting apoptosis and mitochondrial metabolism” Castelli and colleagues. The manuscript provides a nice overview of therapies targeting the extrinsic and intrinsic apoptosis pathways as well as mitochondrila metabolism in the AML context. To increase the relevance of the review, it is advised to include additional data especially it would be relevant to discuss the roles of autophagy in the control of apoptosis and mitochondrial metabolism. In fact, BCL-2 inhibitors (especially BH3-mimetic molecules) do not only regulate apoptosis but also autophagy (through inducing the dissociation of the autophagic protein Beclin 1 from BCL-2). Moreover, autophagy is regulated by the mTORC1/AMPK pathway which plays a key role in cellular metabolism and in fact encouraging results were obtained in studies where this pathway was targeted in AML.
In detail, the following changes are proposed:
Line 370 page 10: Please explain in more detail the link between SPHK1 and the BH3 only protein.
Line 491 page 12: Please mention the name of targets of miR-106 that are involved in oxidative phosphorylation
Page 13: Please talk about the link between autophagy and mitochondrial metabolism in the context of AML therapy
Line 538 : page 3 : Please cite and discuss the following paper related the role L-asparaginase in the control of glutamine metabolism in AML « Inhibiting glutamine uptake represents an attractive new strategy for treating acute myeloid leukemiaLise Willems et al. Blood 2013 122:3521-3532 »
Line 628 page 15: Please mention the possible role of autophagy in resistance of leukemia cells to TRAIL and cite the following paper : The Autophagy Machinery Controls Cell Death Switching between Apoptosis and Necroptosis.Goodall ML & al. Dev Cell. 2016 May 23;37(4):337-349.
Line 697 page 16: Emricasan is a pan caspase inhibitor but not « a caspase-8 activator » . Please correct the sentence.
Finally, given that a large part of the manuscript is related to therapies for targeting the intrinsic or extrinsic apoptotic cell death pathway, a figure presenting this pathway would be helpful.
Reviewer 3 Report
The review manuscript submitted by Castelli et al. focuses on an overview of emerging therapeutic approaches for AML patients that intrinsic and extrinsic apoptotic signaling and might also exploit metabolic abnormalities in AML cells. The manuscript is fairly informative with lots of current data on Bcl-2, p53, TRAIL and metabolism targeted drugs and approaches. However, due to this accumulation of a significant amount of data also somehow lacks clarity and might be difficult to comprehend. It needs clarification and re-shuffling in some sections.
Major points:
1. As Bcl-2 and other anti-apoptotic members-targeting therapies apparently represent one of the major hopes for (not just) AML patients, these proteins and their signaling pathways should merit better and deeper introduction supported by some figures/tables and not just one paragraph… Some minor points – BH3-only effectors tBid, Bim and Puma possess both activating (Bax/Bak) and sensitizing (anti-apoptotic Bcl-2 proteins) functions, TP53 is not per se initiator of Bcl-2 signaling (it just transactivates two of the BH-only proteins Puma and Noxa and its role in transcription independent inactivation of Bcl-2 proteins is questionable). BH3-profiling should be also explained earlier (e.g. already on pg. 4) than in conclusions. Tables I and II should be merged as they are both focused on targeting anti-apoptotic Bcl-2 proteins.
2. In the chapter on metabolism targeting in AML cells would be beneficial to logically separate OXPHOS and glycolysis targeting and also present a comprehensive table on these emerging drugs and if possible clinical trials.
3. Chapters on TRAIL and caspase-8 (and likely also on ONC 2011) should be merged and presented in more understandable mode – i.e. with full mechanical introduction on TRAIL-induced signaling (apoptosis, necroptosis, non-death signaling), in which caspase-8 activation and Bid cleavage play an essential role and only then present data on their possible targeting with novel agents. A figure on TRAIL-induced signaling might be also quite illustrative. ONC 201 likely induces also the unfolded protein response (activation of ATF4 and CHOP) – should be at least mentioned.
Minor points:
1. Many typos throughout the manuscript.
2. Venetoclax is ABT-119 similarly as Novitoclax is ABT-263 - should be mentioned.
3. In Table 1 would be informative to include additional column with staring date and status of presented clinical trials. ACTRN12616000445471 is run in Australia and not under NCT register.
4. Hypomethylating agents is incorrect formulation as 5-Azacytidine and Decitabine are cytidine analogues that upon their incorporation into DNA inhibit DNA methytransferases and thus indirectly induce S-pkase arrest and DNA hypomethylation – should be explained.
5. Mode of action of Mcl-1 targeting agents UMI-77, ML3M, A-1210477 is not described.
6. On pg. 16 – BAX and BAK and XIAP – XIAP is inhibited by SMAC/DIABLO, which is released from the mitochondrial intermembrane space through BAX/BAK channels in the OMM.
7. Phase II dosing of ONC 201on pg. 17 is an unnecessary detail…
8. Conclusion part is too long – should be more strait and comprehensive.